# Integrated Model Selection and Scalability in Functional Data Analysis Through Bayesian Learning

Wenzheng Tao [1,2], Sarang Joshi [2,3,*] and Ross Whitaker [1,2]

1   School of Computing, The University of Utah, Salt Lake City, UT 84112, USA; wztao@cs.utah.edu (W.T.); whitaker@sci.utah.edu (R.W.)
2   Scientific Computing and Imaging Institute, The University of Utah, Salt Lake City, UT 84112, USA
3   Biomedical Engineering, The University of Utah, Salt Lake City, UT 84112, USA
*   Correspondence: sarang.joshi@utah.edu

**Abstract:** Functional data, including one-dimensional curves and higher-dimensional surfaces, have become increasingly prominent across scientific disciplines. They offer a continuous perspective that captures subtle dynamics and richer structures compared to discrete representations, thereby preserving essential information and facilitating the more natural modeling of real-world phenomena, especially in sparse or irregularly sampled settings. A key challenge lies in identifying low-dimensional representations and estimating covariance structures that capture population statistics effectively. We propose a novel Bayesian framework with a nonparametric kernel expansion and a sparse prior, enabling the direct modeling of measured data and avoiding the artificial biases from regridding. Our method, Bayesian scalable functional data analysis (BSFDA), automatically selects both subspace dimensionalities and basis functions, reducing the computational overhead through an efficient variational optimization strategy. We further propose a faster approximate variant that maintains comparable accuracy but accelerates computations significantly on large-scale datasets. Extensive simulation studies demonstrate that our framework outperforms conventional techniques in covariance estimation and dimensionality selection, showing resilience to high dimensionality and irregular sampling. The proposed methodology proves effective for multidimensional functional data and showcases practical applicability in biomedical and meteorological datasets. Overall, BSFDA offers an adaptive, continuous, and scalable solution for modern functional data analysis across diverse scientific domains.

**Keywords:** functional data analysis; principal component analysis; dimension reduction; sparse Bayesian learning; variational Bayesian inference; nonparametric methods; model selection

## 1. Introduction

The emergence of big data across diverse fields, such as biomedicine, finance, and physical modeling, has catalyzed the need for advanced analytical methodologies capable of handling complex, high-dimensional datasets that conventional discrete data analysis approaches cannot always process effectively. Such datasets often require analysis that captures and interprets their continuous and potentially high-dimensional complexities—a central promise of *functional data analysis* (FDA) [1,2]. Foundational work established FDA's capacity to treat each observation as an entire function [3], be it a curve, surface, or higher-dimensional structure, thereby extracting richer insights than conventional discrete point analyses. Over the past decade, FDA's scope has widened significantly to accommodate

high-dimensional applications, with theoretical and computational advances emerging across various contexts [4,5].

A pivotal technique within FDA is *functional principal component analysis* (fPCA), which serves as a dimension reduction tool similar to classical PCA and factor analysis. Unlike classical PCA, however, fPCA operates, in principle, in an infinite-dimensional function space to capture dominant modes of variation and reduce complexity [6]. Despite its conceptual elegance, existing fPCA and similar FDA models often assume that data are observed on a shared, finite grid, often relying on heuristic imputation or posterior estimation to handle any missing entries [7–9]. This assumption conveniently facilitates the adoption of established linear algebraic methods but compromises the integrity of FDA by introducing significant information loss and high-computational demands in high-dimensional applications.

An ideal approach would represent each function at its naturally sampled measurement points rather than forcing all observations onto a shared grid, thus preserving crucial information and avoiding the need for heuristic resampling. This point is critical when considering that, given only a finite number of data points, infinitely many functions can interpolate these points, each reflecting different inductive biases about smoothness or shape [10]. Conventional smoothing or regridding methods (e.g., polynomial interpolation) introduce biases that may distort the underlying function's actual behavior. In contrast, we will achieve more accurate and unbiased predictions by concurrently updating the function estimation and the population-level statistics governing the estimation, such as those encoded in the covariance operator. Such an approach requires the direct modeling of the function from its original measurement points, rather than imposing artificial grids.

To mitigate these limitations, several studies have proposed alternative strategies. For instance, Ref. [3] developed a nonparametric technique for the estimation of the mean and covariance for functional data under smoothness assumptions while also discussing a continuous formulation and the necessary discretization in practical applications. In [11,12], the authors extended fPCA to sparse and irregular longitudinal designs by smoothing the covariance estimate and then discretizing. Nonetheless, classical discretization steps often result in significant information loss and computational burdens.

As functional data's size and complexity grew, researchers turned to flexible basis expansions, including sinusoids (Fourier), wavelets, polynomials, and B-splines, for a finite-dimensional representation of functional data that is convenient and accurate in computation, avoiding the drawbacks of explicit approximation and resampling [2,13,14]. However, a core challenge remains in selecting a suitable model. For instance, researchers must choose the number and form (e.g., smoothness), along with the dimensionality of the representational subspace. In approximation, the placement of basis functions is also essential. Evenly spaced nodes remain popular for their simplicity but may be suboptimal. Alternative node allocations may be better, such as Chebyshev nodes for superior accuracy [15] or sparse grids to reduce the combinatorial growth of the computational complexity [16].

Existing studies tend to rely on choosing the hyperparameters manually [6] or on cross-validation [3,14,17], which are known to be computationally prohibitive. Others employ approximated cross-validation [13] or marginal likelihood [8], but these still require the exhaustive testing of all candidate models. Methods with sparse Bayesian priors [7,18] for model selection allow model selection with a single optimization. In [19,20], the authors use shrinkage or sparse priors for data-adaptive basis selection to ensure minimal but effective sets of basis functions. Notably, Ref. [21] proposed the Bayesian and Akaike information criteria, demonstrating state-of-the-art performance in simulation studies for sparse and dense functional data.

In addition, probabilistic FDA emerges as a sophisticated adaptation of probabilistic methods tailored to incorporate the flexibility of latent variable models to manage functional data. A Bayesian latent factor regression model (LFRM) [18], for example, extends conventional regression to accommodate complex structures and dependencies in functional data, providing a robust framework to handle the complexities inherent in functional data. However, these Bayesian approaches are often limited by the computational demands of Monte Carlo methods in high dimensions [8]. To address increasingly high-dimensional FDA problems, recent efforts have emphasized scalability. For instance, Ref. [6] introduced FDA for images with a fixed basis or grid. In [17], they further reduced the complexity in 2D fPCA via tensor product B-splines. Meanwhile, Ref. [22] applied a Bayesian framework with basis expansion, adaptive regularization, and Gibbs sampling to 2D functional data in the form of EEG studies on children with autism. Furthermore, Ref. [23] leveraged a parsimonious basis representation and variational Bayes to achieve computational efficiency, making it suitable for 3D brain imaging data. A Bayesian nonparametric model [24] leverages variational inference for efficient computation in high-dimensional functional time series and uses an Indian buffet process to automatically select latent factors. Nonetheless, it focuses on 1D functional observations with temporal dependencies and a common sampling grid.

In parallel, the broader method of principal component analysis (PCA) remains a fundamental and effective tool. Classical PCA, rooted in eigendecomposition [25], effectively extracts dominant modes of variation in many settings but does not inherently accommodate the probabilistic nature of real-world data and their inherent uncertainties. Thus, Ref. [26] introduced *probabilistic PCA* (PPCA), which incorporates a probability distribution to manage these uncertainties more effectively. PCA has since evolved to address missing data [27], model selection [28], and complex data types [29]. In the context of functional data, these concepts motivate new approaches that unify probabilistic methodologies, latent factor models, and kernel expansions for continuous domains [1].

Within Bayesian machine learning, various priors have been proposed for sparse or robust formulations of PCA. Specifically, *sparse Bayesian learning* (SBL) [30], with its mechanism automatic relevance determination (ARD) [31,32], has proven adept in promoting parsimonious solutions [33]. SBL has emerged in Bayesian PCA [28], applying an iterative method to evaluate the relevance of each component and select the internal dimensionality by disregarding the redundant ones. In [34], the authors applied SBL to optimize the combination of base kernels to enhance model performance. A matrix completion method [35] uses ARD to select the factorization rank and dual graph priors to promote smoothness along rows and columns for the effective interpolation of missing entries, although they are more relaxed than a strict continuity constraint for FDA. These methods often exploit variational techniques or accelerated optimization [36], thereby balancing model complexity with computational tractability. In functional data contexts, where representations are infinite-dimensional, SBL offers a compelling framework for advanced FDA methods by efficiently handling sparse expansions and adaptively adjusting the model complexity.

In summary, despite these efforts to advance functional data analysis, several challenges persist. Existing methods often exhibit limitations in accuracy and efficiency when sampling is sparse, automatic model selection is essential, and the dimensionality is high [23]. Concurrently, probabilistic PCA and SBL frameworks illustrate powerful strategies to incorporate versatility and adaptivity for such data complexities, while their adaptation to FDA is still evolving. These gaps underscore the need for a robust, flexible, and computationally feasible approach, unifying ideas from FDA, PPCA, and SBL, that manages the continuous and high-dimensional nature of modern datasets.

### 1.1. Contributions

This manuscript proposes a novel *Bayesian framework for functional principal component analysis* that leverages nonparametric kernel expansions, sparse Bayesian learning for model selection, and efficient variational inference (VI). We abbreviate the proposed method as *BSFDA* (Bayesian scalable functional data analysis). (The code is available at https://github.com/WeeenZh/BSFDA, accessed on 21 March 2025). *BSFDA* addresses critical gaps in existing FDA techniques with irregular sampling, high-dimensional scalability, and the selection of both basis functions and principal components. Specifically, our approach offers the following:

- **Joint selection of optimum latent factors and sparse basis functions**: This eliminates constraints on parametric representation dimensionality, avoids information loss from discretization, and extends naturally to higher dimensions or non-Euclidean spaces through nonparametric kernel expansion. It further enhances the interpretability by adaptively choosing the model complexity without testing multiple models separately. We achieve these improvements using a Bayesian paradigm that provides robust and accurate posterior estimates while supporting uncertainty quantification.

- **Scalability across domain dimensionality and data size:** The proposed method uses VI for faster computation compared to Markov chain Monte Carlo (MCMC) methods, while still being accurate in terms of the estimation of the intrinsic dimensionality and overall covariance structure. BSFDA reduces the overall computation by partitioning the parameters into smaller update groups and introducing a slack variable to further subdivide the weighting matrix (which is part of the kernel structure) into even smaller parts [18], updating fewer blocks at a time and considering all model options. Introducing a slack variable makes the optimization process more efficient by separating different variable groups. This approach scales well with the data size and works efficiently even with large, complex datasets. We demonstrate this on the 4D global oceanic temperature dataset (ARGO), which consists of 127 million data points spanning across the globe for 27 years, with depths of up to 200 m [37].

### 1.2. Outline

Together, these contributions position our work at the intersection of functional principal component analysis [1] and sparse Bayesian learning [30], enabling the robust, flexible, and computationally feasible analysis of high-dimensional functional data. The remainder of this paper is organized as follows. In Section 2, we describe the proposed Bayesian functional PCA framework in detail, highlighting the nonparametric kernel expansions and sparse Bayesian priors. Next, in Sections 3 and 4, we discuss the variational inference procedure and the reduced active block updating step, illustrating how these techniques jointly provide scalability and accuracy. In Section 5, we then present extensive empirical studies demonstrating the factor selection accuracy, covariance operator estimation, and performance in large-scale 4D applications. Finally, in Section 6, we conclude with a discussion of potential extensions and open directions, emphasizing the broader implications of our work for large-scale, high-dimensional functional data analysis.

## 2. Formulation

### 2.1. Generative Model

In conventional fPCA, the data are assumed to be samples of functions that are elements of an appropriately smooth function space [1]. Using this assumption, data samples acquired at discrete points are typically interpolated to the continuum using tools such as splines, Fourier basis functions, or wavelets. In our work, we assume that the functions $y_i : R^M \mapsto R$ are outcomes of an $M$-dimensional stochastic process. As in classical

fPCA, we assume that $y_i$ is in a class of functions that can be approximated through a truncated, finite expansion, which is a weighted summation of $K$ kernel functions $\{\phi_k\}_{k=1}^K$:

$$y_i(x) = \sum_{k=1}^K w_{ik}\phi_k(x), \tag{1}$$

with $w_{ik}$ being random variables, $\phi_k(x) = \mathcal{K}(x, \mathbb{X}_k)$, $\mathcal{K}$ is the kernel function, and $\mathbb{X}_k$ is the $k$-th location.

Thus, $y_i(x)$s are realizations of a finite-dimensional stochastic process. Conventionally, the covariance operator of these functions is discretized, and the leading eigenfunctions form the estimated principal component loadings, following the Karhunen–Loève theorem [1]. By contrast, we establish a flexible Bayesian framework of fPCA following the form of probabilistic PCA [26], where the principal subspace is identifiable up to an arbitrary rotation and does not enforce the orthogonality of the loadings. Nevertheless, it is straightforward to recover classical eigenfunctions from the final covariance estimation over an arbitrary grid in this Bayesian framework.

The observed data are $P$ independent, noisy samples of the functions $\{y_i\}_{i=1}^P$ at index $\{X_i \in R^{N_i \times M}\}_{i=1}^P$, where $N_i$ is the number of measured samples for the $i$th function $y_i$ and $X_{in} \in R^M$ is the location of the $n$th measurement in the domain of the sample. The observations are $\{Y_i\}_{i=1}^P$, where $Y_{in} = y_i(X_{in}) + E_{in}$, where $E_{in}$ is white Gaussian noise of variance $\sigma^2$.

We also assume that the functions span a low-dimensional subspace of dimension $J << K$. We model this stochastically by assuming that the weights, $w_i \in R^K$, are given by $w_{ik} = \sum_{j=1}^J Z_{ij}W_{jk} + \bar{Z}_k$, where $W \in R^{J \times K}$ are the principal component loading coefficients and $Z_i \in R^J$ are standard normal variables [26]. This model is therefore

$$Y_{in} = \sum_{k=1}^K \left( \left( \sum_{j=1}^J \left( Z_{ij}W_{jk} \right) + \bar{Z}_k \right) \phi_k(X_{in}) \right) + E_{in} = (Z_iW + \bar{Z})\Phi_{i\cdot n} + E_{in}, \tag{2}$$

where $\Phi_{i\cdot n} = [\phi_1(X_{in}), \dots, \phi_K(X_{in})]^T$ are the evaluations of the basis functions at the $n$-th index of the $i$-th sample function.

The choice of the kernel family usually benefits from knowledge of the dataset's characteristics, such as the periodicity or domain geometry. Our framework is flexible across various kernel families, but we employ Gaussian kernels for both one-dimensional and multidimensional data by default, with the initial length-scale selection carried out through cross-validation, which will be refined through our sparse prior described below. To avoid disproportionately favoring larger length scales, we normalize each kernel's scaling coefficient using its square integral over the observational domain.

### 2.2. Sparse Prior

For effective model selection, we introduce a sparse prior over the coefficients of the basis functions [28]. The sparse prior in the proposed model is based on automatic relevance detection (ARD) [28]. ARD evaluates the importance of a feature with a precision parameter estimated from the data. The model uses $\{\alpha_j\}_{j=1}^J$ and $\{\beta_k\}_{k=1}^K$ for the numbers of components and basis functions, respectively, while $\eta$ signifies the overall magnitude of the mean coefficients:

$$\bar{Z}_k \sim \mathcal{N}(0, \eta^{-1}\beta_k^{-1}), \forall k = 1 : K \tag{3}$$

$$W_{jk} \sim \mathcal{N}(0, \alpha_j^{-1}\beta_k^{-1}), \forall j = 1 : J, k = 1 : K \tag{4}$$

In the model, $\alpha_j, \beta_k, \eta, \sigma^{-2}$ are all variables of precision parameters, coming naturally with a conjugate prior of Gamma distribution that facilitates efficient posterior optimization. The probabilistic graphical model is depicted in Figure 1. Setting $a_0, b_0$ to a small value yields a vague Gamma prior that approximates a noninformative (Jeffreys-type) prior.

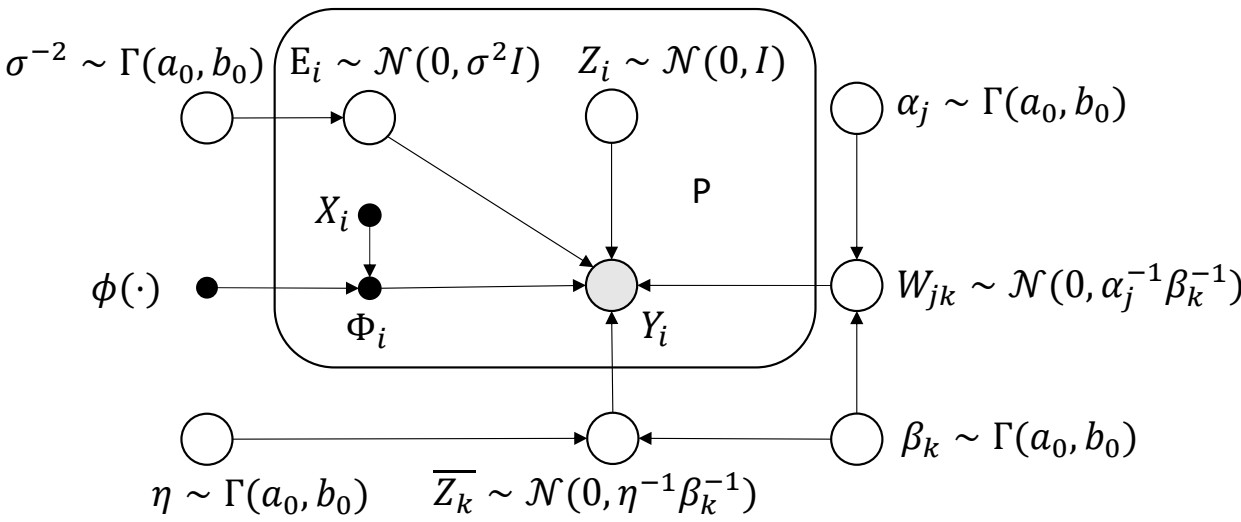

**Figure 1.** Probabilistic graphical model for the full model.

## 3. Methods

Based on the proposed formulation in Section 2, we estimate $\Pr[\Theta|X, Y, a_0, b_0]$, the posterior of the unobserved values $\Theta = \{Z, W, \bar{Z}, \sigma, \alpha, \beta, \eta\}$. This inference gives the point estimates of $\Theta$ and the posterior predictive distribution of new data. For notational convenience, $X, a_0$, and $b_0$ are omitted.

Using Bayes' theorem, $\Pr[\Theta|Y] = \frac{\Pr[Y|\Theta]\Pr[\Theta]}{\Pr[Y]}$, but the exact posterior distribution is intractable because the evidence $\Pr[Y] = \int \Pr[\Theta, Y]d\Theta$ is intractable. Therefore, an approximate inference strategy is proposed. To facilitate this, we utilize variational inference (VI) [38], choosing a surrogate density from a parameterized family, denoted as $\mathcal{Q}$, to approximate the posterior. Compared with classical methods such as Markov chain Monte Carlo (MCMC) sampling, VI is typically faster [38]. In our experiments, VI is about 85 times faster for the original Bayesian PCA formulation [28], as shown in Appendix F.2.

### 3.1. Variational Bayesian Inference

Variational inference optimizes $\mathcal{Q}$ by maximizing the lower bound $\mathcal{L}$ (minimizing the KL divergence between the actual and surrogate distributions):

$$\mathbb{E}_{\mathcal{Q}}\left[\ln \frac{\Pr[\Theta, Y]}{\mathcal{Q}(\Theta)}\right] = -\mathrm{KL}(\mathcal{Q}(\Theta)||\Pr[\Theta|Y]) + \ln \Pr[Y] \propto -\mathrm{KL}(\mathcal{Q}(\Theta)||\Pr[\Theta|Y]). \quad (5)$$

The mean field variational family is used for $\mathcal{Q}$. It simplifies the optimization by assuming that the surrogate posterior distributions are independent, allowing each variable in the posterior to be optimized independently: $\mathcal{Q}_\Theta = \prod_i \mathcal{Q}_{\Theta_i}$. The posterior for each variable is chosen to be conjugate, further simplifying the optimization. Thus, the posteriors of the component scores $Z$, the weighting matrix $W$, and the mean weights $\bar{Z}$ are normal distributions. Here, $W$ is vectorized via vec $(W)$ without altering its normality assumption. Meanwhile, the posteriors of the precision variables of noise $\sigma^{-2}$, components $\alpha$, basis functions $\beta$, and mean weights $\eta$ are Gamma distributions:

$$Q_Z(Z) = \prod_i Q_{Z_i}(Z_i) = \prod_i \mathcal{N}(Z_i|\mu_{Z_i}, \Sigma_{Z_i}) \tag{6}$$

$$Q_W(W) = \mathcal{N}(\text{vec}(W)|\mu_{\text{vec}(W)}, \Sigma_{\text{vec}(W)}) \tag{7}$$

$$Q_{\bar{Z}}(\bar{Z}) = \mathcal{N}(\bar{Z}|\mu_{\bar{Z}}, \Sigma_{\bar{Z}}) \tag{8}$$

$$Q_\sigma(\sigma) = \Gamma(\sigma^{-2}|a_\sigma, b_\sigma) \tag{9}$$

$$Q_\alpha(\alpha) = \prod_j Q_{\alpha_j} = \prod_j \Gamma(\alpha_j|a_{\alpha_j}, b_{\alpha_j}) \tag{10}$$

$$Q_\beta(\beta) = \prod_k Q_{\beta_k} = \prod_k \Gamma(\beta_k|a_{\beta_k}, b_{\beta_k}) \tag{11}$$

$$Q_\eta(\eta) = \Gamma(\eta, |a_\eta, b_\eta) \tag{12}$$

Update Steps

In mean field approximation using the surrogate posterior $Q_\Theta = \prod_i Q_{\Theta_i}$ conditioned on observations $Y$, the lower bound is maximized with respect to each unknown $\Theta_i$. With the conjugate prior, the optimal updates (denoted with "$\leftarrow$") make the moments of $Q_{\Theta_i}$ equal to the moments conditioned on the remaining parts of $Q_\Theta$ [38]:

$$Q_{\Theta_i} \leftarrow \frac{\exp\left(\mathbb{E}_{Q_{/\Theta_i}}[\ln(\Pr[Y, \Theta])]\right)}{\int \exp\left(\mathbb{E}_{Q_{/\Theta_i}}[\ln(\Pr[Y, \Theta])]\right) d\Theta_i} \tag{13}$$

From Equation (13), detailed update rules for each variable are presented subsequently, and the derivations of these formulas are given in the Appendix part.

**Updates for the parameters of the posterior for the precision of components** $Q_{\alpha_j}, \forall j = 1 : J$:

$$a_{\alpha_j} \leftarrow a_0 + \frac{K}{2}, \tag{14}$$

$$b_{\alpha_j} \leftarrow b_0 + \frac{1}{2}\sum_{k=1}^K \mathbb{E}_{Q_{/\alpha_j}}[W_{jk}^2 \beta_k] = b_0 + \frac{1}{2}\sum_{k=1}^K \left(\left(\Sigma_{W_{jk}} + \mu_{W_{jk}}^2\right)\frac{a_{\beta_k}}{b_{\beta_k}}\right), \tag{15}$$

where Equation (14) calculates the corrected degrees of freedom and Equation (15) calculates the corrected sum of squares. As $a_0$ and $b_0$ approach 0, the expectation of precision $\alpha_j$, which is $\mathbb{E}_{Q_{\alpha_j}}[\alpha_j] = \frac{a_{\alpha_j}}{b_{\alpha_j}}$, is exactly the inverse of the empirical or sample variance.

**Updates for the parameters of the posterior of the precision of the mean weights** $Q_\eta$:

$$a_\eta \leftarrow a_0 + \frac{K}{2}, \tag{16}$$

$$b_\eta \leftarrow b_0 + \frac{1}{2}\sum_{k=1}^K \mathbb{E}_{Q_{/\eta}}[\bar{Z}_k^2 \beta_k] = b_0 + \frac{1}{2}\sum_{k=1}^K \left(\left(\Sigma_{\bar{Z}k} + \mu_{\bar{Z}k}^2\right)\frac{a_{\beta_k}}{b_{\beta_k}}\right) \tag{17}$$

**Updates for the parameters of the posterior of the precision of basis functions** $Q_{\beta_k}, \forall k = 1 : K$:

$$a_{\beta_k} \leftarrow a_0 + \frac{J+1}{2}, \tag{18}$$

$$b_{\beta_k} \leftarrow b_0 + \frac{1}{2}\mathbb{E}_{Q_{/\beta_k}}[\bar{Z}_k^2 \eta + \sum_{j=1}^J W_{jk}^2 \alpha_j]$$

$$= b_0 + \frac{1}{2}\left(\left(\Sigma_{\bar{Z}kk} + \mu_{\bar{Z}k}^2\right)\frac{a_\eta}{b_\eta} + \sum_{j=1}^J \left(\left(\Sigma_{W_{jk}} + \mu_{W_{jk}}^2\right)\frac{a_{\alpha_j}}{b_{\alpha_j}}\right)\right) \tag{19}$$

**Updates for the parameters of the posterior of the mean weights** $Q_{\bar{Z}}$:

$$\Sigma_{\bar{Z}} \leftarrow \left( \mathbb{E}_{\mathcal{Q}_{/\bar{Z}}} \left[ \sigma^{-2} \sum_{i=1}^{P} \Psi_i + \eta \, \mathrm{diag}(\beta) \right] \right)^{-1} = \left( \frac{a_\sigma}{b_\sigma} \sum_{i=1}^{P} \Psi_i + \frac{a_\eta}{b_\eta} \, \mathrm{diag}(\frac{a}{b}) \right)^{-1}, \tag{20}$$

$$\mu_{\bar{Z}} \leftarrow \left( \mathbb{E}_{\mathcal{Q}_{/\bar{Z}}} \left[ \sigma^{-2} \right] \sum_{i=1}^{P} (Y_i - \mathbb{E}_{\mathcal{Q}_{/\bar{Z}}}[Z_i W] \Phi_i) \Phi_i^T \right) \Sigma_{\bar{Z}} = \left( \frac{a_\sigma}{b_\sigma} \sum_{i=1}^{P} (Y_i - \mu_{Z_i} \mu_W \Phi_i) \Phi_i^T \right) \Sigma_{\bar{Z}} \tag{21}$$

where $\mathrm{diag}(\beta)$ denotes the diagonal matrix with diagonal entries given by $\beta$. Equation (20) indicates that the eigenvectors of $\Sigma_{\bar{Z}}$ are solely determined by the sum of Gram matrices $\sum_{i=1}^{P} \Psi_i$, where $\Psi_i = \Phi_i \Phi_i^T$, while the eigenvalues of $\Sigma_{\bar{Z}}$ have a negative correlation with the scale of $\sum_{i=1}^{P} \Psi_i$, the prior $\eta \, \mathrm{diag}(\beta)$, and data-dependent term $\sigma^{-2}$. This is sensible because, for instance, large noise would result in large uncertainty in $\bar{Z}$. In Equation (21), the data residuals, excluding the component scores, are projected into the $K$-dimensional space through the inner product, with $\Phi_i$ and summed over all sample functions to calculate the mean weights.

**Updates for the parameters of the posterior of the weights $\mathcal{Q}_W$:**

$$\Sigma_{\mathrm{vec}(W)} \leftarrow \mathbb{E}_{\mathcal{Q}_{/W}} \left[ \sigma^{-2} \sum_{i=1}^{P} \left( \Psi_i^T \otimes (Z_i^T Z_i) \right) + \mathrm{diag}(\beta) \otimes \mathrm{diag}(\alpha) \right]^{-1}$$

$$= \left( \frac{a_\sigma}{b_\sigma} \sum_{i=1}^{P} \left( \Psi_i^T \otimes (\mu_{Z_i}{}^T \mu_{Z_i} + \Sigma_{Z_i}) \right) + \mathrm{diag}\left(\frac{a}{b}\right) \otimes \mathrm{diag}\left(\frac{c}{d}\right) \right)^{-1}, \tag{22}$$

$$\mu_{\mathrm{vec}(W)} \leftarrow \mathbb{E}_{\mathcal{Q}_{/W}} \left[ -\sigma^{-2} \sum_{i=1}^{P} \mathrm{vec}\left( \left( \Phi_i(\Phi_i^T \bar{Z}^T - Y_i^T) Z_i \right)^T \right)^T \right] \Sigma_{\mathrm{vec}(W)}$$

$$= -\frac{a_\sigma}{b_\sigma} \sum_{i=1}^{P} \mathrm{vec}\left( \left( \Phi_i(\Phi_i^T \mu_{\bar{Z}}{}^T - Y_i^T) \mu_{Z_i} \right)^T \right)^T \Sigma_{\mathrm{vec}(W)} \tag{23}$$

Equation (22) is similar to Equation (20), because it is correlated with $\Phi_i$, its prior $\mathrm{diag}(\beta) \otimes \mathrm{diag}(\alpha)$, and data-dependent terms $\sigma^{-2}$ and $Z_i$. In Equation (23), the data residual excluding the mean function is used to estimate the expectation of $W$.

**Updates for the parameters of the posterior of the component scores $\mathcal{Q}_{Z_i}$:**

$$H_{ijk} \leftarrow \mathbb{E}_{\mathcal{Q}_{/Z_i}}[W_j \Psi_i W_k^T] = \mathrm{Tr}(\mathbb{E}_{\mathcal{Q}_{/Z_i}}[W_k^T W_j] \Psi_i)$$

$$= \mathrm{Tr}\left( \left( \Sigma_{[W_k, W_j]} + \mu_{[W_j]}^T \mu_{[W_k]} \right) \Psi_i \right), \forall j = 1:K, k = 1:K, \tag{24}$$

$$\Sigma_{Z_i} \leftarrow \left( \mathbb{E}_{\mathcal{Q}_{/Z_i}}[\sigma^{-2} W \Psi_i W^T + I] \right)^{-1} = [\frac{a_\sigma}{b_\sigma} H_i + I]^{-1}, \tag{25}$$

$$\mu_{Z_i} \leftarrow \mathbb{E}_{\mathcal{Q}_{/Z_i}}[\sigma^{-2}(Y_i - \bar{Z}\Phi_i)\Phi_i^T W^T] \Sigma_{Z_i} = \frac{a_\sigma}{b_\sigma}(Y_i - \mu_{\bar{Z}}\Phi_i)\Phi_i^T(\mu_W)^T \Sigma_{Z_i}, \tag{26}$$

where $H_i$ is a temporary variable denoting the Gram matrix of weighted kernel functions $W\Phi_i$, and $\Sigma_{[W_k, W_j]}$ denotes the covariance between $W_k^T$ and $W_j$ in $\mathcal{Q}$.

**Updates for the parameters of the posterior of the noise $\mathcal{Q}_\sigma$:**

$$a_\sigma \leftarrow a_0 + \frac{1}{2} \sum_i N_i, \tag{27}$$

$$b_\sigma \leftarrow b_0 + \frac{1}{2} \mathbb{E}_{\mathcal{Q}_{/\sigma}} \left[ \sum_i ||Y_i - (Z_i W + \bar{Z})\Phi_i||_2^2 \right]$$

$$= b_0 + \frac{1}{2} \sum_i (Y_i Y_i^T - 2Y_i \left( \mu_{Z_i} \mu_W \Phi_i \right)^T - 2Y_i \left( \mu_{\bar{Z}} \Phi_i \right)^T + 2\mu_{Z_i} \mu_W \Psi_i (\mu_{\bar{Z}})^T$$

$$+ \mathrm{Tr}\left( \left( \Sigma_{\bar{Z}} + (\mu_{\bar{Z}})^T \mu_{\bar{Z}} \right) \Psi_i \right) + \frac{1}{2} \mathrm{vec}(H^T)^T \sum_i \mathrm{vec}\left( \mathrm{vec}(\Psi_i) \mathrm{vec}(\Sigma_{Z_i} + \mu_{Z_i}{}^T \mu_{Z_i})^T \right), \tag{28}$$

where $H$ is a temporary variable that is updated by

$$H_{j+kM} \leftarrow \mathbb{E}_{\mathcal{Q}_{/\sigma}}\left[\text{vec}(W_k W_j^T)^T\right] = \text{vec}(\Sigma_{[W_k,W_j]} + \mu_{[W_j]}^T \mu_{[W_k]})^T, \forall j = 1 : K, k = 1 : K \quad (29)$$

Nearly noninformative (vague) priors, i.e., with almost zero $a_0, b_0$, introduce an inherent identifiability ambiguity in our formulation—specifically, in the product of the precision parameters $\alpha, \beta$, and $\eta$ (Equations (20) and (22)). In our model, scaling $\alpha$ and $\eta$ by a specific factor while inversely scaling $\beta$ leaves the product (and hence the lower bound in Equation (5) unchanged. This inherent ambiguity can lead $\alpha, \beta$, and $\eta$ to converge to extreme values, thereby challenging the numerical stability during optimization. To mitigate this issue, we adopt a heuristic constraint to ensure that the smallest values of $\alpha$ and $\beta$ remain within one order of magnitude of each other. Specifically, we enforce $\left|\log_{10}\left(\frac{\min(\alpha)}{\min(\beta)}\right)\right| \leq 1$. If an update to any $\alpha_j$ or $\beta_k$ would violate this constraint, this particular update is skipped, and the rest of the parameters remain updated. This strategy does not alter the algorithm's overall structure but stabilizes the optimization by preventing unnecessary flexibility in the precision parameters.

### 3.2. Scalable Update Strategy

The scalability of our algorithm so far is primarily challenged by the need to optimize the variational lower bound, $\mathcal{L}$, over $K$ basis functions. As indicated by Equation (22), the time complexity is $\mathcal{O}(K^6)$ (or, alternatively, $\mathcal{O}(K^2 P \max_i(N_i))$, typically dominated by the former), which becomes prohibitive when $K$ is large. In practice, however, only a small subset of these basis functions is necessary for an accurate representation—those with non-negligible weights under our sparse prior.

To address this, we focus the updates on the subspace of active basis functions, denoted as $K^{(a)}$, which comprises only those functions with non-negligible weights. The remaining basis functions, whose influence is minimal, are held fixed during optimization. Furthermore, the number of active principal components is noted as $J^{(a)}$ and set equal to $K^{(a)}$, ensuring that the model spans the full range of possible ranks from 1 to $K^{(a)}$. Consequently, we optimize $\mathcal{Q}^{(a)}$ using updates derived with regard to the objective $K^{(a)}$-dimensional lower bound $\mathcal{L}^{(a)}$ as an efficient surrogate of the full updates of $\mathcal{Q}$ with regard to the full lower bound $\mathcal{L}$, using only $K^{(a)}$ active basis functions. Meanwhile, the active dimensionality of the model is adjusted dynamically during optimization by activating or deactivating basis functions based on their precision parameters. For clarity, variables associated with the active subspace are annotated with the superscript $(a)$ (e.g., $a_{\alpha_j}^{(a)} = a_0 + \frac{K^{(a)}}{2}$ versus $a_{\alpha_j} = a_0 + \frac{K}{2}$).

#### 3.2.1. Implicit Factorization

For notational clarity, we reorder the rows and columns of our parameter matrices to separate active components from inactive ones. Specifically, we partition the matrices as follows:

$$Z_i = \begin{pmatrix} Z_{iA} & Z_{iB} \end{pmatrix}, \bar{Z} = \begin{pmatrix} \bar{Z}_A & \bar{Z}_B \end{pmatrix}, \alpha = \begin{pmatrix} \alpha_A & \alpha_B \end{pmatrix}, \beta = \begin{pmatrix} \beta_A & \beta_B \end{pmatrix},$$
$$W = \begin{pmatrix} W_A & W_B \\ W_C & W_D \end{pmatrix}, \Phi_i = \begin{pmatrix} \Phi_{iA} \\ \Phi_{iB}, \end{pmatrix} \quad (30)$$

Here, the subscript $A$ denotes variables belonging to the active subspace (i.e., those corresponding to $K^{(a)}$ basis functions), and $B, C$, and $D$ denote the inactive components.

Notably, the cross terms $W_B$ and $W_C$ involve both active and inactive components; these are updated implicitly, as proven in the Appendix part.

Following the strategy in [39], a basis function is deemed inactive if its precision exceeds a high threshold, i.e., $\alpha_j > \epsilon^{-1}$ and $\beta_k > \epsilon^{-1}$ as $\epsilon \to 0$. In the limit, the inactive basis functions decouple from the active ones, leading to the following mean field factorization:

$$\mathcal{Q}_W = \mathcal{Q}_{W_A}\mathcal{Q}_{W_B}\mathcal{Q}_{W_C}\mathcal{Q}_{W_D} \tag{31}$$

$$\mathcal{Q}_{\bar{Z}} = \mathcal{Q}_{\bar{Z}_A}\mathcal{Q}_{\bar{Z}_B} \tag{32}$$

$$\mathcal{Q}_{Z_i} = \mathcal{Q}_{Z_{iA}}\mathcal{Q}_{Z_{iB}} \tag{33}$$

The factorization of $\alpha$ and $\beta$ was already obtained in Equations (10) and (11). These factorizations allow us to decouple the update for the active subspace with the proof provided in the Appendix part.

It implies that only updates for $\mathcal{Q}_{Z_{iA}}, \mathcal{Q}_{W_A}, \mathcal{Q}_{W_B}, \mathcal{Q}_{W_C}, \mathcal{Q}_{\bar{Z}_A}, \mathcal{Q}_{\alpha_A}, \mathcal{Q}_{\beta_A}, \mathcal{Q}_\sigma, \mathcal{Q}_\eta$ are required, as shown in Figure 2. This strategy reduces the computational complexity from $\mathcal{O}(K^6)$ to $\mathcal{O}(K^{(a)^6})$. Moreover, we initialize $W$ as an identity matrix and set the active $\alpha_A$ to all ones and the inactive $\alpha_B$ to infinite. In this way, we can initialize the remaining active dimensions $K^{(a)}$, e.g., $Z_{iA}$, using a modified, multi-instance version of a relevance vector machine [30], which performs fast analytical maximum-likelihood updates, as detailed in Appendix E in the Appendix part.

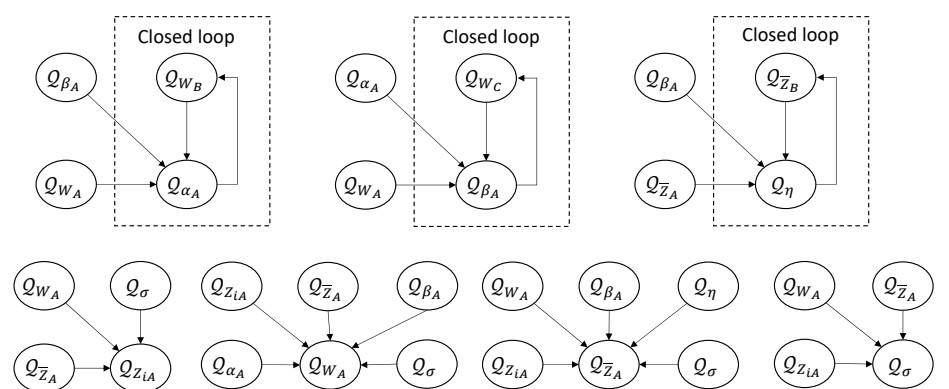

**Figure 2.** Diagrams of variational inference algorithm for all parameters. The top three diagrams each have a closed loop and a closed-form overall transfer function.

### 3.2.2. Low-Dimensional Lower Bound

This section shows how to optimize these active surrogates, e.g., $\mathcal{Q}_{\bar{Z}_A}$, using updates of $\mathcal{Q}^{(a)}$ with regard to the $K^{(a)}$-dimensional lower bound $\mathcal{L}^{(a)}$, which ultimately optimizes the full lower bound $\mathcal{L}$. To distinguish between the two, we denote the active surrogate posterior for the full model as $\mathcal{Q}_{\bar{Z}_A}$ and that for the reduced $K^{(a)}$-dimensional model as $\mathcal{Q}_{\bar{Z}_A}^{(a)}$. The active Gaussian surrogate posteriors are shared, e.g., $\mathcal{Q}_{\bar{Z}_A} = \mathcal{Q}_{\bar{Z}_A}^{(a)} = \mathcal{N}(\bar{Z}_A | \mu_{\bar{Z}A}, \Sigma_{\bar{Z}A})$. This sharing implies that updating $\mathcal{Q}^{(a)}$ is equivalent to updating $\mathcal{Q}$, so we set the moments of the active distributions of the full model to match those of the reduced model. However, the surrogate posterior Gamma distributions differ between the two models. For example, the update of $\mathbb{E}_{\mathcal{Q}^{(a)}}[\alpha_A]$ depends solely on $\mathcal{Q}_{W_A}$, whereas $\mathbb{E}_{\mathcal{Q}}[\alpha_A]$ also incorporates a cross term $\mathcal{Q}_{W_B}$ corresponding to the remaining $(K - K^{(a)})$ dimensions. This difference is reflected in how the scale parameters depend on the number of active versus total basis functions, as shown in Equations (14), (16) and (18). Nonetheless, we prove that, in the limit $\epsilon \to 0$, the fixed point of the $K^{(a)}$-dimensional updates of the complete surrogate $\mathcal{Q}$

equals that of the reduced surrogate $\mathcal{Q}^{(a)}$. Consequently, the updates for $\mathcal{Q}_{\alpha_A}$, $\mathcal{Q}_{\beta_A}$, and $\mathcal{Q}_\eta$ are derived directly from the expectations of the reduced model $\mathcal{Q}_{\alpha_A}^{(a)}$, $\mathcal{Q}_{\beta_A}^{(a)}$, $\mathcal{Q}_\eta^{(a)}$:

$$\mathbb{E}_\mathcal{Q}[\alpha_A] \leftarrow \mathbb{E}_{\mathcal{Q}^{(a)}}[\alpha_A] \Leftrightarrow b_{\alpha_j} \leftarrow \frac{a_{\alpha_j}}{a_{\alpha_j}^{(a)}} b_{\alpha_j}^{(a)}, \forall j \leq J^{(a)}, \tag{34}$$

$$\mathbb{E}_\mathcal{Q}[\beta_A] \leftarrow \mathbb{E}_{\mathcal{Q}^{(a)}}[\beta_A] \Leftrightarrow b_{\beta_k} \leftarrow \frac{a_{\beta_k}}{b_{\beta_k}} b_{\beta_k}^{(a)}, \forall k \leq K^{(a)}, \tag{35}$$

$$\mathbb{E}_\mathcal{Q}[\eta] \leftarrow \mathbb{E}_{\mathcal{Q}^{(a)}}[\eta] \Leftrightarrow b_\eta \leftarrow \frac{b_\eta^{(a)}}{a_\eta^{(a)}} a_\eta \tag{36}$$

These update Equations (34)–(36) are proven to optimize $\mathcal{L}$ in Theorems A1 and A2 in the Appendix part.

### 3.2.3. Heuristic for Activation of Basis Functions

The proposed method selects a relatively small set of basis functions from a potentially extensive set of possibilities. The computational costs are mitigated by recognizing that inactive basis functions do not interact with those that are active (with non-negligible weights). The inactive components are removed in the final model and excluded from the active subspace optimization. However, our method essentially optimizes over the full space, and thus we have the algorithm allowing for the reactivation of inactive basis functions to ensure optimization across the entire subspace. Due to computational constraints, we consider functions for activation sequentially rather than all at once. Thus, we propose Algorithm 1 to introduce unseen basis functions into the active set using a selective strategy akin to the heuristic approach described in [39].

The algorithm selects the top function, $\phi_{Bk}$, from the inactive basis functions $\{\phi_{Bk}\}_k$ by gauging their correlations with residuals and applying an angle-based threshold $\tau_{ang}$ relative to the subspace of $\phi_A$. The correlation with residuals for $\phi_{Bk}$ is measured by $\sum_i \left( \Phi_{iBk}(Y_i - \mathbb{E}_{\mathcal{Q}^{(a)}}[Z_{iA}W_A + \bar{Z}_A]\Phi_{iA})^T \right)^2$. The angle-based threshold ensures a meaningful distinction from active functions. Next, the current active surrogate posterior is expanded by a dimension for $\tilde{\phi_{Bk}}$, initiating optimization from the numerical maximum $\tau_{max}$. Postoptimization, the function is retained if it falls below $\tau_{max}$. Otherwise, the algorithm terminates. Efficiently, in trial optimization, the approach replaces one function with precision $\tau_{max}$, if present.

---

**Algorithm 1** Search for new basis functions to activate

---

Sort inactive basis functions $\{\phi_{Bk}\}_k$ by correlation with residuals.
Filter through $\{\phi_{Bk}\}_k$, selecting the most correlated one as $\phi_{Bk}$.
Copy current active surrogate $\mathcal{Q}^{(a)}(\Theta)$ posterior to $\mathcal{Q}_k^{(a)'}(\Theta)$.
Expand dimension in $\mathcal{Q}_k^{(a)'}(\Theta)$ for $\phi_{Bk}$.
Optimize $\mathcal{Q}_k^{(a)'}(\Theta)$ for three iterations using mean field approximation.
**if** expected precision is within threshold **then**
    $\mathcal{Q}^{(a)}(\Theta) \leftarrow \mathcal{Q}_k^{(a)'}(\Theta)$.
**end if**

---

## 4. Faster Variant

To enhance the computational efficiency of our primary algorithm, we introduce a faster variant, denoted as BSFDA$^{\text{Fast}}$. This approach leverages conditional independence among the columns of $W$, enabling separate updates and thereby reducing the computational complexity. Similar strategies have been described in [18,28]. The model is defined with an introduced variable $\zeta_i$ for the coefficient noise as follows:

$$\theta_i = Z_i W + \zeta_i, \tag{37}$$

$$\zeta_{ik} \sim \mathcal{N}(0, \varsigma_k^2 \beta_k^{-1}). \tag{38}$$

Similarly to the above, we assign a conjugate Gamma prior to the precision:

$$\varsigma_k^{-2} \sim \Gamma(a_0, b_0). \tag{39}$$

This formulation ensures that the columns of $W$ are conditionally independent, allowing the variational distribution to factorize as $\mathcal{Q}_\mathcal{W} = \prod_k \mathcal{Q}_{W_{\cdot k}}$, thereby facilitating separate updates for each column. Consequently, the time complexity is reduced from $\mathcal{O}(K^{(a)^6})$ to $\mathcal{O}(K^{(a)^3})$.

To align with the original model, it is necessary for $\zeta$ and the associated variance parameters $\varsigma$ to approach zero. Having $\varsigma$ too high would allow the coefficient noise to corrupt the signal, biasing the model toward underestimating the true signal levels, particularly because this noise operates in the coefficient space where it introduces smooth, correlated variation (low entropy, like signals), which is harder to eliminate than high-frequency white noise (maximum entropy). Injecting the same amount of noise leads to the unbiased estimation of the signals but increases the estimation variance. Conversely, as $\varsigma$ decreases, the columns of $W$ become dependent, violating the independence assumption inherent in variational inference. This dependency degrades the approximation quality and slows down the optimization process. Such dependency issues are well documented in both the variational inference and MCMC literature—with recent efforts addressing them via structured VI [38] or blocked/collapsed Gibbs sampling [40]. Empirical validations of this noise impact are conducted with both BSFDA[Fast] in Section 5.1 and with Bayesian PCA [28] in Appendix F.2 in the Appendix part.

To balance the trade-off between the optimization speed and accuracy, we adopt a strategy of gradually decreasing the values of $\varsigma_k$ during the optimization iterations. Specifically, we initialize $\varsigma_k$ with a relatively large value and linearly decrease it from $10^{-2}$ to $10^{-5}$ over the first half of the iterations. After reaching $10^{-5}$, $\varsigma_k$ is fixed for the remaining iterations. This gradual reduction ensures that the algorithm initially maintains its efficiency, with benefits from minimizing the interdependency among the columns of $W$ to accelerating convergence while later preserving the quality of the approximation by preventing the noise from obscuring the signal components. We unify the scales by scaling the basis functions so that $Z_i$ is standard normal and $W$ is an identity matrix in initialization. Empirical evaluations indicate that the strategy above is effective in most applications.

By implementing these modifications, BSFDA[Fast] offers a practical solution that substantially accelerates the algorithm without a significant loss in accuracy, making it well suited for large-scale, high-dimensional functional data analysis.

## 5. Results

The proposed method demonstrates its effectiveness through simulations and applications to observed datasets.

### 5.1. Simulation Results

In the simulations, we evaluate the functional data analysis performance in terms of model selection, the estimated covariance accuracy, and extendability to multidimensional domains.

The model selection metric is the accuracy in estimating the number of principal components, which is the dimension of the compact subspace of signal variations. The

configuration of the simulations in this section aligns with that established in [21], covering various scenarios. The simulated datasets are derived from a latent generative model with variables $\mathcal{Z}_i$ with dimension $r$ for the $i$-th sample function and noise corruption with a standard deviation of $\sigma$: $Y_i = \sum_{j=1}^{r}\left(\mathcal{Z}_{ij}f_j(X_i)\right) + g(X_i) + E_i$, $\mathcal{Z}_{ij} \sim \mathcal{N}(0, v_j)$, $E_j \sim \mathcal{N}(0, \sigma^2 I)$, where $\{f_j\}_{j=1}^{r}$ represent eigenfunctions, $\{v_j\}_{j=1}^{r}$ are the eigenvalues, and $g : R \mapsto R$ signifies the mean function. Here, we consider five scenarios.

**Scenario 1:** Data generated with $g = 5(x - 0.6)^2$, $r = 3$, $v = (0.6, 0.3, 0.1)$, $\sigma^2 = 0.2$, $f_1(x) = 1$, $f_2(x) = \sqrt{2}\sin(2\pi x)$, $f_3(x) = \sqrt{2}\cos(2\pi x)$. Here, $v_3 < \sigma^2$, i.e., the noise has larger variance than the smallest signal.

**Scenario 2:** Similar to Scenario 1, but the third eigenfunction is replaced by a function with higher frequencies $f_3(x) = \sqrt{2}\cos(4\pi x)$, and the principal component scores follow a skewed Gaussian mixture model. Specifically, the $j$-th component score has a one-in-three probability of following a $\mathcal{N}(2\sqrt{v_j/3}, v_j/3)$ distribution and a two-in-three probability of following $\mathcal{N}(-\sqrt{v_j/3}, v_j)$, for $j = 1, 2, 3$.

**Scenario 3:** Data generated with $g = 12.5(x - 0.5)^2 - 1.25$, $r = 3$, $v = (4, 2, 1)$, $\sigma^2 = 0.5$, $f_1(x) = 1$, $f_2(x) = \sqrt{2}\cos(2\pi x)$, $f_3(x) = \sqrt{2}\sin(4\pi x)$.

**Scenario 4:** Same as Scenario 3, but the component scores are generated from a Gaussian mixture model as in Scenario 2.

**Scenario 5:** Data from $g = 12.5(x - 0.5)^2 - 1.25$, $r = 6$, $v = (4, 3.5, 3, 2.5, 2, 1.5)$, $\sigma^2 = 0.5$, $f_1(x) = 1$, $f_{2k}(x) = \sqrt{2}\sin(2k\pi x)$ for $k = 1, 2, 3$, $f_{2k+1}(x) = \sqrt{2}\cos(2k\pi x)$ for $k = 1, 2$, $j$-th component score obeying $\mathcal{N}(0, v_j)$.

In each scenario, the simulations produce 200 sample functions. We investigate three cases with sparse, medium, and dense sampling by assigning the number of observations per sample function $N_i = \{5, 10, 50\}$. Each case in each scenario is repeated 200 times. The method's performance is compared to that of *fpca* from [13], the AIC and BIC in the 2022 release of *pace* [11], the modified AIC and BIC in [21], and all the competing methods in [21]. For *fpca*, we set the candidate numbers of basis functions as [8,10,15,20] and the candidate dimensions of the process as [2,3,4,5] for Scenarios 1–4 and [4,5,6,7,8] for Scenario 5. The other parameters are all set to the defaults. Due to its consistent overestimation of the true number of components—likely resulting from interference by correlated noise and less sparse precision priors—we exclude LFRM [18] from further comparisons (see Appendix F.1.1 in the Appendix part).

Each estimation uses 10 length scales of functions, which are selected using cross-validation and k-means clustering. This adaptive strategy allows the algorithm to choose distinct length scales at different locations of the definition domain, thereby accommodating varying smoothness characteristics inherent in complex functional data—a level of flexibility that is not possible when using a regular grid that forces a single length scale across the entire domain [18]. Sparse sampling in Scenario 5 uses five length scales to avoid overfitting. Figure 3 shows the length scales and centers of the selected kernel basis functions for three different numbers of sample points, $N_i$, in a random repetition of Scenario 5. The results reveal that the selected length scales mainly concentrate around 0.07, with a few as high as 0.35—suggesting that the lower length scales capture finer, high-frequency variations. The higher length scales model the overall, lower-frequency quadratic mean structure and the constant baseline component. Furthermore, the estimated density functions of the selected length scales exhibit consistent patterns across the three sampling densities, and the method selects 9, 11, and 12 basis functions, respectively, demonstrating the algorithm's adaptive fidelity and complexity based on the available observations. The Appendix part showcases the uncertainty evaluation in Figure A2.

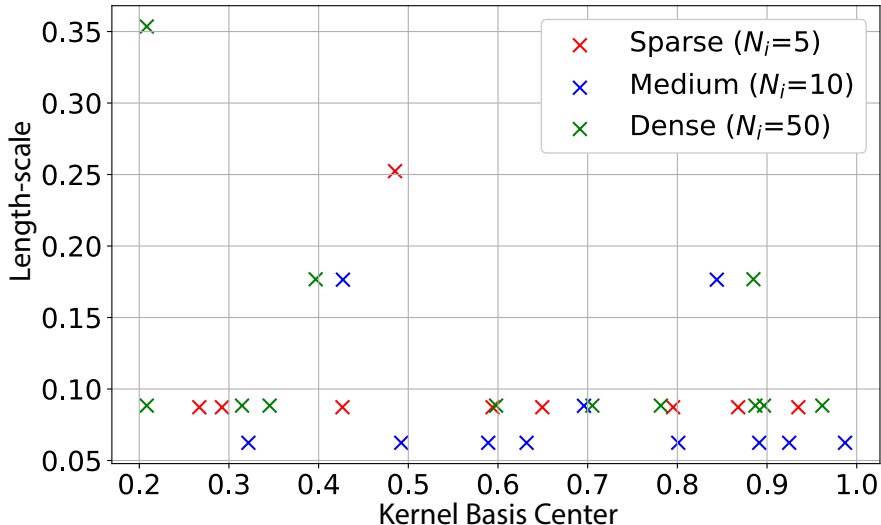

**Figure 3.** Length scales and centers of selected kernel basis functions in a random repetition for three different *m* values in Scenario 5.

Tables 1–5 show the results. The results for the first five methods are from [21]. Of 15 cases, the proposed *BSFDA* exhibits the highest accuracy in 12. In the other three cases, the accuracy of *BSFDA* is comparable to the best result and is always above 0.950. BSFDA^Fast demonstrates performance comparable to that of *BSFDA* when applied to medium-density and dense datasets with significantly higher efficiency, which we detail in Figure 4. However, its efficacy diminishes with sparse data. This limitation arises because the parameter $\varsigma$ can bias model estimation in scenarios with insufficient data evidence, leading to an underestimation of the signal variance. Consequently, BSFDA^Fast tends to underestimate the number of components, particularly those capturing nuanced variations, in the presence of sparse observations. Nonetheless, with adequate data, BSFDA^Fast achieves performance on par with that of the original model.

**Table 1.** Proportion of accurate estimations for Scenario 1 (r = 3).

| $N_i$ | AIC$_{PACE}$ | AIC | BIC | PC$_{p1}$ | IC$_{p1}$ | AIC$_{PACE}^{2022}$ | BIC$_{PACE}^{2022}$ | fpca | BSFDA | BSFDA$^{Fast}$ |
|---|---|---|---|---|---|---|---|---|---|---|
| 5 | 0.000 | 0.580 | 0.380 | 0.410 | 0.735 | 0.650 | 0.880 | 0.645 | **0.995** | 0.015 |
| 10 | 0.000 | 0.980 | 0.670 | 0.955 | 0.985 | 0.880 | 0.920 | 0.645 | **1.000** | 0.910 |
| 50 | 0.000 | **1.000** | 0.830 | **1.000** | **1.000** | **1.000** | **1.000** | 0.890 | 0.980 | 0.945 |

**Table 2.** Proportion of accurate estimations for Scenario 2 (r = 3).

| $N_i$ | AIC$_{PACE}$ | AIC | BIC | PC$_{p1}$ | IC$_{p1}$ | AIC$_{PACE}^{2022}$ | BIC$_{PACE}^{2022}$ | fpca | BSFDA | BSFDA$^{Fast}$ |
|---|---|---|---|---|---|---|---|---|---|---|
| 5 | 0.005 | 0.630 | 0.245 | 0.375 | 0.605 | 0.570 | 0.620 | 0.475 | **1.000** | 0.040 |
| 10 | 0.000 | 0.710 | 0.665 | 0.570 | 0.805 | 0.825 | 0.850 | 0.640 | **1.000** | 0.995 |
| 50 | 0.000 | 0.630 | 0.795 | 0.955 | 0.945 | **1.000** | **1.000** | 0.950 | **1.000** | 0.950 |

**Table 3.** Proportion of accurate estimations for Scenario 3 (r = 3).

| $N_i$ | AIC$_{PACE}$ | AIC | BIC | PC$_{p1}$ | IC$_{p1}$ | AIC$_{PACE}^{2022}$ | BIC$_{PACE}^{2022}$ | fpca | BSFDA | BSFDA$^{Fast}$ |
|---|---|---|---|---|---|---|---|---|---|---|
| 5 | 0.005 | 0.720 | 0.325 | 0.640 | 0.590 | 0.320 | 0.400 | 0.450 | **0.995** | 0.945 |
| 10 | 0.000 | 0.580 | 0.770 | 0.965 | 0.665 | 0.740 | 0.755 | 0.440 | 0.995 | **1.000** |
| 50 | 0.000 | 1.000 | 0.775 | 1.000 | 1.000 | **1.000** | **1.000** | 0.765 | 0.980 | 0.920 |

**Table 4.** Proportion of accurate estimations for Scenario 4 (r = 3).

| $N_i$ | $\text{AIC}_{\text{PACE}}$ | AIC | BIC | $\text{PC}_{p1}$ | $\text{IC}_{p1}$ | $\text{AIC}_{\text{PACE}}^{2022}$ | $\text{BIC}_{\text{PACE}}^{2022}$ | fpca | BSFDA | $\text{BSFDA}^{\text{Fast}}$ |
|---|---|---|---|---|---|---|---|---|---|---|
| 5 | 0.015 | 0.710 | 0.410 | 0.640 | 0.560 | 0.515 | 0.575 | 0.370 | **1.000** | 0.975 |
| 10 | 0.000 | 0.830 | 0.775 | 0.920 | 0.900 | 0.750 | 0.760 | 0.350 | **0.995** | 0.990 |
| 50 | 0.000 | 0.945 | 0.835 | **1.000** | **1.000** | **1.000** | **1.000** | 0.730 | 0.950 | 0.935 |

**Table 5.** Proportion of accurate estimations for Scenario 5 (r = 6).

| $N_i$ | $\text{AIC}_{\text{PACE}}$ | AIC | BIC | $\text{PC}_{p1}$ | $\text{IC}_{p1}$ | $\text{AIC}_{\text{PACE}}^{2022}$ | $\text{BIC}_{\text{PACE}}^{2022}$ | fpca | BSFDA | $\text{BSFDA}^{\text{Fast}}$ |
|---|---|---|---|---|---|---|---|---|---|---|
| 5 | 0.705 | 0.470 | 0.090 | 0.070 | 0.545 | 0.425 | 0.410 | 0.855 | **0.925** | 0.160 |
| 10 | 0.065 | 0.570 | 0.525 | 0.775 | 0.705 | 0.575 | 0.575 | 0.500 | **1.000** | 0.930 |
| 50 | 0.000 | 0.260 | 0.590 | 0.980 | 0.965 | 0.870 | 0.770 | 0.695 | **0.995** | 0.925 |

### 5.1.1. Mean Squared Error in Covariance Operator

The mean squared error across $X_{\text{grid}}$, a grid of 1000 index points,

$$\frac{||\operatorname{cov}(X_{\text{grid}}, X_{\text{grid}}) - \hat{\operatorname{cov}}(X_{\text{grid}}, X_{\text{grid}})||_F^2}{1000 \times 1000},$$

where $||\cdot||_F$ is the Frobenius norm, measures the accuracy of the estimated covariance. The quadratic measure of the error with the Frobenius norm for covariance estimators has been used by [41]. The methods compared include *fpca* of [13], *pace* of [11] with the AIC and BIC, and *refund-sc* of [12]. Only cases in Scenario 5 are used because of the time constraints (e.g., refund-sc takes 6 h for 20 repetitions with 50 points in Scenario 5). As the most challenging, Scenario 5 should provide the most compelling comparison. The results in Table 6 demonstrate that the proposed method is comparable to the best work in terms of the estimated covariance accuracy. Specifically, dense sampling becomes prohibitive for *refund-sc*. The results highlight the benefit of continuous formulations, as seen in both *fpca* and the proposed method, over the grid-based optimization in conventional methods. $\text{BSFDA}^{\text{Fast}}$ again performs comparably well when the data are adequate.

### 5.1.2. Multidimensional Functional Data Simulation

A simulation experiment with a 4D index set reveals the proposed method's advantages for high-dimensional data, where the gridding strategies of previous methods are impractical. The settings are as follows, with a length scale $l_{\text{s}} = 0.33$:

$$Z_i \sim \mathcal{N}(0, I) \in R^{1 \times 3} \tag{40}$$

$$\phi_0(x) = (\pi l_{\text{s}}^2)^{-2} \exp\left(-\frac{1}{2}\left\|\frac{x - [0.5, 0.5, 0.5, 0.5]}{l_{\text{s}}}\right\|_2^2\right) \tag{41}$$

$$\phi_1(x) = (\pi l_{\text{s}}^2)^{-2} \exp\left(-\frac{1}{2}\left\|\frac{x - [0.4, 0.4, 0.4, 0.4]}{l_{\text{s}}}\right\|_2^2\right) \tag{42}$$

$$\phi_2(x) = (\pi l_{\text{s}}^2)^{-2} \exp\left(-\frac{1}{2}\left\|\frac{x - [0.6, 0.6, 0.6, 0.6]}{l_{\text{s}}}\right\|_2^2\right) \tag{43}$$

$$y_i(x) = Z_{i0} * \sqrt{0.6} * (\phi_0(x) - \phi_1(x)) + Z_{i1} * \sqrt{0.3} * \phi_1(x) + Z_{i2} * \sqrt{0.4} * \phi_2(x) \tag{44}$$

**Table 6.** Mean squared error of covariance $\text{Error}_{\text{CovFunc}}$ for Scenario 5 .

| $N_i$ | $\text{AIC}_{\text{PACE}}^{2022}$ | $\text{BIC}_{\text{PACE}}^{2022}$ | fpca | refund.sc | BSFDA | $\text{BSFDA}^{\text{Fast}}$ |
|---|---|---|---|---|---|---|
| 5 | 12.373 ± 4.026 | 12.377 ± 4.031 | **5.192 ± 6.166** | 8.833 ± 4.730 | 5.814 ± 3.535 | 10.292 ± 12.717 |
| 10 | 10.391 ± 2.521 | 10.391 ± 2.521 | 2.098 ± 1.425 | 5.314 ± 3.501 | **2.068 ± 1.427** | 2.656 ± 1.712 |
| 50 | 9.054 ± 1.683 | 9.054 ± 1.683 | 1.642 ± 1.240 | N/A | **1.638 ± 1.247** | 1.770 ± 1.275 |

The observations include additive noise with a sigma of $4.472 \times 10^{-1}$. The cross-validation selects a length scale of 0.405. The estimated noise sigma is $4.637 \times 10^{-1}$. The proposed method correctly estimates the number of principal components as three and selects 31 basis functions. As shown in Figure 5, the eigenfunctions are correctly estimated. In addition, the estimated mean function is zero, which is accurate.

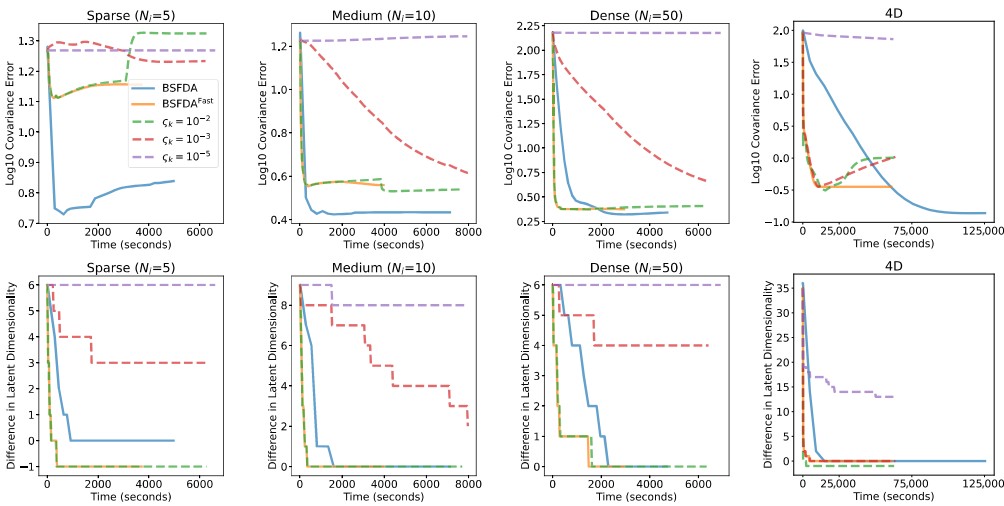

**Figure 4.** Convergence plots for Scenario 5 in Yehua and the 4D simulation. The upper row displays the covariance error against time, and the lower row illustrates the difference between the estimated and true numbers of components.

Next, we present a convergence comparison between BSFDA and $\text{BSFDA}^{\text{Fast}}$ under four schedules for the coefficient noise $\varsigma_k$. Specifically, we compare the default diminishing schedule from $10^{-2}$ to $10^{-5}$ with three fixed settings: $10^{-2}$, $10^{-3}$, and $10^{-5}$. We evaluate the covariance error and the discrepancy between the estimated/true dimensionality in one replicate of each sample density in Scenario 5 and the 4D simulation. For the 4D case, we adopt a default initial $\varsigma_k$ of $10^{-3}$. As illustrated in Figure 4, $\text{BSFDA}^{\text{Fast}}$ achieves comparable accuracy to BSFDA while converging significantly faster than BSFDA in terms of both covariance errors and component estimation for medium and densely sampled data. In the 4D case, $\text{BSFDA}^{\text{Fast}}$ converges in covariance estimation after approximately 10,000 s and in dimensionality after around 4000 s, compared to roughly 100,000 s and 13,000 s, respectively, for BSFDA. However, for sparse data, $\text{BSFDA}^{\text{Fast}}$ exhibits reduced estimation accuracy and underestimates the number of components by one. A similar decline in accuracy is observed in the 4D simulation when data sparsity is high. This limitation arises because the introduction of coefficient noise $\varsigma$ biases the model toward eliminating signals that are deemed insignificant. Moreover, when comparing the three fixed-$\varsigma_k$ variants of the fast algorithm, a clear trade-off emerges: a smaller $\varsigma_k$ reduces the overall error but slows down the optimization due to increased dependency among variables. These results collectively demonstrate the effectiveness of our chosen $\varsigma_k$ schedule in $\text{BSFDA}^{\text{Fast}}$, as it balances both efficiency and accuracy.

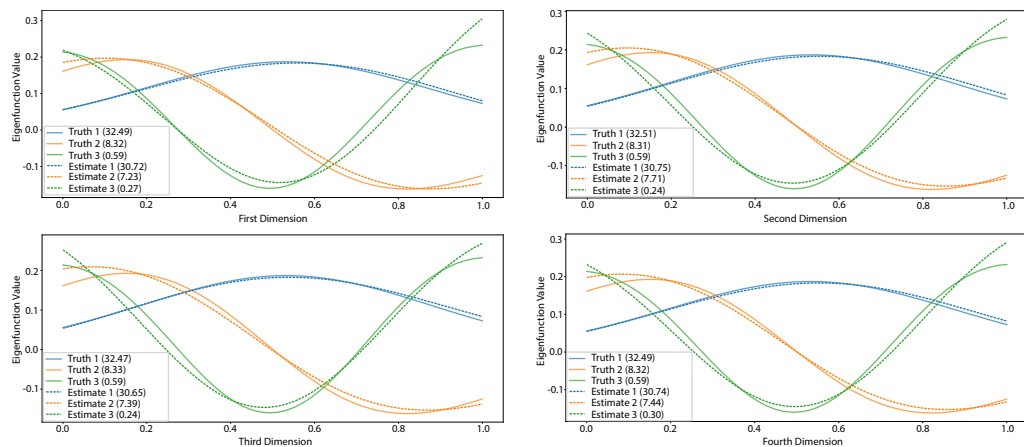

**Figure 5.** Cross-sectional visualization of eigenfunctions (eigenvalues) of the 4D simulation.

*5.2. Results on Public Datasets*

The proposed method's practicality was validated with two application datasets, CD4 and wind speed measurements.

### 5.2.1. CD4

CD4 data, a classical form of functional data, have received attention in [1,11,13]. CD4 cell counts gauge the immune system's response to human immunodeficiency virus (HIV) infection, which leads to a progressive reduction in CD4 cell counts. The Multicenter AIDS Cohort Study (MACS) [42] provided the CD4 data. This dataset consists of CD4 percentages from 283 male human subjects that were HIV-positive, each with 1 to 14 repeated measurements over time in years. Subjects were scheduled for reevaluation at least semiannually. However, missed visits caused the sparse and uneven distribution of measurements. The proposed method used five length scales selected from cross-validation and k-means clustering. Finally, the model selected nine basis functions. Figure 6 displays the estimated mean function, eigenfunctions, and curves of the observations. The mean function reflects the overall decreasing tendency with the progression of the disease. The eigenfunctions are obtained by applying the singular value decomposition of the covariance operator that is discretized (for visualization purposes only) with a grid of 50 evenly spaced points over the whole timeline. The first eigenfunction is relatively flat and mainly captures the subject-specific average magnitude of the CD4 counts, consistent with the findings of [1,11,13]. The second eigenfunction captures the simple linear trend of the variations, as described in [13]. The third eigenfunction captures the piece-wise linear time trend with a breakpoint near 2.5 years from the baseline. Refs. [1,11] found similar eigenfunctions.

### 5.2.2. Wind Speed

The wind speed data, collected from 110 locations across Utah's Salt Lake Valley, range between 11 and 1440 measurements. The proposed method leverages 10 length scales selected from cross-validation and k-means clustering. Figure 7 illustrates the estimated mean function, curves of the observations, eigenfunctions, and covariance. The horizontal axis represents the seconds starting from 12:00 AM Greenwich Mean Time (GMT) on 15 June 2023, which corresponds to 6:00 PM in Salt Lake City. In Figure 7a, the estimated mean function depicts two pronounced peaks observed approximately at 8:00 PM and 6:00 AM, as well as two troughs around 12:00 AM and 12:00 PM. This pattern aligns with the diurnal cycle, particularly highlighting the thermal activity associated with sunset and sunrise. The peaks during sunset and sunrise are due to the interplay of topographical features, which result in specific breezes, such as the land breeze near the Great Salt Lake and the

distinct mountain and valley breezes. The troughs, on the other hand, reflect moments when the atmosphere is at its most stable, with minimal thermal activity disrupting wind patterns. The complexity of the data is distilled and represented using 12 descriptors with 17 basis functions. As Figure 7b shows, the primary eigenfunction is relatively level, indicating that the most significant variation is the location-specific average magnitude. Its profile echoes the influence of sunrise and sunset observed in the mean function, with elevations around 7:00 PM and 5:00 AM and subdued patterns during other times, indicative of similar atmospheric stability. The estimated covariance in Figure 7c highlights variance peaks around 8 PM and 5 AM, as well as a strong correlation between these periods. This underscores the effects of location-specific topographic factors on the wind speed.

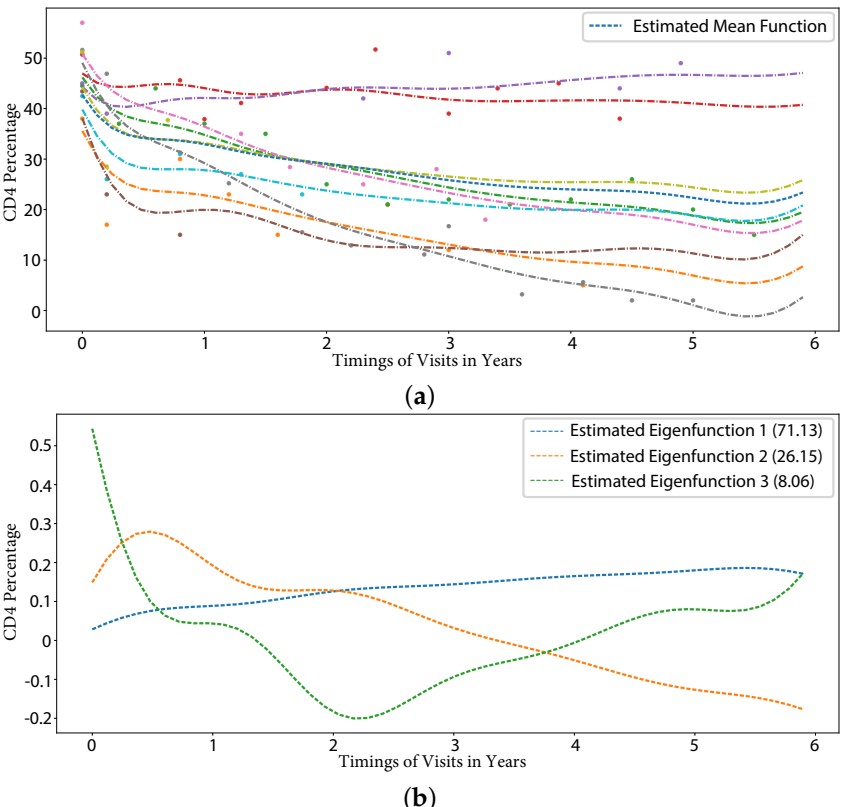

**Figure 6.** Outcomes from the proposed method applied to MACS CD4 datasets. (**a**) Estimated curves for a random selection of nine sampled functions and the mean function. (**b**) Estimated eigenfunctions (eigenvalues).

### 5.2.3. Modeling Large-Scale, Dynamic, Geospatial Data

Here, we demonstrate the scalability regarding both the size of the measurements and the dimensionality of our framework. For this, we apply it to the ARGO dataset, which consists of ocean temperature measurements from more than 4000 locations, at multiple depths and time points [37]. ARGO is a nearly global observing system for the ocean temperature, salinity, and other key variables via autonomous profiling floats. As of 2019, ARGO has generated over 338 gigabytes of data from 15,231 floats [37]. We focused on *high-quality* ("research" mode option in the database API) data from 1998 to 2024 for depths between 0 and 200 m in the open-access snapshot of Argo GDAC of 9 November 2024 [43]. The number of measurement points per year varies widely—from 38,931 to over 11 million, with 127 million in total. Figure 8 illustrates a global map of the sea surface temperature measurements from February 2021, highlighting the dataset's extensive spatial coverage.

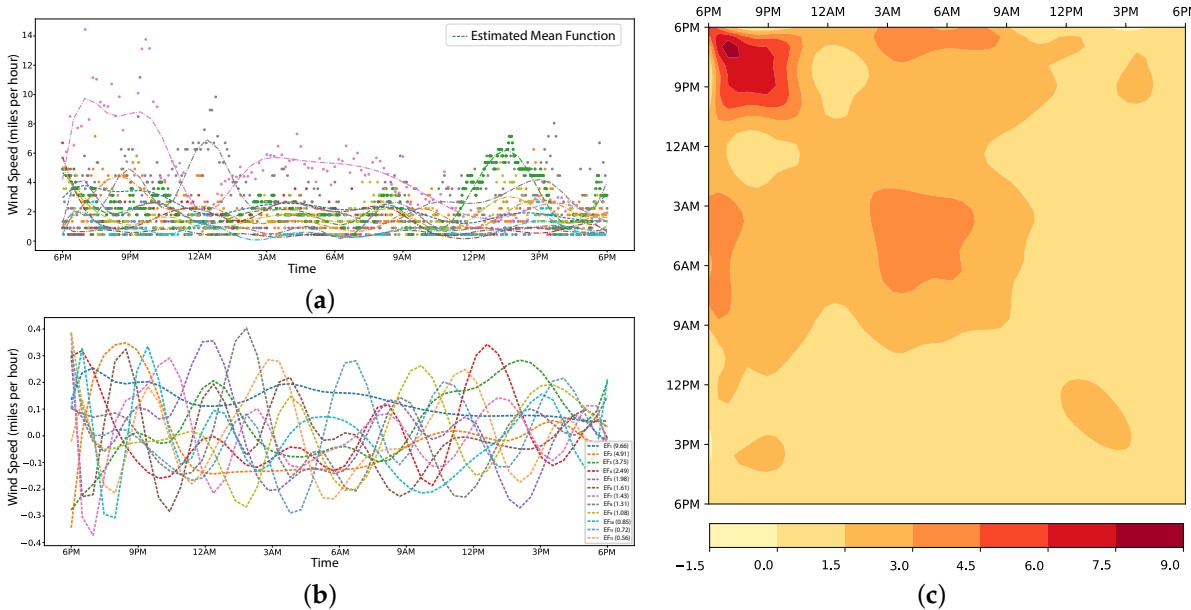

**Figure 7.** Outcomes from the proposed method applied to a wind speed dataset. (**a**) Estimated curves for a random selection of 9 sampled functions (denoted by different colors) and the mean function. (**b**) Estimated eigenfunctions (eigenvalues) denoted as EF. (**c**) Estimated covariance.

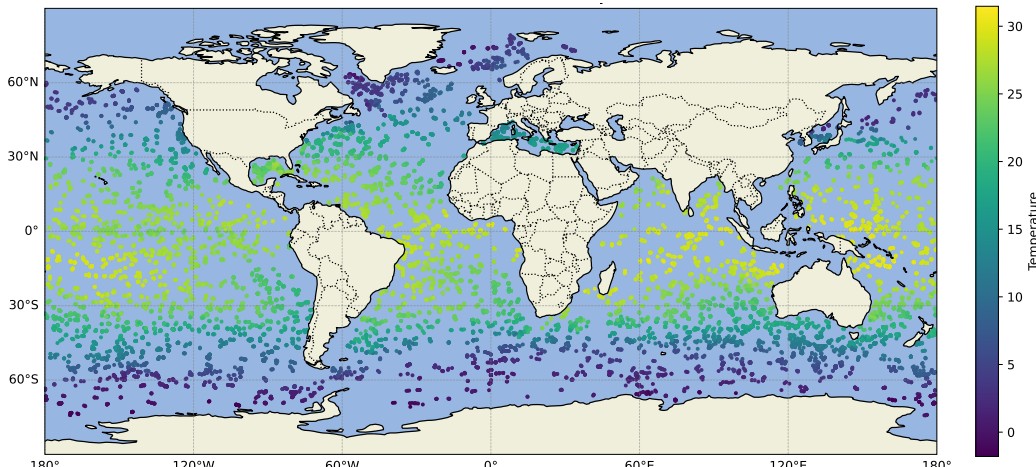

**Figure 8.** Temperature measurements in February 2021 near the sea surface in the ARGO dataset.

In our modeling, each year's data are treated as a single underlying function of four variables: latitude, longitude (on the spherical Earth), depth, and intra-annual time (modeled as a periodic variable). Note that the spatial data lie on a sphere and the time is a circle, assuming the periodicity of the time of the year. Our approach models these measurements holistically—without resorting to moving windows or submodeling—thereby preserving the continuous nature of the data and enabling the extraction of meaningful global, seasonal, and depth-dependent trends. Furthermore, the unique geospatial and temporal structure of the ARGO data, with spatial coordinates on a sphere and time exhibiting periodicity, necessitates specialized modeling techniques. Given that our model is 4D, the 4D kernel is defined as a product of the following kernels, following the design strategy for climatological data in [10]. The geospatial kernel on the sphere is a radial basis function (RBF) on geodesic distances. To ensure periodicity, the temporal kernel is an Exp-Sine-Squared $k(x, x') = \exp\left(-\frac{2\sin^2(\pi|x-x'|)}{l_{\mathrm{s}}}\right)$, where $l_{\mathrm{s}}$ is the length scale. For depth, we use a Gaussian kernel.

The numeric data (excluding metadata) as input to the model were approximately 4 GB. For length scale selection, we used Gaussian process regression on a small subset of

2000 randomly selected data in 2016 (medium size of measurements) for a cross-validated RMSE, which we optimized with a grid search. The specific length scales were set as follows: geodesic length scale of $2 \times 10^3$ km, depth length scale of 70 m, time length scale of 3, and periodicity of 1. For evaluation, we held out 10% of the depth profiles (a single round trip of a buoy from the surface to a depth at the same coordinate) from each year as testing data, following [44]. The total training set contained roughly 114 million points. Because the sample spacing was typically small relative to the selected length scales, we applied agglomerative clustering to 10,000 randomly chosen index points, reducing them to 2000 candidate basis functions. These candidate basis functions—precomputed for efficiency—took roughly 1.7 TB of memory. Computations were performed with 24 threads on a server equipped with 192 Intel® Xeon® Platinum 8360H CPUs @ 3.00 GHz and 3 TB RAM. Initialization was conducted using the modified RVM for 200 iterations for initial basis functions, using stochastic optimization with a 1000 batch size per year. Then, BSFDA$^{Fast}$ was executed for 10,000 iterations, where the heuristic to include new bases also used a 1000 batch size per year. With these computational strategies and heuristics, the entire modeling process was completed in 15 .

The proposed approach selected 163 effective basis functions and condensed them into 16 principle components. The final model occupied merely 50 MB of storage. The interpolation yielded a root mean square error (RMSE) of 1.95 and an $R^2$ of 94.2% on the testing data, reflecting a reasonable balance between global dimension reduction and fidelity. The estimated white noise level was also 1.95, indicating that the training data adequately covered the underlying variability in the ARGO observations, and the final model was reasonably generalizable.

Figure 9 presents 2D visualizations of geospatial interpolations at three depths (in decibars, roughly meters) and a specific time (29 May 2021) around 1° S and 30° W, each with three views. We have chosen one measurement as the central point, denoted by the red circle, and selected a narrow window (±1 decibar, ±1 day) around this center. The cyan and fuchsia circles represent training and testing data, respectively, within this window. Their sizes indicate the distance along the unplotted dimensions (depth and time here), reflecting variations in these dimensions. The visualizations show that the temperatures are warmer near the equator and decrease with depth. The match between the interpolated values and actual measurements demonstrates consistency in capturing broad spatial and vertical variations.

Figure 10 complements this by illustrating interpolation in the depth–time slices while holding the geospatial coordinates fixed, focusing on mixed layer characteristics. The "mixed layer" refers to a region of nearly uniform temperature, which is crucial in understanding thermodynamic potential and nutrient cycling [45]. Here, the plot uses a window of 50 km to include actual measurements, and the circle sizes denote the geodesic distance from the chosen center.

We plot every fifth measurement vertically to reduce overlap and improve the clarity. Figure 10a uses the same center point, 1° S and 30° W, as in Figure 9, exhibiting a shallow mixed layer with pronounced vertical gradients. In contrast, Figure 10b adopts a center at a higher latitude, 49° N and 29° W, where the model reveals a deeper mixed layer. The temperature here remains relatively stable below the surface. The dominant variations are cyclic seasonal changes, which are warmer near the surface around September. As is shown, the vertical sequence of the center and the nearby testing sequence match the interpolation closely. These results confirm that the mid-latitudes exhibit a stronger seasonal cycle [45] and that BSFDA$^{Fast}$ accurately approximates the actual measurements.

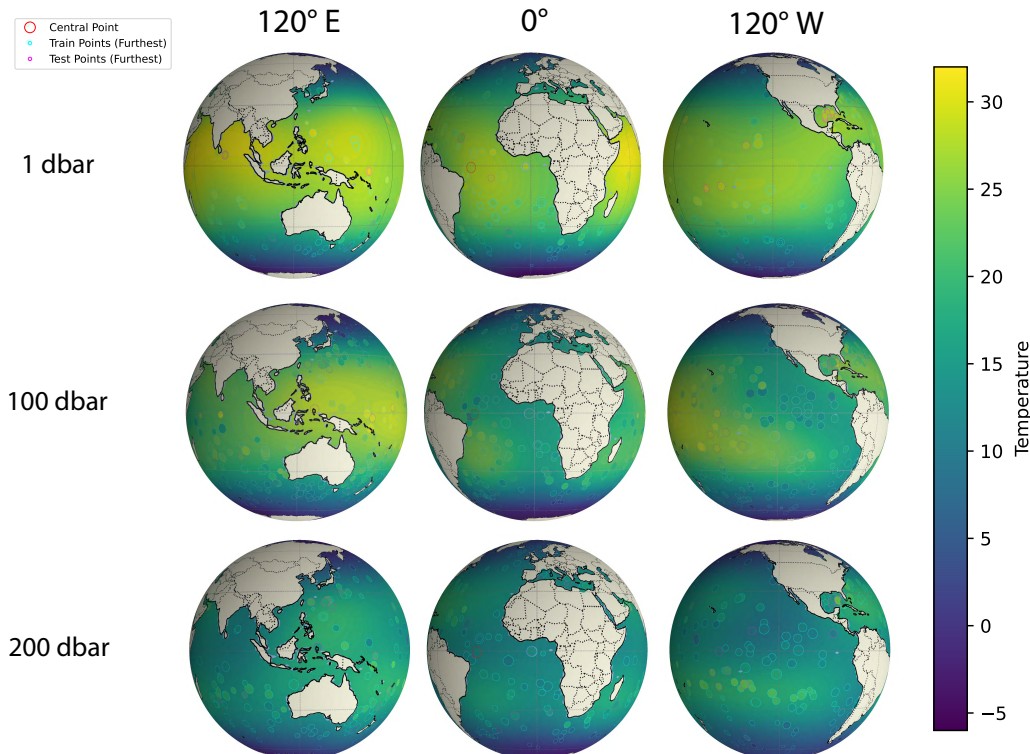

**Figure 9.** Geodesic interpolation from BSFDA$^{\text{Fast}}$ vs. actual ARGO global oceanic measurements at 1, 200, and 300 decibars, at 1° S and 30° W, on May 29. Measurements are represented by circles, with the filling color indicating the temperature. Circle sizes show distance in depth and time from the central point.

To our knowledge, this is the first time that the ARGO dataset has been modeled in a full 4D principal component model, with the correct domain topology. We incorporate the entire period of 27 years, rather than shorter spans (e.g., 2004–2008 or 2007–2016) [44,46,47]. Instead of segmenting the dataset into localized spatiotemporal windows, we process the entire 4D domain (latitude, longitude, depth, and intra-annual time) in a single holistic framework. Previous studies were typically tailored to ARGO datasets and handled each depth, month, or spatial region separately, restricting the correlation estimates to limited windows (e.g., 1000 km and three months) while excluding data with large offsets [44,47]. In addition, they required repeated on-demand model fitting, which can hinder scalability. Our kernel-based framework, by contrast, is broadly applicable to general functional data, only requiring kernel definitions for the domain. Although global dimension reduction inevitably introduces some residual noise, the kernel-based design is extensible to finer spacing or multiple length scales if higher precision is needed. Furthermore, inference with our model is simply the evaluation of the active 163 active basis functions weighted by the 16 principal components. Interpolation over a $300 \times 300$ grid only takes about two seconds. By contrast, previous methods with Gaussian process regression require a weighted sum of all measured data within a certain window. The parametric representations also facilitate straightforward derivative and integration calculations, which are essential in investigating ocean temperature stratification and heat content [44]. In summary, the ARGO dataset provides an ideal testbed for our method, as it captures the dynamic behavior of high-dimensional geospatial data in a continuous framework. A more comprehensive study of ARGO was beyond this study's scope. Nonetheless, the results here confirm the clear advantages of the proposed method for large-scale, high-dimensional functional data.

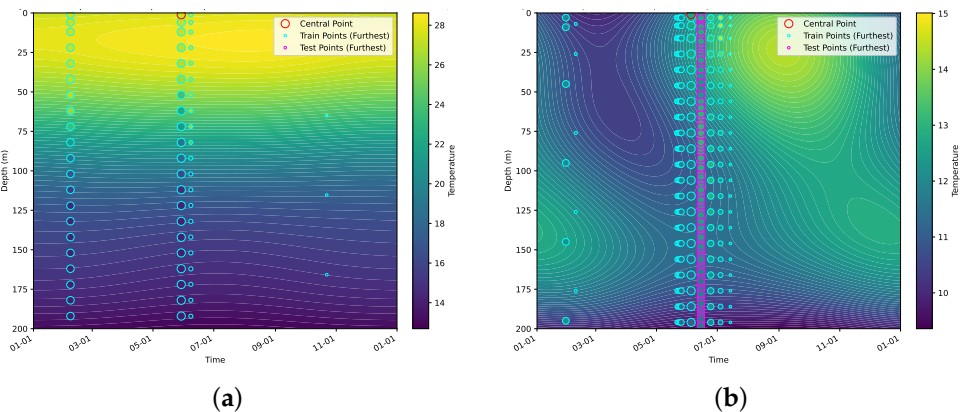

**Figure 10.** Depth–time interpolation from BSFDA$^{\text{Fast}}$ vs. actual ARGO global oceanic measurements at two sites focusing on mixed layer behavior. Measurements are represented by circles (green for training and pink for testing data), with the filling color indicating the temperature. Circle sizes show distance in geodesic space from the central point (denoted as red circles). (**a**) Shallow mixed layers at 1° S and 30° W. (**b**) Deep mixed layers at 49° N and 29° W.

## 6. Discussion and Conclusions

This paper proposes BSFDA, a novel framework for functional data analysis with irregular sampling, integrating model selection and scalability in one unique, coherent, and effective algorithm. Our extensive empirical studies, including both simulations and real-world applications, show that BSFDA offers superior covariance estimation accuracy with remarkable efficiency.

In terms of accuracy, our method excels in model selection, consistently achieving top-tier performance. The accuracy of the covariance operator estimation also rivals that of the best existing methodologies in the field. This shows that our approach can not only handle large and complex datasets but also ensure high accuracy and precision in the results that it produces. Our method's superiority compared to existing techniques is expected owing to the inherent iterative nature of data smoothing and covariance estimation in our approach.

In terms of scalability, our method demonstrates linear growth in time complexity with the size of the dataset, and, impressively, the computations are executed in a small, $K^{(a)}$-dimensional subspace. This property ensures that, as the datasets grow larger and more complex, the performance of our model remains robust and efficient. Additionally, we introduce a faster variant, BSFDA$^{\text{Fast}}$, which performs similarly to BSFDA on medium and dense datasets with a significantly reduced computational cost. This leap in efficiency enabled the full 4D functional modeling, for the first time, of a large-scale oceanic temperature dataset across 27 years (ARGO) [37]. Although BSFDA$^{\text{Fast}}$ can underestimate the signal strength under very sparse sampling, the vanilla BSFDA effectively complements and alleviates this issue.

Although the proposed framework proves effective in various real-world scenarios, it relies on the specific scheduling of coefficient noise levels. This schedule transitions from a faster, more biased model to a slower, less biased one, balancing the convergence speed against the estimation accuracy. Whereas empirical tests validate its advantages, there remain exploratory directions for further enhancement. For instance, the proper incorporation of structured variational inference or injection of artificial, compensatory noise in the observations could enable fast inference with reduced bias at the same time. However, it is necessary to address the increased complexity in optimization and variance in the estimation. Additionally, variational inference based on mean field approximations may underestimate the posterior variance, which is acceptable in many tasks [38] but leaves an open question when the independence assumptions are severely violated. Moreover, variational inference prioritizes computational efficiency over strict theoretical optimal-

ity [23,38], and the inherent infinite-dimensionality of fPCA further complicates any formal asymptotic analysis, even though our empirical studies demonstrate decreasing errors as the sampling density increases.

Looking ahead, it would be interesting to explore how extensions of regular PCA, such as simplified PCA and robust PCA [25], can be integrated within our proposed framework. Domains such as finance, in particular, include large-time-series datasets that often contain more outliers. These extensions will enhance the flexibility and robustness of our method, further improving its adaptability to various data conditions. In addition, we see potential in examining the extensions of functional PCA, such as time warping, dynamics, and manifold learning [1]. In particular, shape analysis emerges as a direct application of time warping. Such extensions would push the boundaries of what our proposed method could achieve, potentially enabling it to handle an even wider array of data structures and complexities.

In conclusion, our research findings affirm the proposed framework's effectiveness and adaptability in advanced functional data analysis. Nonetheless, the method's potential remains broad, and future work promises to widen its scope and refine its performance. By unifying sparse Bayesian learning, kernel-based expansions, and efficient variational inference, BSFDA offers a powerful foundation for large-scale, high-dimensional FDA challenges.

**Author Contributions:** Conceptualization, S.J. and R.W.; Methodology, W.T., S.J. and R.W.; Software, W.T.; Investigation, S.J. and R.W.; Writing—original draft, W.T.; Writing—review & editing, S.J. and R.W.; Visualization, W.T.; Supervision, S.J. and R.W. All authors have read and agreed to the published version of the manuscript.

**Funding:** This research was funded by National Institutes of Health: 5R01DE032366-02A1.

**Data Availability Statement:** The CD4 data are openly available in https://rdrr.io/cran/timereg/man/cd4.html (accessed on 17 April 2025). The wind speed data are private. The ARGO data are openly available in https://www.seanoe.org/data/00311/42182/ (accessed on 17 April 2025).

**Conflicts of Interest:** The authors declare no conflicts of interest.

## Appendix A. System of Notation

Table A1 summarizes the notation used in Sections 2 and 4, providing a reference for the derivations. All vectors in the table are represented as row vectors.

**Table A1.** Symbol definitions in formulation.

| Symbol | Meaning |
| --- | --- |
| $y_i$ | $i$-th sample function |
| $x \in R^M$ | One $M$-dimension index |
| $M$ | Dimension of the index set |
| $K$ | Number of all basis functions |
| $J$ | Number of all components |
| $P$ | Number of sample functions |
| $N_i$ | Number of measurements of the $i$-th sample function |
| $X_i \in R^{Ni \times M}$ | Index set of the $i$-th sample function |
| $Y_i \in R^{Ni}$ | Measurement of the $i$-th sample function |
| $Z_i \in R^J$ | Component scores of the $i$-th sample function |
| $\bar{Z} \in R^K$ | Coefficients of basis functions in the mean function |
| $E_i \in R^{Ni}$ | Measurement errors of the $i$-th sample function |
| $W \in R^{J \times K}$ | Weighing matrix of basis functions in the eigenfunctions |
| $W_{j\cdot} \in R^K, W_{\cdot k}^T \in R^J$ | $j$-th row and $k$-th column of $W$ |
| $\mathcal{K}$ | Kernel function |
| $\alpha_j$ | Scale parameter of $W_{j\cdot}$ ($j$-th component) |
| $\beta_k$ | Scale parameter of $W_{\cdot k}$ ($k$-th basis function) |

**Table A1.** *Cont.*

| Symbol | Meaning |
|---|---|
| $\sigma$ | The standard deviation of measurement errors |
| $\eta$ | The communal scale parameter of $\bar{Z}$ |
| $\{\phi_k : R^M \to R\}_{k=1}^K$ | The union of all the centered kernel functions |
| $\Phi_{ikj} = \phi_k(X_{ij\cdot}) \in R$ | The value of centered kernel function $\phi_k$ at $X_{ij\cdot}$ |
| $\theta_i \in R^K$ | The coefficients of the $i$-th sample function |
| $\zeta_i \in R^K$ | The coefficient noise of the $i$-th sample function |
| $\varsigma_k$ | The scale parameter of the $k$-th coefficient noise |

Table A2 summarizes the notation used in Section 3.

**Table A2.** Notation used in formulating the optimization.

| Symbol | Meaning |
|---|---|
| $\Theta$ | All latent variables |
| $\mathcal{Q}_\cdot$ | The surrogate posterior distribution of variable $\cdot$ |
| $\mathcal{Q}_{/\cdot}$ | The joint surrogate posterior distribution of all variables except $\cdot$ |
| $\mu_\cdot, \Sigma_\cdot$ | The mean and covariance of $\cdot$ in $\mathcal{Q}$, e.g., $\mu_{\text{vec}(W)} \in R^{JK}$, $\Sigma_{\text{vec}(W)} \in R^{JK \times JK}$ |
| $a_\cdot, b_\cdot$ | The shape and rate parameters of $\mathcal{Q}_\cdot$, e.g., $a_{\beta_k}, b_{\beta_k}$ |
| $\mathbb{E}_\mathcal{Q}[\cdot]$ | The expectation of variable $\cdot$ over density $\mathcal{Q}$ |
| $\mathcal{L}$ | The lower bound of surrogate posterior $\mathcal{Q}$ with $K$ basis functions |
| $\Psi_i$ | The Gram matrix of the kernel functions for the $i$-th sample function, $\Phi_i \Phi_i^T$ |
| $K^{(a)}, K^{(e)}$ | The number of active/effective basis functions |
| $J^{(a)}, J^{(e)}$ | The number of active/effective components |
| $\mathcal{P}_i$ | The log likelihood of $Y_i$ in a multisample relevance vector machine |
| $\mathcal{C}_i$ | The covariance of $Y_i$ in a multisample relevance vector machine |
| $\mathcal{S}_i$ | The posterior covariance of $Z_i$ in a multisample relevance vector machine |
| $\mathcal{P}_{Z_i}$ | The log likelihood of $(Y_i, Z_i)$ in a multisample relevance vector machine |
| $\epsilon \to 0$ | The infinitesimal number |
| $\tau_\cdot$ | The threshold/tolerance of $\cdot$ |

## Appendix B. Variational Update Formulae

As defined in Section 2, we consider the following priors and conditional distributions:

$$\Pr[Y|Z, W, \bar{Z}, \sigma] = \prod_i \mathcal{N}\left(Y_i | (Z_i W + \bar{Z})\Phi_i, \sigma^2 I\right) \tag{A1}$$

$$\Pr[Z] = \prod_i \mathcal{N}(Z_i|0, I) \tag{A2}$$

$$\Pr[W|\alpha, \beta] = \prod_{j,k} \mathcal{N}(W_{jk}|0, \alpha_j^{-1}\beta_k^{-1}) \tag{A3}$$

$$\Pr[\bar{Z}] = \prod_k \mathcal{N}(\bar{Z}_k|0, \eta^{-1}\beta_k^{-1}) \tag{A4}$$

$$\Pr[\sigma]\Pr[\alpha]\Pr[\beta]\Pr[\eta] = \Gamma(\sigma^{-2}|a_0, b_0)\prod_{j=1}^J \Gamma(\alpha_j|a_0, b_0)\prod_{k=1}^K \Gamma(\beta_k|a_0, b_0)\Gamma(\eta|a_0, b_0) \tag{A5}$$

For brevity, the joint posterior is shown with the vague Gamma prior parameters $a_0$, $b_0$, and the observation index $X$ omitted:

$$
\begin{aligned}
\Pr[Z, W, \bar{Z}, \sigma, \alpha, \beta, \eta | X, Y, a_0, b_0] &= \Pr[Z, W, \bar{Z}, \sigma, \alpha, \beta, \eta | Y] \\
&= \Pr[Z, W, \bar{Z}, \sigma, \alpha, \beta, \eta, Y](\Pr[Y])^{-1} \\
&\propto \Pr[Z, W, \bar{Z}, \sigma, \alpha, \beta, \eta, Y] \\
&= \Pr[Y|Z, W, \bar{Z}, \sigma]\Pr[Z]\Pr[W|\alpha, \beta]\Pr[\bar{Z}|\eta, \beta]\Pr[\sigma]\Pr[\alpha]\Pr[\beta]\Pr[\eta]
\end{aligned}
\tag{A6}
$$

**Derivation of Equations** (14) **and** (15)

According to Equation (13) and the posterior in Equation (A6), the update formulae for the surrogate distribution $\mathcal{Q}_{\alpha_j}$ are

$$
\begin{aligned}
\mathcal{Q}_{\alpha_j} &\leftarrow \frac{\exp\left(\mathbb{E}_{\mathcal{Q}_{/\alpha_j}}\left[\ln\left(\Pr[Z, W, \bar{Z}, \sigma, \alpha, \beta, \eta, Y]\right)\right]\right)}{\int \exp\left(\mathbb{E}_{\mathcal{Q}_{/\alpha_j}}\left[\ln(\Pr[Z, W, \bar{Z}, \sigma, \alpha, \beta, \eta, Y])\right]\right) d\alpha_j} \\
&\propto \exp\left(\mathbb{E}_{\mathcal{Q}_{/\alpha_j}}\left[\ln\left(\Pr[Z, W, \bar{Z}, \sigma, \alpha, \beta, \eta, Y]\right)\right]\right) \propto \exp\left(\mathbb{E}_{\mathcal{Q}_{/\alpha_j}}\left[\ln\left(\Pr[W|\alpha, \beta]\Pr[\alpha]\right)\right]\right) \\
&\propto \exp\left(\mathbb{E}_{\mathcal{Q}_{/\alpha_j}}\left[-\frac{1}{2}\sum_{k=1}^{K}\left(-\ln(\alpha_j) + W_{jk}^2\alpha_j\beta_k\right) + \left((a_0 - 1)\ln\alpha_j - b_0\alpha_j\right)\right]\right) \\
&\propto \exp\left(\left(\frac{K}{2} + a_0 - 1\right)\ln(\alpha_j) - \alpha_j\left(\frac{1}{2}\sum_{k=1}^{K}\mathbb{E}_{\mathcal{Q}_{/\alpha_j}}\left[\left(W_{jk}^2\beta_k\right)\right] + b_0\right)\right)
\end{aligned}
\tag{A7}
$$

where we have omitted terms of which $\alpha_j$ is conditionally independent. By definition,

$$
\begin{aligned}
\mathcal{Q}_{\alpha_j} &= \exp\left(\ln(\Gamma(\alpha_j|a_{\alpha_j}, b_{\alpha_j}))\right) = \exp\left(\ln\left(\frac{b_{\alpha_j}^{a_{\alpha_j}}}{\Gamma(a_{\alpha_j})}\alpha_j^{a_{\alpha_j}-1}\exp\left(-b_{\alpha_j}\alpha_j\right)\right)\right) \\
&\propto \exp\left((a_{\alpha_j} - 1)\ln\alpha_j - b_{\alpha_j}\alpha_j\right)
\end{aligned}
\tag{A8}
$$

By equating Equations (A7) and (A8), the updates for $\mathcal{Q}_{\alpha_j}$ are

$$
a_{\alpha_j} \leftarrow \frac{K}{2} + a_0
\tag{A9}
$$

$$
b_{\alpha_j} \leftarrow \frac{1}{2}\sum_{k=1}^{K}\mathbb{E}_{\mathcal{Q}_{/\alpha_j}}\left[\left(W_{jk}^2\beta_k\right)\right] + b_0
\tag{A10}
$$

**Derivation of Equations** (16) **and** (17)

According to Equation (13) and the posterior Equation (A6), the update formulae for $\mathcal{Q}_\eta$ are

$$
\begin{aligned}
\mathcal{Q}_\eta &\leftarrow \frac{\exp\left(\mathbb{E}_{\mathcal{Q}_{/\eta}}\left[\ln\left(\Pr[Z, W, \bar{Z}, \sigma, \alpha, \beta, \eta, Y]\right)\right]\right)}{\int \exp\left(\mathbb{E}_{\mathcal{Q}_{/\eta}}\left[\ln(\Pr[Z, W, \bar{Z}, \sigma, \alpha, \beta, \eta, Y])\right]\right) d\eta} \\
&\propto \exp\left(\mathbb{E}_{\mathcal{Q}/\eta}\left[\ln\left(\Pr[Z, W, \bar{Z}, \sigma, \alpha, \beta, \eta, Y]\right)\right]\right) \propto \exp\left(\mathbb{E}_{\mathcal{Q}/\eta}\left[\ln\left(\Pr[\bar{Z}|\eta, \beta]\Pr[\eta]\right)\right]\right) \\
&\propto \exp\left(\mathbb{E}_{\mathcal{Q}/\eta}\left[-\frac{1}{2}\sum_{k=1}^{K}\left(-\ln(\eta) + \bar{Z}_k^2\eta\beta_k\right) + \left((a_0 - 1)\ln\eta - b_0\eta\right)\right]\right) \\
&\propto \exp\left(\left(\frac{K}{2} + a_0 - 1\right)\ln(\eta) - \eta\left(\frac{1}{2}\sum_{k=1}^{K}\mathbb{E}_{\mathcal{Q}/\eta}\left[\left(\bar{Z}_k^2\beta_k\right)\right] + b_0\right)\right)
\end{aligned}
\tag{A11}
$$

where we have omitted terms of which $\eta$ is conditionally independent. By definition,

$$
\begin{aligned}
\mathcal{Q}_\eta &= \exp\left(\ln(\Gamma(\eta|a_\eta, b_\eta))\right) = \exp\left(\ln\left(\frac{b_\eta^{a_\eta}}{\Gamma(a_\eta)}\eta^{a_\eta-1}\exp\left(-b_\eta\eta\right)\right)\right) \\
&\propto \exp\left((a_\eta - 1)\ln\eta - b_\eta\eta\right)
\end{aligned}
\tag{A12}
$$

By equating Equations (A11) and (A12), the updates for $\mathcal{Q}_\eta$ are

$$
a_\eta \leftarrow \frac{K}{2} + a_0
\tag{A13}
$$

$$
b_\eta \leftarrow \frac{1}{2}\sum_{k=1}^{K}\mathbb{E}_{\mathcal{Q}/\eta}\left[\left(\bar{Z}_k^2\beta_k\right)\right] + b_0
\tag{A14}
$$

**Derivation of Equations (18) and (19)**

According to Equation (13) and the posterior Equation (A6), the update formulae for $\mathcal{Q}_{\beta_k}$ are

$$
\mathcal{Q}_{\beta_k} \leftarrow \frac{\exp\left(\mathbb{E}_{\mathcal{Q}_{/\beta_k}}[\ln\left(\Pr[Z, W, \bar{Z}, \sigma, \alpha, \beta, \eta, Y])\right]\right)}{\int \exp\left(\mathbb{E}_{\mathcal{Q}_{/\beta_k}}[\ln(\Pr[Z, W, \bar{Z}, \sigma, \alpha, \beta, \eta, Y])]\right) d\beta_k}
$$

$$
\propto \exp\left(\mathbb{E}_{\mathcal{Q}_{/\beta_k}}[\ln\left(\Pr[Z, W, \bar{Z}, \sigma, \alpha, \beta, \eta, Y])\right]\right)
$$

$$
\propto \exp\left(\mathbb{E}_{\mathcal{Q}_{/\beta_k}}[\ln\left(\Pr[W_{\cdot k}|\alpha, \beta_k]\Pr[\bar{Z}_k|\eta, \beta_k]\Pr[\beta])\right]\right)
$$

$$
\propto \exp\left(\mathbb{E}_{\mathcal{Q}_{/\beta_k}}\left[-\frac{1}{2}\sum_{j=1}^{J}\left(-\ln(\alpha_j\beta_k) + W_{jk}^2\alpha_j\beta_k\right)\right.\right.
$$

$$
\left.\left.-\frac{1}{2}\left(-\ln(\eta\beta_k) + \bar{Z}_k^2\eta\beta_k\right) + (a_0 - 1)\ln\beta_k - b_0\beta_k\right]\right)
$$

$$
\propto \exp\left(\left(\frac{J+1}{2} + a_0 - 1\right)\ln(\beta_k)-\right.
$$

$$
\left.\beta_k\left(\frac{1}{2}\left(\mathbb{E}_{\mathcal{Q}_{/\beta_k}}\left[\left(\bar{Z}_k^2\eta\right)\right] + \sum_{j=1}^{J}\mathbb{E}_{\mathcal{Q}_{/\beta_k}}\left[\left(W_{jk}^2\alpha_j\right)\right]\right) + b_0\right)\right) \tag{A15}
$$

where we have omitted terms of which $\beta_k$ is conditionally independent. By definition,

$$
\mathcal{Q}_{\beta_k} = \exp\left(\ln(\Gamma(\beta_k|a_{\beta_k}, b_{\beta_k}))\right) = \exp\left(\ln\left(\frac{b_{\beta_k}^{a_{\beta_k}}}{\Gamma(a_{\beta_k})}\eta^{a_{\beta_k}-1}\exp\left(-b_{\beta_k}\eta\right)\right)\right)
$$

$$
\propto \exp\left((a_{\beta_k} - 1)\ln\eta - b_{\beta_k}\eta\right) \tag{A16}
$$

By equating Equations (A15) and (A16), the updates for $\mathcal{Q}_\eta$ are

$$
a_{\beta_k} \leftarrow \frac{J+1}{2} + a_0 \tag{A17}
$$

$$
b_{\beta_k} \leftarrow \frac{1}{2}\left(\mathbb{E}_{\mathcal{Q}_{/\beta_k}}\left[\left(\bar{Z}_k^2\eta\right)\right] + \sum_{j=1}^{J}\mathbb{E}_{\mathcal{Q}_{/\beta_k}}\left[\left(W_{jk}^2\alpha_j\right)\right]\right) + b_0 \tag{A18}
$$

**Derivation of Equations (20) and (21)**

According to Equations (13) and the posterior Equation (A6), the update formulae for $\mathcal{Q}_{\bar{Z}}$ are

$$
\mathcal{Q}_{\bar{Z}} \leftarrow \frac{\exp\left(\mathbb{E}_{\mathcal{Q}_{/\bar{Z}}}[\ln\left(\Pr[Z, W, \bar{Z}, \sigma, \alpha, \beta, \eta, Y])\right]\right)}{\int \exp\left(\mathbb{E}_{\mathcal{Q}_{/\bar{Z}}}[\ln(\Pr[Z, W, \bar{Z}, \sigma, \alpha, \beta, \eta, Y])]\right) d\bar{Z}}
$$

$$
\propto \exp\left(\mathbb{E}_{\mathcal{Q}_{/\bar{Z}}}[\ln\left(\Pr[Z, W, \bar{Z}, \sigma, \alpha, \beta, \eta, Y])\right]\right)
$$

$$
\propto \exp\left(\mathbb{E}_{\mathcal{Q}_{/\bar{Z}}}[\ln\left(\Pr[Y|Z, W, \bar{Z}, \sigma]\Pr[\bar{Z}|\eta, \beta])\right]\right)
$$

$$
\propto \exp\left(\mathbb{E}_{\mathcal{Q}_{/\bar{Z}}}\left[-\frac{1}{2}\sum_{i=1}^{P}\left(N_i\ln(2\pi\sigma^2) + \sigma^{-2}||Y - (Z_iW + \bar{Z})\Phi_i||_2^2\right)-\right.\right.
$$

$$
\left.\left.\frac{1}{2}\sum_{k=1}^{K}\left(-\ln(2\pi\eta\beta_k) + \bar{Z}_k^2\eta\beta_k\right)\right]\right)
$$

$$
\propto \exp\left(-\frac{1}{2}\left(\bar{Z}\mathbb{E}_{\mathcal{Q}_{/\bar{Z}}}\left[\sigma^{-2}\sum_{i=1}^{P}\Psi_i + \eta\operatorname{diag}(\beta)\right]\bar{Z}^T-\right.\right.
$$

$$
\left.\left.2\mathbb{E}_{\mathcal{Q}_{/\bar{Z}}}\left[\sigma^{-2}\right]\sum_{i=1}^{P}(Y - \mathbb{E}_{\mathcal{Q}_{/\bar{Z}}}[Z_iW]\Phi_i)\Phi_i^T\bar{Z}^T\right)\right) \tag{A19}
$$

where we have omitted terms of which $\bar{Z}$ is conditionally independent. By definition,

$$\mathcal{Q}_{\bar{Z}} = \exp(\ln(\mathcal{N}(\bar{Z}|\mu_{\bar{Z}}, \Sigma_{\bar{Z}}))) = \exp\left(-\frac{1}{2}\left(\ln|2\pi\Sigma_{\bar{Z}}| + (\bar{Z} - \mu_{\bar{Z}})\Sigma_{\bar{Z}}^{-1}(\bar{Z} - \mu_{\bar{Z}})^T\right)\right)$$

$$\propto \exp\left(-\frac{1}{2}\left(\bar{Z}\Sigma_{\bar{Z}}^{-1}\bar{Z}^T - 2\mu_{\bar{Z}}\Sigma_{\bar{Z}}^{-1}\bar{Z}^T\right)\right) \tag{A20}$$

By equating Equations (A19) and (A20), the updates for $\mathcal{Q}_{\bar{Z}}$ are

$$\Sigma_{\bar{Z}} \leftarrow \left(\mathbb{E}_{\mathcal{Q}_{/\bar{Z}}}\left[\sigma^{-2}\sum_{i=1}^{P}(\Psi_i) + \eta\,\mathrm{diag}(\beta)\right]\right)^{-1} \tag{A21}$$

$$\mu_{\bar{Z}} \leftarrow \left(\mathbb{E}_{\mathcal{Q}_{/\bar{Z}}}\left[\sigma^{-2}\right]\sum_{i=1}^{P}(Y - \mathbb{E}_{\mathcal{Q}_{/\bar{Z}}}[Z_iW]\Phi_i)\Phi_i^T\right)\Sigma_{\bar{Z}} \tag{A22}$$

**Derivation of Equations (22) and (23)**

According to Equation (13) and the posterior Equation (A6), the update formulae for $\mathcal{Q}_W$ are

$$\mathcal{Q}_W \leftarrow \frac{\exp\left(\mathbb{E}_{\mathcal{Q}_{/W}}\left[\ln\left(\Pr[Z, W, \bar{Z}, \sigma, \alpha, \beta, \eta, Y]\right)\right]\right)}{\int \exp\left(\mathbb{E}_{\mathcal{Q}_{/W}}\left[\ln(\Pr[Z, W, \bar{Z}, \sigma, \alpha, \beta, \eta, Y])\right]\right)dW}$$

$$\propto \exp\left(\mathbb{E}_{\mathcal{Q}_{/W}}\left[\ln\left(\Pr[Z, W, \bar{Z}, \sigma, \alpha, \beta, \eta, Y]\right)\right]\right)$$

$$\propto \exp\left(\mathbb{E}_{\mathcal{Q}_{/W}}\left[\ln\left(\Pr[Y|Z, W, \bar{Z}, \sigma]\Pr[W|\alpha, \beta]\right)\right]\right)$$

$$\propto \exp\left(\mathbb{E}_{\mathcal{Q}_{/W}}\left[-\frac{1}{2}\sum_{i=1}^{P}\left(N_i\ln(2\pi\sigma^2) + \sigma^{-2}||Y - (Z_iW + \bar{Z})\Phi_i||_2^2\right)\right]\right)$$

$$\exp\left(\mathbb{E}_{\mathcal{Q}_{/W}}\left[-\frac{1}{2}\left(\ln|2\pi(\mathrm{diag}(\beta) \otimes \mathrm{diag}(\alpha))^{-1}| +\right.\right.\right.$$

$$\left.\left.\left.\mathrm{vec}(W)^T(\mathrm{diag}(\beta) \otimes \mathrm{diag}(\alpha))\,\mathrm{vec}(W)\right)\right]\right)$$

$$\propto \exp\left(\mathbb{E}_{\mathcal{Q}_{/W}}\left[-\frac{1}{2}\left(\sigma^{-2}\sum_{i=1}^{P}\left(-2Y_i\Phi_i^TW^TZ_i^T + 2Z_iW\Psi_i\bar{Z}^T + Z_iW\Psi_iW^TZ_i^T\right)\right)\right]\right)$$

$$\exp\left(\mathbb{E}_{\mathcal{Q}_{/W}}\left[-\frac{1}{2}\left(\mathrm{vec}(W)^T(\mathrm{diag}(\beta) \otimes \mathrm{diag}(\alpha))\,\mathrm{vec}(W)\right)\right]\right)$$

$$\propto \exp\left(-\frac{1}{2}\mathbb{E}_{\mathcal{Q}_{/W}}\left[-2\sigma^{-2}\sum_{i=1}^{P}\mathrm{vec}\left(\left(\Phi_i(\Phi_i^T\bar{Z}^T - Y_i^T)Z_i\right)^T\right)^T\right]\mathrm{vec}(W)\right)$$

$$\exp\left(-\frac{1}{2}\mathrm{vec}(W)^T\mathbb{E}_{\mathcal{Q}_{/W}}\left[\sigma^{-2}\sum_{i=1}^{P}\left(\Psi \otimes (Z_i^TZ_i)\right) + (\mathrm{diag}(\beta) \otimes \mathrm{diag}(\alpha))\right]\mathrm{vec}(W)\right) \tag{A23}$$

where we have omitted terms of which $W$ is conditionally independent. By definition,

$$\mathcal{Q}_W = \exp\left(\ln(\mathcal{N}(\mathrm{vec}(W)|\mu_{\mathrm{vec}(W)}, \Sigma_{\mathrm{vec}(W)}))\right)$$

$$= \exp\left(-\frac{1}{2}\left(\ln|2\pi\Sigma_{\mathrm{vec}(W)}| +\right.\right.$$

$$\left.\left.(\mathrm{vec}(W)^T - \mu_{\mathrm{vec}(W)})\Sigma_{\mathrm{vec}(W)}^{-1}(\mathrm{vec}(W)^T - \mu_{\mathrm{vec}(W)})^T\right)\right)$$

$$\propto \exp\left(-\frac{1}{2}\left(\mathrm{vec}(W)^T\Sigma_{\mathrm{vec}(W)}^{-1}\mathrm{vec}(W) - 2\mu_{\mathrm{vec}(W)}\Sigma_{\mathrm{vec}(W)}^{-1}\mathrm{vec}(W)\right)\right) \tag{A24}$$

By equating Equations (A23) and (A24), the updates for $\mathcal{Q}_W$ are

$$
\Sigma_{\text{vec}(W)} \leftarrow \mathbb{E}_{\mathcal{Q}_{/W}} \left[ \sigma^{-2} \sum_{i=1}^{P} \left( \Psi \otimes (Z_i^T Z_i) \right) + (\text{diag}(\beta) \otimes \text{diag}(\alpha)) \right]^{-1}
$$

$$
\mu_{\text{vec}(W)} \leftarrow \mathbb{E}_{\mathcal{Q}_{/W}} \left[ -\sigma^{-2} \sum_{i=1}^{P} \text{vec} \left( \left( \Phi_i (\Phi_i^T \bar{Z}^T - Y_i^T) Z_i \right)^T \right)^T \right] \Sigma_{\text{vec}(W)} \tag{A25}
$$

**Derivation of Equations (24)–(26)**

According to Equation (13) and the posterior Equation (A6), the update formulae for $\mathcal{Q}_{Z_i}$ are

$$
\mathcal{Q}_{Z_i} \leftarrow \frac{\exp \left( \mathbb{E}_{\mathcal{Q}_{/Z_i}} [\ln (\Pr[Z, W, \bar{Z}, \sigma, \alpha, \beta, \eta, Y])] \right)}{\int \exp \left( \mathbb{E}_{\mathcal{Q}_{/Z_i}} [\ln(\Pr[Z, W, \bar{Z}, \sigma, \alpha, \beta, \eta, Y])] \right) dZ_i}
$$

$$
\propto \exp \left( \mathbb{E}_{\mathcal{Q}_{/Z_i}} [\ln (\Pr[Z, W, \bar{Z}, \sigma, \alpha, \beta, \eta, Y])] \right)
$$

$$
\propto \exp \left( \mathbb{E}_{\mathcal{Q}_{/Z_i}} [\ln (\Pr[Y_i | Z_i, W, \bar{Z}, \sigma] \Pr[Z_i])] \right)
$$

$$
\propto \exp \left( \mathbb{E}_{\mathcal{Q}_{/Z_i}} \left[ -\frac{1}{2} \left( N_i \ln(2\pi\sigma^2) + \sigma^{-2} || Y - (Z_i W + \bar{Z})\Phi_i ||_2^2 + J \ln(2\pi) + Z_i Z_i^T \right) \right] \right)
$$

$$
\propto \exp \left( -\frac{1}{2} \left( Z_i \mathbb{E}_{\mathcal{Q}_{/Z_i}} \left[ \sigma^{-2} W \Psi_i W^T + I \right] Z_i^T - 2 \mathbb{E}_{\mathcal{Q}_{/Z_i}} \left[ \sigma^{-2} (Y_i - Z_i \Phi_i) \Phi_i^T W^T \right] Z_i^T \right) \right) \tag{A26}
$$

where we have omitted terms of which $Z_i$ is conditionally independent. By definition,

$$
\mathcal{Q}_{Z_i} = \exp \left( \ln(\mathcal{N}(Z_i | \mu_{Z_i}, \Sigma_{Z_i})) \right)
$$

$$
= \exp \left( -\frac{1}{2} \left( \ln |2\pi \Sigma_{Z_i}| + (Z_i - \mu_{Z_i}) \Sigma_{Z_i}^{-1} (Z_i - \mu_{Z_i})^T \right) \right)
$$

$$
\propto \exp \left( -\frac{1}{2} \left( Z_i \Sigma_{Z_i}^{-1} Z_i^T - 2 \mu_{Z_i} \Sigma_{Z_i}^{-1} Z_i^T \right) \right) \tag{A27}
$$

By equating Equations (A26) and (A27), the updates for $\mathcal{Q}_{\bar{Z}}$ are

$$
\Sigma_{Z_i} \leftarrow \left( \mathbb{E}_{\mathcal{Q}_{/Z_i}} [\sigma^{-2} W \Psi_i W^T + I] \right)^{-1} \tag{A28}
$$

$$
\mu_{Z_i} \leftarrow \mathbb{E}_{\mathcal{Q}_{/Z_i}} [\sigma^{-2} (Y_i - \bar{Z}\Phi_i) \Phi_i^T W^T] \Sigma_{Z_i} \tag{A29}
$$

**Derivation of Equations (27)–(29)**

According to Equation (13) and the posterior Equation (A6), the update formulae for $\mathcal{Q}_\sigma$ are

$$
\mathcal{Q}_\sigma \leftarrow \frac{\exp \left( \mathbb{E}_{\mathcal{Q}_{/\sigma}} [\ln (\Pr[Z, W, \bar{Z}, \sigma, \alpha, \beta, \eta, Y])] \right)}{\int \exp \left( \mathbb{E}_{\mathcal{Q}_{/\sigma}} [\ln(\Pr[Z, W, \bar{Z}, \sigma, \alpha, \beta, \eta, Y])] \right) d\sigma}
$$

$$
\propto \exp \left( \mathbb{E}_{\mathcal{Q}_{/\sigma}} [\ln (\Pr[Z, W, \bar{Z}, \sigma, \alpha, \beta, \eta, Y])] \right)
$$

$$
\propto \exp \left( \mathbb{E}_{\mathcal{Q}_{/\sigma}} [\ln (\Pr[Y_i | Z_i, W, \bar{Z}, \sigma] \Pr[\sigma])] \right)
$$

$$
\propto \exp \left( \mathbb{E}_{\mathcal{Q}_{/\sigma}} \left[ -\frac{1}{2} \sum_{i=1}^{P} \left( N_i \ln(2\pi\sigma^2) + \sigma^{-2} || Y - (Z_i W + \bar{Z})\Phi_i ||_2^2 \right) \right] \right)
$$

$$
\exp \left( \mathbb{E}_{\mathcal{Q}_{/\sigma}} \left[ \left( (a_0 - 1) \ln \sigma^{-2} - b_0 \sigma^{-2} \right) \right] \right)
$$

$$
\propto \exp \left( \left( a_0 + \frac{1}{2} \sum_i N_i - 1 \right) \ln (\sigma^{-2}) - \right.
$$

$$
\left. \sigma^{-2} \left( b_0 + \frac{1}{2} \mathbb{E}_{\mathcal{Q}_{/\sigma}} [\sum_i || Y_i - (Z_i W + \bar{Z})\Phi_i ||_2^2] \right) \right) \tag{A30}
$$

where we have omitted terms of which $\sigma$ is conditionally independent. By definition,

$$\mathcal{Q}_\sigma = \exp\left(\ln(\Gamma(\sigma^{-2}|a_\sigma, b_\sigma))\right) = \exp\left(\ln\left(\frac{b_\sigma^{a_\sigma}}{\Gamma(a_\sigma)}(\sigma^{-2})^{a_\sigma-1}\exp\left(-b_\sigma\sigma^{-2}\right)\right)\right)$$

$$\propto \exp\left((a_\sigma - 1)\ln\sigma^{-2} - b_\sigma\sigma^{-2}\right) \tag{A31}$$

By equating Equations (A30) and (A31), the updates for $\mathcal{Q}_\sigma$ are

$$a_\sigma \leftarrow a_0 + \frac{1}{2}\sum_i N_i \tag{A32}$$

$$b_\sigma \leftarrow b_0 + \frac{1}{2}\mathbb{E}_{\mathcal{Q}_{/\sigma}}\left[\sum_i ||Y_i - (Z_i W + \bar{Z})\Phi_i||_2^2\right] \tag{A33}$$

## Appendix C. Scalable Update for BSFDA

*Appendix C.1. Implicit Factorization*

We initialize the inactive precision parameters as

$$\mathbb{E}_{\mathcal{Q}_{\alpha_j}}[\alpha_j] = \epsilon^{-1}, \forall j > J^{(a)} \tag{A34}$$

$$\mathbb{E}_{\mathcal{Q}_{\beta_k}}[\beta_k] = \epsilon^{-1}, \forall k > K^{(a)} \tag{A35}$$

Under these settings and subsequent variational updates (using Equations (A34) and (A35)), in the limit as $\epsilon \to 0$, the surrogate distributions satisfy

$$\mu_{Z_{iB}} = 0, \ \Sigma_{Z_iB} = \epsilon I^{(J-J^{(a)})}, \ \Sigma_{Z_i[A,B]} = \left(\Sigma_{Z_i[B,A]}\right)^T = 0 \tag{A36}$$

$$\mu_{\bar{Z}B} = 0, \ \Sigma_{\bar{Z}[B,B]} = \epsilon I^{(K-K^{(a)})}, \ \Sigma_{\bar{Z}[A,B]} = \Sigma_{\bar{Z}[B,A]}^T = 0 \tag{A37}$$

$$\mu_{\text{vec}(W)_B} = 0, \ \mu_{\text{vec}(W)_C} = 0, \ \mu_{\text{vec}(W)_D} = 0, \ \Sigma_{\text{vec}(W)[B,B]} = \epsilon I^{(K^{(a)}J - K^{(a)}J^{(a)})},$$

$$\Sigma_{\text{vec}(W)[C,C]} = \epsilon I^{(J^{(a)}K - J^{(a)}K^{(a)})}, \ \Sigma_{\text{vec}(W)[D,D]} = \epsilon I^{(JK + J^{(a)}K^{(a)} - JK^{(a)} - J^{(a)}K)},$$

$$\Sigma_{\text{vec}(W)[x,y]} = 0, \forall(x,y) \notin \{(A,A), (B,B), (C,C), (D,D)\} \tag{A38}$$

For convenience, we initialize $\mathcal{Q}$ with the above properties.

**Lemma A1.** *If $\mathcal{Q}_{\alpha_j}[\alpha_j] = \epsilon, \forall j \geq J^{(a)}$ and $\mathcal{Q}_{\beta_k}[\beta_k] = \epsilon, \forall k \geq K^{(a)}$, then the variational distribution over $W$ factorizes as $\mathcal{Q}_W = \mathcal{Q}_{W_A}\mathcal{Q}_{W_B}\mathcal{Q}_{W_C}\mathcal{Q}_{W_D}$ in the limit as $\epsilon \to 0$.*

**Proof.** We express the distribution as

$$\mathcal{Q}_W = \mathcal{N}(\text{vec}(W)|\mu_{\text{vec}(W)}, \Sigma_{\text{vec}(W)})$$

$$= \exp\left(-\frac{1}{2}\left(\ln|2\pi\Sigma_{\text{vec}(W)}| + \mu_{\text{vec}(W)}\Sigma_{\text{vec}(W)}^{-1}\mu_{\text{vec}(W)}^T\right)\right).$$

The factorization holds if the off-diagonal block matrices in $\Sigma_{\text{vec}(W)}$, e.g., $\Sigma_{[W_A, W_B]}$, are all zero, i.e., the blocks are mutually independent. Initially, this is ensured by the definition for the initial status in Equation (A38). Thus, we only need to show that the statement remains true after $\mathcal{Q}_W$ is updated, i.e., after Equation (22) is applied with the inactive scale parameters $\mathcal{Q}_{\alpha_j}[\alpha_j]$ and $\mathcal{Q}_{\beta_k}[\beta_k]$ fixed at $\epsilon$. First, we regard $\Sigma_{[W_{ABC}]}$, i.e., the covariance of the union of $W_A, W_B, W_C$ after vectorization, as one block. By the block matrix inversion formula, we obtain $\Sigma_{[W_{ABC}, W_D]} \propto \epsilon^2 \to 0$ and consequently $\mathcal{Q}_W = \mathcal{Q}_{W_{ABC}}\mathcal{Q}_{W_D}$. Next,

we apply the block matrix inversion formula to $\Sigma_{[W_{ABC}]}$ in Equation (22) and we obtain $(\Sigma_{[W_B,W_A]}, \Sigma_{[W_B,W_C]}, \Sigma_{[W_C,W_A]}, \Sigma_{[W_C,W_B]}) \propto \epsilon \to 0$, yielding the desired factorization. □

**Lemma A2.** *If* $\mathcal{Q}_{\beta_k}[\beta_k] = \epsilon, \forall k \geq K^{(a)}$, *then the implicit factorization* $\mathcal{Q}_{\bar{Z}} = \mathcal{Q}_{\bar{Z}_A}\mathcal{Q}_{\bar{Z}_B}$ *holds in the limit as* $\epsilon \to 0$.

**Proof.** The proof is similar to the proof for Lemma A1. Because $\mathcal{Q}_{\bar{Z}} = \mathcal{N}(\mu_{\bar{Z}}, \Sigma_{\bar{Z}})$, we need only the off-diagonal block to be zero, i.e., $\Sigma_{\bar{Z}[A,B]} = 0$. Initially, this is ensured by definition for the initial status in Equation (A37). $\mathcal{Q}_{\bar{Z}}$ is updated by Equation (20). Applying the block matrix inversion formula with the inactive $\mathcal{Q}_{\beta_k}[\beta_k]$, we obtain $\Sigma_{\bar{Z}[A,B]} \propto \epsilon \to 0$, establishing the factorization. □

**Lemma A3.** *If* $j \geq J^{(a)}$ *or* $k \geq K^{(a)}$, *then* $\mathbb{E}_{\mathcal{Q}_{/Z_i}}[W_{kj'}W_{jk'}] \propto \mathcal{O}(\epsilon), \forall j' = 1 : J, k' = 1 : K$ *in the limit as* $\epsilon \to 0$.

**Proof.** For the initial status, apparently, the largest $\mathbb{E}_{\mathcal{Q}_{/Z_i}}[W_{kj'}W_{jk'}]$ is $\mathbb{E}_{\mathcal{Q}_{/Z_i}}[W_{kj'}^2] = \epsilon$. Because either $\mathcal{Q}_{\alpha_j}[\alpha_j] = \epsilon$ or $\mathcal{Q}_{\beta_k}[\beta_k] = \epsilon$, after updates from Equations (22) and (23) are applied, $\mathbb{E}_{\mathcal{Q}_{/Z_i}}[W_{kj'}W_{jk'}] = \Sigma_{[W_{kj'},W_{jk'}]} + \mu_{\text{vec}(W)_{kj'}}\mu_{\text{vec}(W)_{jk'}} \propto \mathcal{O}(\epsilon)$ by the Woodbury matrix identity. □

**Lemma A4.** *If* $\mathbb{E}_{\mathcal{Q}_{\alpha_j}}[\alpha_j] = \frac{a_{\alpha_j}}{b_{\alpha_j}} = \epsilon^{-1}, \forall j \geq J^{(a)}$, *then the implicit factorization* $\mathcal{Q}_{Z_i} = \mathcal{Q}_{Z_{iA}}\mathcal{Q}_{Z_{iB}}$ *holds in the limit as* $\epsilon \to 0$.

**Proof.** The proof is similar to the proof for Lemma A1. Because $\mathcal{Q}_{Z_i} = \mathcal{N}(\mu_{Z_i}, \Sigma_{Z_i})$, only $\Sigma_{Z_i[A,B]} = 0$ is needed. Initially, this is ensured by definition for the initial status Equation (A36). $\mathcal{Q}_Z$ is updated by Equations (24) and (25). In Equation (24), when $j \geq J^{(a)}$ or $k \geq K^{(a)}$, $C_{ijk} = \text{Tr}(\mathbb{E}_{\mathcal{Q}_{/Z_i}}[W_{k\cdot}^T W_{j\cdot}]\Psi_i) = \sum_{(j',k')}\left(\mathbb{E}_{\mathcal{Q}_{/Z_i}}[W_{kj'}W_{jk'}](\Psi_i)_{k'j'}\right) \propto \mathcal{O}(\epsilon) \to 0$ applying Lemma A3. Applying the block matrix inversion formula to Equation (25), $\Sigma_{Z_i[AB]} \propto \mathcal{O}(\epsilon) \to 0$, thus proving the implicit factorization. □

*Appendix C.2. Scale Parameters*

Here, we state the theorems that justify the use of updating rules for $\mathcal{Q}^{(a)}{}_{\alpha_j}$ based on $\mathcal{L}^{(a)}$ to update $\mathcal{Q}_{\alpha_j}$ (and, similarly, $\mathcal{Q}^{(a)}{}_{\beta_k}$ for $\mathcal{Q}_{\beta_k}$, $\mathcal{Q}^{(a)}{}_{\eta}$ for $\mathcal{Q}_{\eta}$), and it does maximize $\mathcal{L}$ ultimately.

**Lemma A5.** $\forall W_{jk} \in W_B \cup W_C$, *i.e., either* $(j > J^{(a)})$ *or* $(k > K^{(a)})$, *after updating* $\mathcal{Q}_{W_B}$ *and* $\mathcal{Q}_{W_C}$ *by Equations* (22) *and* (23), $\mathbb{E}_{\mathcal{Q}}[W_{jk}^2] = \frac{b_{\alpha_j}b_{\beta_k}}{a_{\alpha_j}a_{\beta_k}}$.

**Proof.** According to Equations (A34) and (A35), if $(j > J^{(a)})$ or $(k > K^{(a)})$, either $\mathbb{E}_{\mathcal{Q}}[\alpha_j] = \epsilon^{-1}$ or $\mathbb{E}_{\mathcal{Q}}[\beta_k] = \epsilon^{-1}$, respectively.

In the limit as $\epsilon \to 0$, using Equation (22) and the block matrix inversion formula, we obtain

$$\Sigma_{W_{jk}} \leftarrow \lim_{\epsilon \to 0}\left(\left(\mathbb{E}_{\mathcal{Q}_{/W}}[\text{diag}(\beta) \otimes \text{diag}(\alpha) + \sigma^{-2}\sum_i\left((\Psi_i) \otimes (Z_i^T Z_i)\right)]\right)^{-1}\right)_{[j+kM,j+kM]}$$

$$= \lim_{\epsilon \to 0}\left(\left(\mathbb{E}_{\mathcal{Q}_{/W}}[\alpha_j\beta_k]\right)^{-1} + \mathcal{O}(\epsilon^2)\right) = \left(\mathbb{E}_{\mathcal{Q}_{/W}}[\alpha_j\beta_k]\right)^{-1} = \frac{b_{\alpha_j}b_{\beta_k}}{a_{\alpha_j}a_{\beta_k}} \tag{A39}$$

In the limit as $\epsilon \to 0$ and using Equation (23),

$$
\mu_{W_{jk}} \leftarrow \lim_{\epsilon \to 0} \left( -\frac{a_\sigma}{b_\sigma} \sum_i \mathrm{vec}\left( \left( \Phi_i (\mu_{\bar{Z}} \Phi_i - Y_i)^T \mu_{Z_i} \right)^T \right)^T \Sigma_{\mathrm{vec}\,(W)} \right)_{[1, j+kM]}
$$

$$
= \left( -\frac{a_\sigma}{b_\sigma} \sum_i \mathrm{vec}\left( \left( \Phi_i (\mu_{\bar{Z}} \Phi_i - Y_i)^T \mu_{Z_i} \right)^T \right)^T \right) \left( \Sigma_{\mathrm{vec}\,(W)} \right)_{\cdot (j+kM)} \in \mathcal{O}(\epsilon) \qquad (A40)
$$

Equation (A40) uses the fact that elements in $\left( \Sigma_{\mathrm{vec}\,(W)} \right)_{\cdot (j+kM)}$ are all $\mathcal{O}(\epsilon)$ based on the block matrix inversion formula. Thus,

$$
\lim_{\epsilon \to 0} \mathbb{E}_{\mathcal{Q}}[W_{jk}^2] = \lim_{\epsilon \to 0} \left( \Sigma_{W_{jk}} + (\mu_{W_{jk}})^2 \right) = \lim_{\epsilon \to 0} \left( \Sigma_{W_{jk}} + \mathcal{O}(\epsilon^2) \right)
$$

$$
= \lim_{\epsilon \to 0} \left( \frac{b_{\alpha_j} b_{\beta_k}}{a_{\alpha_j} a_{\beta_k}} + \mathcal{O}(\epsilon^2) \right) = \frac{b_{\alpha_j} b_{\beta_k}}{a_{\alpha_j} a_{\beta_k}} \qquad (A41)
$$

□

**Lemma A6.** $\forall k > K^{(a)}$, after updating $\mathcal{Q}_{\bar{Z}_B}$ by Equations (20) and (21), $\mathbb{E}_{\mathcal{Q}}[\bar{Z}_k^2] = \frac{b_\eta}{a_\eta} \epsilon$.

**Proof.** If $k > K^{(a)}$, $\mathbb{E}_{\mathcal{Q}}[\beta_k] = \epsilon^{-1}$.
Then, using Equation (20) and the block matrix inversion formula, we have

$$
\Sigma_{\bar{Z}kk} \leftarrow \lim_{\epsilon \to 0} \left( \left( \mathbb{E}_{\mathcal{Q}_{/\bar{Z}}} \left[ \sum_{i=1}^{P} \left( \sigma^{-2} \Psi_i \right) + \eta \, \mathrm{diag}(\beta) \right] \right)^{-1} \right)_{kk}
$$

$$
= \lim_{\epsilon \to 0} \left( \left( \sum_{i=1}^{P} \left( \frac{a_\sigma}{b_\sigma} \Psi_i \right) + \frac{a_\eta}{b_\eta} \, \mathrm{diag}(\frac{a}{b}) \right)^{-1} \right)_{kk}
$$

$$
= \lim_{\epsilon \to 0} \left( \frac{b_\eta b_{\beta_k}}{a_\eta a_{\beta_k}} + \mathcal{O}(\epsilon^2) \right) = \frac{b_\eta b_{\beta_k}}{a_\eta a_{\beta_k}} \qquad (A42)
$$

Using Equation (21),

$$
\mu_{\bar{Z}k} \leftarrow \lim_{\epsilon \to 0} \left( \left( \mathbb{E}_{\mathcal{Q}_{/\bar{Z}}} \left[ \sigma^{-2} \right] \sum_{i=1}^{P} (Y - \mathbb{E}_{\mathcal{Q}_{/\bar{Z}}}[Z_i W] \Phi_i) \Phi_i^T \right) \Sigma_{\bar{Z}} \right)_{1k}
$$

$$
= \lim_{\epsilon \to 0} \left( \left( \frac{a_\sigma}{b_\sigma} \sum_{i=1}^{P} (Y - \mu_{Z_i} \mu_W \Phi_i) \Phi_i^T \right) \Sigma_{\bar{Z}} \right)_{1k}
$$

$$
= \lim_{\epsilon \to 0} \left( \frac{a_\sigma}{b_\sigma} \sum_{i=1}^{P} (Y - \mu_{Z_i} \mu_W \Phi_i) \Phi_i^T \right) \Sigma_{\bar{Z} \cdot k} \in \mathcal{O}(\epsilon) \qquad (A43)
$$

Equation (A43) uses the fact that elements in $\Sigma_{\bar{Z} \cdot k}$ are all $\in \mathcal{O}(\epsilon)$.

$$
\mathbb{E}_{\mathcal{Q}}[\bar{Z}_k^2] = \lim_{\epsilon \to 0} \left( \Sigma_{\bar{Z}kk} + \mu_{\bar{Z}k}^2 \right) = \lim_{\epsilon \to 0} \left( \Sigma_{\bar{Z}kk} + \mathcal{O}(\epsilon^2) \right)
$$

$$
= \lim_{\epsilon \to 0} \left( \frac{b_\eta b_{\beta_k}}{a_\eta a_{\beta_k}} + \mathcal{O}(\epsilon^2) \right) = \lim_{\epsilon \to 0} \left( \frac{b_\eta}{a_\eta} \epsilon + \mathcal{O}(\epsilon^2) \right) = \frac{b_\eta}{a_\eta} \epsilon \qquad (A44)
$$

□

**Theorem A1.** $\forall j \leq J^{(a)}$, updates of $\mathcal{Q}_{\alpha_j}$ and $\mathcal{Q}_{W_B}$ will converge at $\mathbb{E}_{\mathcal{Q}_{\alpha_j}}[\alpha_j] = \mathbb{E}_{\mathcal{Q}_{\alpha_j}^{(a)}}[\alpha_j]$ given that $\mathbb{E}_{\mathcal{Q}_{\beta_k}}[\beta_k] = \mathbb{E}_{\mathcal{Q}_{\beta_k}^{(a)}}[\beta_k], \forall k \leq K^{(a)}$, $a_0 = b_0 = 0$ and the conditions in Equations (A35) and (A36) are satisfied in the limit as $\epsilon \to 0$.

**Proof.** Assume that $\mathcal{Q}_{\alpha_j}^{(a)}$ has just been updated using Equations (14) and (15), i.e., $\forall j \le J^{(a)}$

$$a_{\alpha_j}^{(a)} = a_0 + \frac{K^{(a)}}{2} \tag{A45}$$

$$b_{\alpha_j}^{(a)} = b_0 + \frac{1}{2} \sum_{k=1}^{K^{(a)}} \mathbb{E}_{\mathcal{Q}_{/\alpha_j}^{(a)}} [W_{jk}^2 \beta_k]$$

$$= b_0 + \frac{1}{2} \sum_{k=1}^{K^{(a)}} \left( \left( \Sigma_{W_{jk}} + \mu_{W_{jk}}^2 \right) \frac{a_{\beta_k}^{(a)}}{b_{\beta_k}^{(a)}} \right) \tag{A46}$$

The updates for $\mathcal{Q}_\alpha$ derived from $\mathcal{L}$ are

$$b_{\alpha_j} \leftarrow b_0 + \frac{1}{2} \sum_{k=1}^{K} \left( \left( \Sigma_{W_{jk}} + \mu_{W_{jk}}^2 \right) \frac{a_{\beta_k}}{b_{\beta_k}} \right)$$

$$= b_0 + \frac{1}{2} \sum_{k=1}^{K^{(a)}} \left( \left( \Sigma_{W_{jk}} + \mu_{W_{jk}}^2 \right) \frac{a_{\beta_k}}{b_{\beta_k}} \right)$$

$$+ \frac{1}{2} \sum_{k=K^{(a)}+1}^{K} \left( \left( \Sigma_{W_{jk}} + \mu_{W_{jk}}^2 \right) \frac{a_{\beta_k}}{b_{\beta_k}} \right)$$

$$= b_{\alpha_j}^{(a)} + \frac{1}{2} \sum_{k=K^{(a)}+1}^{K} \left( \left( \Sigma_{W_{jk}} + \mu_{W_{jk}}^2 \right) \frac{a_{\beta_k}}{b_{\beta_k}} \right) \tag{A47}$$

It involves $W_{jk}, k > K^{(a)}$ and therefore they need to be kept updated. Applying Theorem A5 for Equation (A47), we can obtain

$$b_{\alpha_j} \leftarrow b_{\alpha_j}^{(a)} + \frac{1}{2} \sum_{k=K^{(a)}+1}^{K} \left( \left( \frac{b_{\alpha_j} b_{\beta_k}}{a_{\alpha_j} a_{\beta_k}} \right) \frac{a_{\beta_k}}{b_{\beta_k}} \right)$$

$$= b_{\alpha_j}^{(a)} + \frac{1}{2} (K - K^{(a)}) \frac{b_{\alpha_j}}{a_{\alpha_j}} \tag{A48}$$

Applying Equation (A48) in an iterative manner, we will obtain a sequence of updates for $a_{\alpha_j}$. Solving

$$b_{\alpha_j} = b_{\alpha_j}^{(a)} + \frac{1}{2} (K - K^{(a)}) \frac{b_{\alpha_j}}{\frac{K}{2}} \tag{A49}$$

$$\Rightarrow b_{\alpha_j} = (1 - \frac{1}{2}(K - K^{(a)}) \frac{2}{K})^{-1} b_{\alpha_j}^{(a)} = \frac{K}{K^{(a)}} b_{\alpha_j}^{(a)} \tag{A50}$$

Thus, we find that the sequence will converge at

$$b_{\alpha_j} \leftarrow \frac{K}{K^{(a)}} b_{\alpha_j}^{(a)} \tag{A51}$$

As a result, $\mathbb{E}_{\mathcal{Q}_{\alpha_j}}[\alpha_j] = \frac{a_{\alpha_j}}{b_{\alpha_j}} = \frac{a_{\alpha_j}^{(a)}}{b_{\alpha_j}^{(a)}} = \mathbb{E}_{\mathcal{Q}_{\alpha_j}^{(a)}}[\alpha_j].$ $\square$

**Theorem A2.** $\forall k \le K^{(a)}$, *updates of* $\mathcal{Q}_{\beta_k}$ *and* $\mathcal{Q}_{W_C}$ *will converge at* $\mathbb{E}_{\mathcal{Q}_{\beta_k}}[\beta_k] = \mathbb{E}_{\mathcal{Q}_{\beta_k}^{(a)}}[\beta_k]$ *given that* $\mathbb{E}_{\mathcal{Q}_{\alpha_j}}[\alpha_j] = \mathbb{E}_{\mathcal{Q}_{\alpha_j}^{(a)}}[\alpha_j], \forall j \le J^{(a)}, a_0 = b_0 = 0$ *and the conditions in Equations (A35) and (A36) are satisfied in the limit as* $\epsilon \to 0$.

**Proof.** Assume that $\mathcal{Q}_{\beta_k}^{(a)}$ has just been updated using Equations (18) and (19), i.e.,

$$a_{\beta_k}^{(a)} = a_0 + \frac{K^{(a)} + 1}{2} \tag{A52}$$

$$b_{\beta_k}^{(a)} \leftarrow b_0 + \frac{1}{2} \mathbb{E}_{\mathcal{Q}_{/\beta_k}^{(a)}} [\bar{Z}_k^2 + \sum_{j=1}^{J^{(a)}} W_{jk}^2 \alpha_j]$$

$$= b_0 + \frac{1}{2} \left( \left( \Sigma_{\bar{Z}kk} + \mu_{\bar{Z}k}^2 \right) + \sum_{j=1}^{J^{(a)}} \left( \left( \Sigma_{W_{jk}} + \mu_{W_{jk}}^2 \right) \frac{a_{\alpha_j}^{(a)}}{b_{\alpha_j}^{(a)}} \right) \right) \tag{A53}$$

The update for $\mathcal{Q}_{\beta_k}$ derived from $\mathcal{L}$ is

$$b_{\beta_k} \leftarrow b_0 + \frac{1}{2} \left( \left( \Sigma_{\bar{Z}kk} + \mu_{\bar{Z}k}^2 \right) + \sum_{j=1}^{J} \left( \left( \Sigma_{W_{jk}} + \mu_{W_{jk}}^2 \right) \frac{a_{\alpha_j}}{b_{\alpha_j}} \right) \right)$$

$$= b_0 + \frac{1}{2} \left( \left( \Sigma_{\bar{Z}kk} + \mu_{\bar{Z}k}^2 \right) + \sum_{j=1}^{J^{(a)}} \left( \left( \Sigma_{W_{jk}} + \mu_{W_{jk}}^2 \right) \frac{a_{\alpha_j}}{b_{\alpha_j}} \right) \right)$$

$$+ \frac{1}{2} \left( \sum_{j=J^{(a)}+1}^{J} \left( \left( \Sigma_{W_{jk}} + \mu_{W_{jk}}^2 \right) \frac{a_{\alpha_j}}{b_{\alpha_j}} \right) \right)$$

$$= b_{\beta_k}^{(a)} + \frac{1}{2} \left( \sum_{j=J^{(a)}+1}^{J} \left( \left( \Sigma_{W_{jk}} + \mu_{W_{jk}}^2 \right) \frac{a_{\alpha_j}}{b_{\alpha_j}} \right) \right) \tag{A54}$$

It involves $W_{jk}, j > J^{(a)}$ and therefore they need to be kept updated. Applying Theorem A5 for Equation (A54), we can obtain

$$b_{\beta_k} \leftarrow b_{\beta_k}^{(a)} + \frac{1}{2} \sum_{j=J^{(a)}+1}^{J} \left( \left( \frac{b_{\beta_k} b_{\alpha_j}}{a_{\beta_k} a_{\alpha_j}} \right) \frac{a_{\alpha_j}}{b_{\alpha_j}} \right) \tag{A55}$$

$$\Rightarrow = b_{\beta_k}^{(a)} + \frac{1}{2} (K - K^{(a)}) \frac{b_{\beta_k}}{a_{\beta_k}} \tag{A56}$$

Applying Equation (A56) in an iterative manner, we will obtain a sequence of $b_{\beta_k}$. Solving

$$b_{\beta_k} = b_{\beta_k}^{(a)} + \frac{1}{2} (K - K^{(a)}) \frac{b_{\beta_k}}{\frac{K+1}{2}} \tag{A57}$$

$$b_{\beta_k} = (1 - \frac{1}{2} (K - K^{(a)}) \frac{2}{K+1})^{-1} b_{\beta_k}^{(a)} = \frac{K+1}{K^{(a)}+1} b_{\beta_k}^{(a)} \tag{A58}$$

Thus, we find that the sequence will converge at

$$b_{\beta_k} \leftarrow \frac{K+1}{K^{(a)}+1} b_{\beta_k}^{(a)} \tag{A59}$$

As a result, $\mathbb{E}_{\mathcal{Q}}[\beta_k] = \frac{a_{\beta_k}}{b_{\beta_k}} = \frac{a_{\beta_k}^{(a)}}{b_{\beta_k}^{(a)}} = \mathbb{E}_{\mathcal{Q}^{(a)}}[\beta_k]$. $\square$

**Theorem A3.** *Updates of $\mathcal{Q}_\eta$ and $\mathcal{Q}_{\bar{Z}_B}$ will converge at $\mathbb{E}_{\mathcal{Q}_\eta}[\eta] = \mathbb{E}_{\mathcal{Q}_\eta^{(a)}}[\eta]$ given that $\mathbb{E}_{\mathcal{Q}_{\beta_k}}[\beta_k] = \mathbb{E}_{\mathcal{Q}_{\beta_k}^{(a)}}[\beta_k], \forall k \leq K^{(a)}, a_0 = b_0 = 0$ and the conditions in Equations (A35) and (A36) are satisfied in the limit as $\epsilon \to 0$.*

**Proof.** Assume that $\mathcal{Q}_\eta^{(a)}$ has just been updated using Equations (16) and (17), i.e.,

$$a_\eta^{(a)} \leftarrow a_0 + \frac{K^{(a)}}{2} \tag{A60}$$

$$b_\eta^{(a)} \leftarrow b_0 + \frac{1}{2} \sum_{k=1}^{K^{(a)}} \mathbb{E}_{\mathcal{Q}_{/\eta}}[\bar{Z}_k^2 \beta_k]$$

$$= b_0 + \frac{1}{2} \sum_{k=1}^{K^{(a)}} \left( \left( \Sigma_{\bar{Z}k} + \mu_{\bar{Z}k}^2 \right) \frac{a_{\beta_k}^{(a)}}{b_{\beta_k}^{(a)}} \right) \tag{A61}$$

The update for $\mathcal{Q}_\eta$ derived from $\mathcal{L}$ is

$$b_\eta \leftarrow b_0 + \frac{1}{2} \sum_{k=1}^{K} \left( \left( \Sigma_{\bar{Z}k} + \mu_{\bar{Z}k}^2 \right) \frac{a_{\beta_k}}{b_{\beta_k}} \right)$$

$$= b_0 + \frac{1}{2} \sum_{k=1}^{K^{(a)}} \left( \left( \Sigma_{\bar{Z}k} + \mu_{\bar{Z}k}^2 \right) \frac{a_{\beta_k}}{b_{\beta_k}} \right) + \frac{1}{2} \sum_{k=K^{(a)}+1}^{K} \left( \left( \Sigma_{\bar{Z}k} + \mu_{\bar{Z}k}^2 \right) \frac{a_{\beta_k}}{b_{\beta_k}} \right)$$

$$= b_\eta^{(a)} + \frac{1}{2} \sum_{k=K^{(a)}+1}^{K} \left( \left( \Sigma_{\bar{Z}k} + \mu_{\bar{Z}k}^2 \right) \frac{a_{\beta_k}}{b_{\beta_k}} \right) \tag{A62}$$

It involves $\bar{Z}_k, k > K^{(a)}$ and therefore they need to be kept updated. Applying Lemma A6 for Equation (A62), we can obtain

$$b_\eta \leftarrow b_\eta^{(a)} + \frac{1}{2} \sum_{k=K^{(a)}+1}^{K} \left( \left( \frac{b_\eta b_{\beta_k}}{a_\eta a_{\beta_k}} \right) \frac{a_{\beta_k}}{b_{\beta_k}} \right) \tag{A63}$$

$$= b_\eta^{(a)} + \frac{1}{2} (K - K^{(a)}) \frac{b_\eta}{a_\eta} \tag{A64}$$

Applying Equation (A64) in an iterative manner, we will obtain a sequence of updates for $b_\eta$. Solving

$$b_\eta = b_\eta^{(a)} + \frac{1}{2} (K - K^{(a)}) \frac{b_\eta}{\frac{K}{2}} \tag{A65}$$

$$b_\eta = (1 - \frac{1}{2}(K - K^{(a)}) \frac{2}{K})^{-1} b_\eta^{(a)} = \frac{K}{K^{(a)}} b_\eta^{(a)} \tag{A66}$$

Thus, we find that the sequence will converge at

$$b_\eta \leftarrow \frac{K}{K^{(a)}} b_\eta^{(a)} \tag{A67}$$

As a result, $\mathbb{E}_{\mathcal{Q}}[\eta] = \frac{b_\eta}{a_\eta} = \frac{b_\eta^{(a)}}{a_\eta^{(a)}} = \mathbb{E}_{\mathcal{Q}^{(a)}}[\eta]$. $\square$

In practice, due to limitations in numerical representation, we restrict the values so that the active precision parameter estimates do not truly reach infinity:

$$\mathbb{E}_{\mathcal{Q}_{\alpha_j}}[\alpha_j] \leq \tau_{\max}, \forall j \leq J^{(a)} \tag{A68}$$

$$\mathbb{E}_{\mathcal{Q}_{\beta_k}}[\beta_k] \leq \tau_{\max}, \forall k \leq K^{(a)} \tag{A69}$$

*Appendix C.3. Weights and Noise*

We next describe how to update $\mathcal{Q}_{Z_A}, \mathcal{Q}_{\bar{Z}_A}, \mathcal{Q}_{W_A}, \mathcal{Q}_\sigma$ in a scalable manner, using computation in the $K^{(a)}$-dimension subspace only.

**Theorem A4.** *$\mathcal{L}$ and $\mathcal{L}^{(a)}$ share the same update rule for $Z_{iA}$, i.e.,*

$$
\begin{aligned}
H_{iAjk} &\leftarrow \mathbb{E}_{\mathcal{Q}_{/Z_i}}[W_{Aj}\Phi_{iA}\Phi_{iA}^T W_{Ak}^T] = Tr(\mathbb{E}_{\mathcal{Q}_{/Z_i}}[W_{Ak}^T W_{Aj}]\Phi_{iA}\Phi_{iA}^T) \\
&= Tr\left(\left(\Sigma_{[W_{Ak},W_{Aj}]} + \mu_{[W_{Aj}]}^T \mu_{[W_{Ak}]}\right)\Phi_{iA}\Phi_{iA}^T\right), \forall j = 1:J^{(a)}, k = 1:K^{(a)}
\end{aligned}
\tag{A70}
$$

$$
\Sigma_{Z_{iA}} \leftarrow \left(\mathbb{E}_{\mathcal{Q}_{/Z_i}}[\sigma^{-2}W_A\Phi_{iA}\Phi_{iA}^T W_A^T + I]\right)^{-1} = [\frac{a_\sigma}{b_\sigma}H_{iA} + I]^{-1}
\tag{A71}
$$

$$
\mu_{iA} \leftarrow \mathbb{E}_{\mathcal{Q}_{/Z_i}}[\sigma^{-2}(Y_i - \bar{Z}\Phi_{iA})\Phi_{iA}^T W_A^T]\Sigma_{Z_{iA}} = \frac{a_\sigma}{b_\sigma}(Y_i - \mu_{\bar{Z}A}\Phi_{iA})\Phi_{iA}^T(\mu_{W_A})^T\Sigma_{Z_{iA}}
\tag{A72}
$$

**Proof.** Applying Lemma A3 to Equation (24), we have

$$
\begin{aligned}
H_{iAjk} &\leftarrow Tr\left(\left(\Sigma_{[W_{Ak},W_{Aj}]} + \mu_{[W_{Aj}]}^T \mu_{[W_{Ak}]}\right)\Phi_{iA}\Phi_{iA}^T\right) + \mathcal{O}(\epsilon) \\
&\rightarrow Tr\left(\left(\Sigma_{[W_{Ak},W_{Aj}]} + \mu_{[W_{Aj}]}^T \mu_{[W_{Ak}]}\right)\Phi_{iA}\Phi_{iA}^T\right), \forall j = 1:J^{(a)}, k = 1:K^{(a)}
\end{aligned}
\tag{A73}
$$

Applying the block matrix inversion formula to Equation (25), we have

$$
\begin{aligned}
\Sigma_{Z_{iA}} &\leftarrow \left(\mathbb{E}_{\mathcal{Q}_{/Z_i}}[\sigma^{-2}W_A\Phi_{iA}\Phi_{iA}^T W_A^T + I]\right)^{-1} + \mathcal{O}(\epsilon^2) \\
&\rightarrow \left(\mathbb{E}_{\mathcal{Q}_{/Z_i}}[\sigma^{-2}W_A\Phi_{iA}\Phi_{iA}^T W_A^T + I]\right)^{-1} = [\frac{a_\sigma}{b_\sigma}H_{iA} + I]^{-1}
\end{aligned}
\tag{A74}
$$

Applying block matrix multiplication and Theorem A5 to Equation (26) conditioned on Equation (A37), we obtain

$$
\begin{aligned}
\mu_{iA} &\leftarrow \mathbb{E}_{\mathcal{Q}_{/Z_i}}[\sigma^{-2}(Y_i - \bar{Z}\Phi_{iA})\Phi_{iA}^T W_A^T]\Sigma_{Z_{iA}} + \mathcal{O}(\epsilon) \\
&\rightarrow \frac{a_\sigma}{b_\sigma}(Y_i - \mu_{\bar{Z}A}\Phi_{iA})\Phi_{iA}^T(\mu_{W_A})^T\Sigma_{Z_{iA}}
\end{aligned}
\tag{A75}
$$

$\square$

**Theorem A5.** *$\mathcal{L}$ and $\mathcal{L}^{(a)}$ share the same update rule for $\bar{Z}_A$, i.e.,*

$$
\begin{aligned}
\Sigma_{\bar{Z}A} &\leftarrow \left(\mathbb{E}_{\mathcal{Q}_{/\bar{Z}}}\left[\sum_{i=1}^P \left(\sigma^{-2}\Phi_{iA}\Phi_{iA}^T\right) + \eta\, \text{diag}(\beta_A)\right]\right)^{-1} \\
&= \left(\sum_{i=1}^P \left(\frac{a_\sigma}{b_\sigma}\Phi_{iA}\Phi_{iA}^T\right) + \frac{a_\eta}{b_\eta}\text{diag}(\frac{a_A}{b_A})\right)^{-1}
\end{aligned}
\tag{A76}
$$

$$
\begin{aligned}
\mu_{\bar{Z}A} &\leftarrow \left(\mathbb{E}_{\mathcal{Q}_{/\bar{Z}}}\left[\sigma^{-2}\right]\sum_{i=1}^P (Y - \mathbb{E}_{\mathcal{Q}_{/\bar{Z}}}[Z_{iA}W_A]\Phi_{iA})\Phi_{iA}\right)\Sigma_{\bar{Z}A} \\
&= \left(\frac{a_\sigma}{b_\sigma}\sum_{i=1}^P (Y - \mu_{iA}\mu_{W_A}\Phi_{iA})\Phi_{iA}\right)\Sigma_{\bar{Z}A}
\end{aligned}
\tag{A77}
$$

**Proof.** Applying the block matrix inversion formula to Equation (20) conditioned on $\mathbb{E}_{\mathcal{Q}_{/\bar{Z}}}[\beta_k] = \epsilon^{-1}, \forall k > K^{(\text{a})}$, we have

$$
\begin{aligned}
\Sigma_{\bar{Z}A} &\leftarrow \left( \mathbb{E}_{\mathcal{Q}_{/\bar{Z}}} \left[ \sum_{i=1}^{P} \left( \sigma^{-2} \Phi_{iA} \Phi_{iA}^T \right) + \eta \operatorname{diag}(\beta_A) \right] \right)^{-1} + \mathcal{O}(\epsilon) \\
&\to \left( \mathbb{E}_{\mathcal{Q}_{/\bar{Z}}} \left[ \sum_{i=1}^{P} \left( \sigma^{-2} \Phi_{iA} \Phi_{iA}^T \right) + \eta \operatorname{diag}(\beta_A) \right] \right)^{-1}
\end{aligned}
\tag{A78}
$$

Applying block matrix multiplication and Theorem A5 to Equation (21) conditioned on Equation (A36), we have

$$
\begin{aligned}
\mu_{\bar{Z}A} &\leftarrow \left( \mathbb{E}_{\mathcal{Q}_{/\bar{Z}}} \left[ \sigma^{-2} \right] \sum_{i=1}^{P} (Y - \mathbb{E}_{\mathcal{Q}_{/\bar{Z}}} [Z_{iA} W_A] \Phi_{iA}) \Phi_{iA} \right) \Sigma_{\bar{Z}A} + \mathcal{O}(\epsilon) \\
&\to \left( \mathbb{E}_{\mathcal{Q}_{/\bar{Z}}} \left[ \sigma^{-2} \right] \sum_{i=1}^{P} (Y - \mathbb{E}_{\mathcal{Q}_{/\bar{Z}}} [Z_{iA} W_A] \Phi_{iA}) \Phi_{iA} \right) \Sigma_{\bar{Z}A}
\end{aligned}
\tag{A79}
$$

□

**Theorem A6.** *$\mathcal{L}$ and $\mathcal{L}^{(a)}$ share the same update rule for $W_A$, i.e.,*

$$
\begin{aligned}
\Sigma_{\operatorname{vec}(W)} &\leftarrow \mathbb{E}_{\mathcal{Q}_{/W}} \left[ \sigma^{-2} \sum_{i=1}^{P} \left( (\Phi_{iA}^T \Phi_{iA}) \otimes (Z_{iA}^T Z_{iA}) \right) + \operatorname{diag}(\beta_A) \otimes \operatorname{diag}(\alpha_A) \right]^{-1} \\
&= \left( \frac{a_\sigma}{b_\sigma} \sum_{i=1}^{P} \left( (\Phi_{iA}^T \Phi_{iA}) \otimes (\mu_{iA}^T \mu_{iA} + \Sigma_{Z_i A}) \right) + \operatorname{diag}\left( \frac{a_A}{b_A} \right) \otimes \operatorname{diag}\left( \frac{c_A}{d_A} \right) \right)^{-1}
\end{aligned}
\tag{A80}
$$

$$
\begin{aligned}
\mu_{\operatorname{vec}(W)} &\leftarrow \mathbb{E}_{\mathcal{Q}_{/W}} \left[ -\sigma^{-2} \sum_{i=1}^{P} \operatorname{vec}\left( \left( \Phi_{iA} (\Phi_{iA}^T \bar{Z}_A^T - Y_i^T) Z_{iA} \right)^T \right)^T \right] \Sigma_{\operatorname{vec}(W)A} \\
&= -\frac{a_\sigma}{b_\sigma} \sum_{i=1}^{P} \operatorname{vec}\left( \left( \Phi_{iA} (\Phi_{iA}^T \mu_{\bar{Z}A}^T - Y_i^T) \mu_{iA} \right)^T \right)^T \Sigma_{\operatorname{vec}(W)A}
\end{aligned}
\tag{A81}
$$

**Proof.** Applying the block matrix inversion formula to Equation (22) conditioned on $\mathbb{E}_{\mathcal{Q}_{/\bar{Z}}}[\beta_k] = \epsilon^{-1}, \forall k > K^{(\text{a})}$ and $\mathbb{E}_{\mathcal{Q}_{/\bar{Z}}}[\alpha_j] = \epsilon^{-1}, \forall j > J^{(\text{a})}$, we have

$$
\begin{aligned}
\Sigma_{\operatorname{vec}(W)} &\leftarrow \mathbb{E}_{\mathcal{Q}_{/W}} \left[ \sigma^{-2} \sum_{i=1}^{P} \left( (\Phi_{iA}^T \Phi_{iA}) \otimes (Z_{iA}^T Z_{iA}) \right) + \operatorname{diag}(\beta_A) \otimes \operatorname{diag}(\alpha_A) \right]^{-1} + \mathcal{O}(\epsilon) \\
&\to \mathbb{E}_{\mathcal{Q}_{/W}} \left[ \sigma^{-2} \sum_{i=1}^{P} \left( (\Phi_{iA}^T \Phi_{iA}) \otimes (Z_{iA}^T Z_{iA}) \right) + \operatorname{diag}(\beta_A) \otimes \operatorname{diag}(\alpha_A) \right]^{-1}
\end{aligned}
\tag{A82}
$$

Applying block matrix multiplication and Theorem A5 to Equation (23) conditioned on Equations (A36) and (A37), we obtain

$$
\begin{aligned}
\mu_{\operatorname{vec}(W)} &\leftarrow \mathbb{E}_{\mathcal{Q}_{/W}} \left[ -\sigma^{-2} \sum_{i=1}^{P} \operatorname{vec}\left( \left( \Phi_{iA} (\Phi_{iA}^T \bar{Z}_A^T - Y_i^T) Z_{iA} \right)^T \right)^T \right] \Sigma_{\operatorname{vec}(W)A} + \mathcal{O}(\epsilon) \\
&\to \mathbb{E}_{\mathcal{Q}_{/W}} \left[ -\sigma^{-2} \sum_{i=1}^{P} \operatorname{vec}\left( \left( \Phi_{iA} (\Phi_{iA}^T \bar{Z}_A^T - Y_i^T) Z_{iA} \right)^T \right)^T \right] \Sigma_{\operatorname{vec}(W)A}
\end{aligned}
\tag{A83}
$$

□

**Theorem A7.** $\mathcal{L}$ *and* $\mathcal{L}^{(a)}$ *share the same update rule for* $\sigma$, *i.e.*,

$$a_\sigma \leftarrow a_0 + \frac{1}{2}\sum_i N_i \tag{A84}$$

$$b_\sigma \leftarrow b_0 + \frac{1}{2}\mathbb{E}_{\mathcal{Q}_{/\sigma}}\left[\sum_i ||Y_i - (Z_{iA}W_A + \bar{Z}_A)\Phi_{iA}||_2^2\right]$$

$$= b_0 + \frac{1}{2}\sum_i (Y_i Y_i^T - 2Y_i(\mu_{iA}\mu_{W_A}\Phi_{iA})^T - 2Y_i(\mu_{\bar{Z}A}\Phi_{iA})^T + 2\mu_{iA}\mu_{W_A}\Phi_{iA}\Phi_{iA}^T(\mu_{\bar{Z}A})^T$$

$$+ \text{Tr}\left(\left(\Sigma_{\bar{Z}A} + (\mu_{\bar{Z}A})^T\mu_{\bar{Z}A}\right)\Phi_{iA}\Phi_{iA}^T\right))$$

$$+ \frac{1}{2}\text{vec}(G_A^T)^T \sum_i \text{vec}\left(\text{vec}(\Phi_{iA}\Phi_{iA}^T)\,\text{vec}(\Sigma_{Z_iA} + \mu_{iA}^T\mu_{iA})^T\right), \tag{A85}$$

*where*

$$G_{A(j+kM)} \leftarrow \mathbb{E}_{\mathcal{Q}_{/\sigma}}\left[\text{vec}(W_{Ak}W_{Aj}^T)^T\right]$$

$$= \text{vec}(\Sigma_{[W_{Ak},W_{Aj}]} + \mu_{\text{vec}\,(W)\,[W_{Aj}]}^T\mu_{\text{vec}\,(W)\,[W_{Ak}]})^T, \forall j = 1:K^{(a)}, k = 1:K^{(a)} \tag{A86}$$

**Proof.** Applying block matrix multiplication and Theorem A5 to Equation (28) conditioned on Equations (A36) and (A37), we have

$$b_\sigma \leftarrow b_0 + \frac{1}{2}\mathbb{E}_{\mathcal{Q}_{/\sigma}}\left[\sum_i ||Y_i - (Z_{iA}W_A + \bar{Z}_A)\Phi_{iA}||_2^2\right] + \mathcal{O}(\epsilon)$$

$$\rightarrow b_0 + \frac{1}{2}\mathbb{E}_{\mathcal{Q}_{/\sigma}}\left[\sum_i ||Y_i - (Z_{iA}W_A + \bar{Z}_A)\Phi_{iA}||_2^2\right] \tag{A87}$$

$\square$

We show that $\mathcal{Q}_{Z_{iA}}, \mathcal{Q}_{W_A}, \mathcal{Q}_{\bar{Z}_{iA}}$, and $\mathcal{Q}_\sigma$ share the same update formulas as those derived from the low-dimensional lower bound: $\mathcal{Q}^{(a)}{}_{Z_{iA}}, \mathcal{Q}^{(a)}{}_{W_A}, \mathcal{Q}^{(a)}{}_{\bar{Z}_{iA}}, \mathcal{Q}^{(a)}{}_\sigma$. Thus, in practice, it suffices to update $\mathcal{Q}^{(a)}$; we can then increase $K^{(a)}$ by including new basis functions. This process proves to implicitly maximize $\mathcal{L}$ with $\mathcal{Q}$.

*Appendix C.4. Low-Dimensional Lower Bound*

We now have updating formulas for the parameters in the active subspace. $\mathcal{Q}_{Z_{iA}}$ is updated by Equations (A70)–(A72). $\mathcal{Q}_{W_A}$ is updated by Equations (A80) and (A81). $\mathcal{Q}_{\bar{Z}_A}$ is updated by Equations (A76) and (A77). $\mathcal{Q}_{\alpha_A}, \mathcal{Q}_{\beta_A}, \mathcal{Q}_\eta$ are updated by Theorems A1–A3, with the companion of implicit updates of $\mathcal{Q}_{W_B}, \mathcal{Q}_{W_C}$. $\mathcal{Q}_\sigma$ is updated by Equations (A84)–(A86). All the updating rules are identical to those derived from the low-dimensional lower bound $\mathcal{L}^{(a)}$ with $K^{(a)}$ basis functions. Therefore, in practice, all we need is to optimize $\mathcal{L}^{(a)}$, with time complexity of $\mathcal{O}\left(K^{(a)^2}\max\left(K^{(a)^4}, P\max_i(N_i)\right)\right)$, as described in Theorem A8, and then check if a new basis function should be included in the model.

For numerical stability, we scale $\phi, b$ such that $\min_k(\mathbb{E}_{Q_\beta}[\beta_k]) = \min_k(\frac{c_k}{d_k}) = 1$ at the beginning of Algorithm A1.

**Theorem A8.** *The lower bound* $\mathcal{L}$ *can be optimized using Algorithm A1 with time complexity of* $\mathcal{O}\left(K^{(a)^2}\max\left(K^{(a)^4}, P\max_i(N_i)\right)\right)$.

**Proof.** The proof is a consequence of Theorems A1–A7. $\square$

---

**Algorithm A1** Variational inference

---

**Require:** $\mu_{Z_i}, \Sigma_{Z_i}, \mu_{\text{vec}(W)}, \Sigma_{\text{vec}(W)}, \mu_{\bar{Z}}, \Sigma_{\bar{Z}}, a_\sigma, b_\sigma, a_{\alpha_j}, b_{\alpha_j}, a_{\beta_k}, b_{\beta_k}, \forall i, j, k$     ▷ Multisample
   RVM
  **while** True **do**
     $\mathcal{L}^{(a)} \leftarrow \text{lowerbound}(\mathcal{Q}^{(a)})$
     Update $\mathcal{Q}^{(a)}$ with respect to all parameters using mean field approximation
     **if** $\text{lowerbound}(\mathcal{Q}^{(a)}) - \mathcal{L}^{(a)} < \tau_{\text{con}}$ **then**     ▷ Insignificant increase
       Search for new basis functions using Algorithm 1
       **if** not found **then**     ▷ Converged
         break
       **end if**
     **end if**
     Remove dimensions associated with the precision of the maximum values
  **end while**
  Get rid of dimensions associated with $\alpha_j \geq \min_j(\alpha_j)\tau_{\text{eff}}$ and $\beta_k \geq \min_k(\beta_k)\tau_{\text{eff}}$

---

## Appendix D. Scalable Update for BSFDA$^{\text{Fast}}$

For brevity, we denote the covariance of $\zeta_i$ as $S$, i.e., $\zeta_i \sim \mathcal{N}(0, S)$. $S$ is diagonal and $S_{kk} = \varsigma_k^2 \beta_k^{-1}$. The variational update formulas are as follows:

$$\Sigma_{\theta_i} \leftarrow \mathbb{E}_{\mathcal{Q}/\theta_i}\left[\Phi_i \Phi_i^T \sigma^{-2} + S^{-1}\right]^{-1} \tag{A88}$$

$$\mu_{\theta_i} \leftarrow \mathbb{E}_{\mathcal{Q}/\theta_i}\left[\left((Y_i - \bar{Z}\Phi_i)\Phi_i^T \sigma^{-2} + Z_i W S^{-1}\right)\right]\Sigma_{\theta_i} \tag{A89}$$

$$\Sigma_{Z_i} \leftarrow \mathbb{E}_{\mathcal{Q}/Z_i}\left[W S^{-1} W^T + I\right]^{-1} \tag{A90}$$

$$\mu_{Z_i} \leftarrow \mathbb{E}_{\mathcal{Q}/Z_i}\left[\theta_i S^{-1} W^T\right]\Sigma_{Z_i} \tag{A91}$$

$$a_{\varsigma_k} \leftarrow a_0 + \frac{P}{2} \tag{A92}$$

$$b_{\varsigma_k} \leftarrow \mathbb{E}_{\mathcal{Q}/\varsigma_k}\left[b_0 + \frac{1}{2}\sum_i (\theta_{ik} - Z_i W_{\cdot k})^2 \beta_k\right] \tag{A93}$$

$$\Sigma_{W_{\cdot k}} \leftarrow \mathbb{E}_{\mathcal{Q}/W_{\cdot k}}\left[\varsigma_k^{-2}\beta_k \sum_i Z_i^T Z_i + \beta_k \,\text{diag}(\alpha)\right]^{-1} \tag{A94}$$

$$\mu_{W_{\cdot k}} \leftarrow \mathbb{E}_{\mathcal{Q}/W_{\cdot k}}\left[\varsigma_k^{-2}\beta_k \sum_i (\theta_{ik} Z_i)\right]\Sigma_{W_{\cdot k}} \tag{A95}$$

$$a_{\beta_k} \leftarrow a_0 + \frac{1 + K + P}{2} \tag{A96}$$

$$b_{\beta_k} \leftarrow \mathbb{E}_{\mathcal{Q}/\beta_k}\left[b_0 + \frac{1}{2}\left[\bar{Z}_k^2 \eta + \sum_j (W_{jk}^2 \alpha_j) + \sum_i (\theta_{ik} - Z_i W_{\cdot k})^2 \varsigma_k^{-2}\right]\right] \tag{A97}$$

$$\Sigma_{\bar{Z}} \leftarrow \mathbb{E}_{\mathcal{Q}/\bar{Z}}\left[\sigma^{-2}\sum_i (\Phi_i \Phi_i^T) + \eta \,\text{diag}(\beta)\right]^{-1} \tag{A98}$$

$$\mu_{\bar{Z}} \leftarrow \mathbb{E}_{\mathcal{Q}/\bar{Z}}\left[\sigma^{-2}\sum_i\left[(Y_i - \theta_i \Phi_i)\Phi_i^T\right]\right]\Sigma_{\bar{Z}} \tag{A99}$$

$$a_{\sigma^{-2}} \leftarrow a_0 + \frac{1}{2}\sum_i N_i \tag{A100}$$

$$b_{\sigma^{-2}} \leftarrow \mathbb{E}_{\mathcal{Q}/\sigma}\left[b_0 + \frac{1}{2}\sum_i ||Y_i - (\bar{Z} + \theta_i)\Phi_i||_2^2\right] \tag{A101}$$

Notably, the columns of $W$ become conditionally independent of the introduction of the slack variable $\theta$, akin to the strategy described in [18,28]. Then, the surrogate posterior of $W$ factorizes over the columns, thereby requiring the calculation of the covariance of each column separately, instead of the entire $W$ at once. Thus, the computational complexity is significantly reduced. This factorization is introduced on top of the existing factorizations; thus, the low-dimensional optimization strategy of BSFDA also applies to BSFDA$^{\text{Fast}}$.

## Appendix E. Fast Initialization

In order to efficiently obtain a good initialization for the unknowns to be estimated, e.g., $Z, \bar{Z}, \beta$ and $\sigma$, we approximate the model so that we can adopt a fast strategy maximizing the marginal likelihood using direct differentiation, which is similar to [39]. This initial $\beta$ serves to select the $K^{(a)}$ basis functions to start with.

We introduce $\tilde{Z}$ for easier marginalization:

$$Y_i = \tilde{Z}_i \Phi_i + E_i \tag{A102}$$

$$\tilde{Z}_{ik} = \frac{Z_{ik}}{\sqrt{\beta_k}} + \bar{Z}_k \sim \mathcal{N}(\bar{Z}_k, \beta_k^{-1}) \tag{A103}$$

$$\bar{Z}_k \sim \mathcal{N}(0, \beta_k^{-1}) \tag{A104}$$

$$\beta_k \sim \Gamma(\beta_k | a_0, b_0), \sigma^{-2} \sim \Gamma(\sigma^{-2} | a_0, b_0) \tag{A105}$$

$$E_i \sim \mathcal{N}(0, \sigma^2 I) \tag{A106}$$

The approximated probabilistic graphical model is shown in Figure A1.

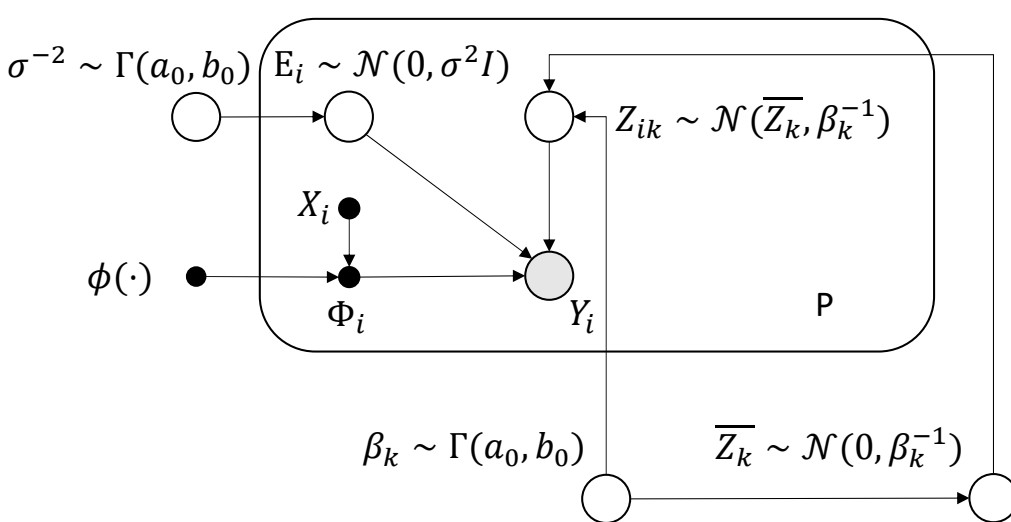

**Figure A1.** Probabilistic graphical model for the simplified model.

*Appendix E.1. Maximum Likelihood Estimation*

We apply maximum likelihood estimation for point estimates of $\bar{Z}, \beta, \sigma$.

$$\bar{Z}^*, \beta^*, \sigma^* \leftarrow \arg\min_{\bar{Z}, \beta, \sigma} \mathcal{P}, \tag{A107}$$

where $\mathcal{P} = -\ln \Pr[Y | \bar{Z}, \beta, \sigma]$. Conditioned on these estimates, we can calculate the expectation of $Z$.

**Optimization of $\beta, \bar{Z}$**

We set the differentiation to zero, i.e., $\frac{\partial \mathcal{P}}{\partial \beta_k} = 0$, and obtain

$$\beta_k \leftarrow \begin{cases} \theta_k, \text{if } \theta_k > 0 \\ \infty, \text{otherwise} \end{cases} \tag{A108}$$

where

$$\theta_k = \left( \frac{\sum_{i=1}^{P} s_{ik}^2}{\sum_{i=1}^{P} (q_{ik}^2 - s_{ik})} \right) \tag{A109}$$

$$q_{ik} = \Phi_{ik} \mathcal{C}_{i/k}^{-1} (Y - \bar{Z}\Phi_i)^T \tag{A110}$$

$$s_{ik} = \Phi_{ik} \mathcal{C}_{i/k}^{-1} \Phi_{ik}^T \tag{A111}$$

$$\mathcal{C}_{i/k} = \mathcal{C}_i - \Phi_{ik}^T \beta_k^{-1} \Phi_{ik} \tag{A112}$$

$$\mathcal{C}_i = \Phi_i^T \operatorname{diag}(\beta^{-1})\Phi_i + \sigma^2 I = \sum_{k=1}^{K} \Phi_{ik}^T \beta_k^{-1} \Phi_{ik} + \sigma^2 I \tag{A113}$$

We differentiate $\mathcal{P}$ with respect to $\bar{Z}$ and zero the derivative, i.e., $\frac{\partial \mathcal{P}}{\bar{Z}} = 0$, to obtain

$$\bar{Z} \leftarrow \sum_{i=1}^{P} \left( Y_i \mathcal{C}_i^{-1} \Phi_i^T \right) \left( \sum_{i=1}^{P} (\Phi_i \mathcal{C}_i^{-1} \Phi_i^T) \right)^{-1} \tag{A114}$$

We approximate Equation (A114) by $\bar{Z}_A \leftarrow \sum_{i=1}^{P} \left( Y_i \mathcal{C}_i^{-1} \Phi_{iA}^T \right) \left( \sum_{i=1}^{P} (\Phi_{iA} \mathcal{C}_i^{-1} \Phi_{iA}^T) \right)^{-1}$ and $\bar{Z}_B \leftarrow 0$. This way, we can apply the update with only the active basis functions.

**Optimization of $\sigma$:**

We use EM to optimize $\sigma$. In the E-step,

$$\mathbb{E}_{\mathcal{Q}_{\bar{Z}}}[\tilde{Z}_i] \leftarrow \sigma^{-2} (Y_i - \bar{Z}\Phi_i)\Phi_i^T \mathcal{S}_i \tag{A115}$$

$$\mathbb{E}_{\mathcal{Q}_{\bar{Z}}}[\tilde{Z}_i^T \tilde{Z}_i] \leftarrow \mathcal{S}_i + \mathbb{E}_{\mathcal{Q}_{\bar{Z}}}[\tilde{Z}_i]^T \mathbb{E}_{\mathcal{Q}_{\bar{Z}}}[\tilde{Z}_i], \tag{A116}$$

where $\mathcal{S}_i = (\Psi_i \sigma^{-2} + \operatorname{diag}(\beta))^{-1}$.

In the M-step,

$$\begin{aligned} \sigma^{-2} &\leftarrow \frac{\sum_{i=1}^{P} \mathbb{E}_{\mathcal{Q}_{\bar{Z}}} \left[ ||Y_i - (\tilde{Z}_i + \bar{Z})\Phi_i||_2^2 \right]}{\sum_{i=1}^{P} N_i} \\ &= \frac{\sum_{i=1}^{P} (Y_i - \bar{Z}\Phi_i)(Y_i - \bar{Z}\Phi_i - 2\Phi_i^T \mathbb{E}_{\mathcal{Q}_{\bar{Z}}}[\tilde{Z}_i]^T)^T + \operatorname{Tr}(\mathbb{E}_{\mathcal{Q}_{\bar{Z}}}[\tilde{Z}_i^T \tilde{Z}_i]\Psi_i)}{\sum_{i=1}^{P} N_i} \end{aligned} \tag{A117}$$

The optimization iterates between the E-step Equations (A115) and (A116) and the M-step Equation (A117).

In practice, we need only $\mathbb{E}_{\mathcal{Q}_{\bar{Z}}}[\tilde{Z}_{iA}], \mathbb{E}_{\mathcal{Q}_{\bar{Z}}}[\tilde{Z}_{iA}^T \tilde{Z}_{iA}]$, and $\mathcal{S}_{iA}$, and they can be calculated using the $K^{(a)}$ active basis functions. Thus, similarly to [39], all computations can be operated with only the active basis functions and thus it is computationally efficient. This is described in Algorithm A2.

$$\mathcal{P} = -\ln \Pr[Y|\bar{Z}, \beta, \sigma] = -\sum_{i=1}^{P} \ln \Pr[Y_i|\bar{Z}, \beta, \sigma] = \sum_{i=1}^{P} \mathcal{P}_i \tag{A118}$$

$$\mathcal{P}_i = \int \Pr[Y_i|\tilde{Z}_i, \bar{Z}, \beta, \sigma] \Pr[\tilde{Z}_i|\bar{Z}, \beta] d\tilde{Z}_i = \mathbb{E}_{\tilde{Z}_i \sim \mathcal{N}(\bar{Z}, \beta)}[\Pr[Y_i|\tilde{Z}_i, \sigma]] = \mathcal{N}(Y_i|\bar{Z}\Phi_i, \mathcal{C}_i) \tag{A119}$$

$$\Pr[Y_i|\tilde{Z}_i, \bar{Z}, \beta, \sigma] = \mathcal{N}(Y_i|(\tilde{Z}_i + \bar{Z})\Phi_i, \sigma^2 I) \tag{A120}$$

---

**Algorithm A2** Multisample relevance vector machine

---

**while** $\mathcal{P}$ is not converged **do**

 $k \leftarrow$ a random number that satisfies $\mathrm{CosSim}(\phi_k, \phi_A) \leq \tau_{\mathrm{sim}}$     $\triangleright\ \mathcal{O}(K^{(\mathrm{a})^3})$

 $s_{ik} \leftarrow \Phi_{ik}\mathcal{C}_{i/k}^{-1}\Phi_{ik}^T, \forall i$    $\triangleright$ Sparsity factor. $\mathcal{O}\left(P\max(K^{(\mathrm{a})^3}, \max_i(N_i)^2)\right)$

 $q_{ik} \leftarrow \Phi_{ik}\mathcal{C}_{i/k}^{-1}(Y - \bar{Z}_A\Phi_{iA})^T, \forall i$       $\triangleright$ Quality factor.

$\mathcal{O}\left(P\max_i(N_i)\max(K^{(\mathrm{a})}, \max_i(N_i))\right)$

 $\theta_k \leftarrow \left(\frac{\sum_{i=1}^P s_{ik}^2}{\sum_{i=1}^P (q_{ik}^2 - s_{ik})}\right)$

 **if** $\theta > 0$ **then**

  $\beta_k \leftarrow \theta_k$             $\triangleright$ Precision is finite

 **else**

  $\beta_k \leftarrow \infty$      $\triangleright$ Precision is infinite and the dimension is removed

 **end if**

 $\Phi_{iA} \leftarrow$ All $\Phi_{ik}$ that has $\beta_k < \infty, \forall i$

 $\mathcal{C}_i = \sum_{\beta_k < \infty}\Phi_{ik}^T\beta_k^{-1}\Phi_{ik} + \sigma^2 I, \forall i$

 $\bar{Z}_A \leftarrow \sum_{i=1}^P\left(Y_i\mathcal{C}_i^{-1}\Phi_{iA}^T\right)\left(\sum_{i=1}^P(\Phi_{iA}\mathcal{C}_i^{-1}\Phi_{iA}^T)\right)^{-1}$         $\triangleright$

$\mathcal{O}\left(PK^{(\mathrm{a})}\max\left(K^{(\mathrm{a})}, \max_i(N_i)\right)^2\right)$

 $\mathcal{S}_{iA} \leftarrow (\Phi_{iA}\Phi_{iA}^T\sigma^{-2} + \mathrm{diag}(\beta_A))^{-1}, \forall i$

 $\mathbb{E}_{\mathcal{Q}_{\tilde{Z}}}[\tilde{Z}_{iA}] \leftarrow \sigma^{-2}(Y_i - \bar{Z}_A\Phi_{iA})\Phi_{iA}^T\mathcal{S}_{iA}, \forall i$    $\triangleright\ \mathcal{O}(PK^{(\mathrm{a})^2}\max_i(N_i))$

 $\mathbb{E}_{\mathcal{Q}_{\tilde{Z}}}[\tilde{Z}_{iA}^T\tilde{Z}_{iA}] \leftarrow \mathcal{S}_{iA} + \mathbb{E}_{\mathcal{Q}_{\tilde{Z}}}[\tilde{Z}_{iA}]^T\mathbb{E}_{\mathcal{Q}_{\tilde{Z}}}[\tilde{Z}_{iA}], \forall i$

 $\sigma \leftarrow \frac{\sum_{i=1}^P(Y_i - \bar{Z}_A\Phi_{iA})(Y_i - \bar{Z}_A\Phi_{iA} - 2\Phi_{iA}^T\mathbb{E}_{\mathcal{Q}_{\tilde{Z}}}[\tilde{Z}_{iA}]^T)^T + \mathrm{Tr}(\mathbb{E}_{\mathcal{Q}_{\tilde{Z}}}[\tilde{Z}_{iA}^T\tilde{Z}_{iA}]\Phi_{iA}\Phi_{iA}^T)}{\sum_{i=1}^P N_i}$

**end while**

---

We apply Sylvester's determinant theorem to Equation (A113) and obtain

$$|\mathcal{C}_i| = |\mathcal{C}_{i/k}||I + \beta_k^{-1}\Phi_{ik}^T\mathcal{C}_{i/k}^{-1}\Phi_{ik}| \tag{A121}$$

We apply the Woodbury matrix identity to Equation (A113) and obtain

$$\mathcal{C}_i^{-1} = \mathcal{C}_{i/k}^{-1} - \mathcal{C}_{i/k}^{-1}\Phi_{ik}^T(\beta_k + \Phi_{ik}\mathcal{C}_{i/k}^{-1}\Phi_{ik}^T)^{-1}\Phi_{ik}\mathcal{C}_{i/k}^{-1} \tag{A122}$$

We first expand $\mathcal{P}_i$

$$\begin{aligned}
\mathcal{P}_i &= \ln\Pr[Y_i|\bar{Z}, \sigma, \beta] \\
&= -\frac{1}{2}\sum_i \ln|2\pi\mathcal{C}_i| + (Y_i - \bar{Z}\Phi_i)\mathcal{C}_i^{-1}(Y_i - \bar{Z}\Phi_i)^T \\
&= -\frac{1}{2}\Big(N_i\ln(2\pi) + \ln|\mathcal{C}_{i/k}| + \ln|I + \beta_k^{-1}\Phi_{ik}\mathcal{C}_{i/k}^{-1}\Phi_{ik}^T| + (Y_i - \bar{Z}\Phi_i)\mathcal{C}_i^{-1}(Y_i - \bar{Z}\Phi_i)^T \\
&\quad - (\beta_k + \Phi_{ik}\mathcal{C}_{i/k}^{-1}\Phi_{ik}^T)^{-1}||\Phi_{ik}\mathcal{C}_{i/k}^{-1}(Y - \bar{Z}\Phi_i)^T||_2^2\Big) \\
&= \mathcal{P}_{i/k} + \frac{1}{2}\Big(\ln\beta_k - \ln|\beta_k + s_{ik}| + \frac{q_{ik}^2}{\beta_k + sik}\Big) \tag{A123}
\end{aligned}$$

where we plug in Equations (A121) and (A122) and define $q_{ik}, s_{ik}$ in a similar way to [39]. The sparsity factor $s_{ik}$ can be seen to be a measure of the extent to which the basis function $\phi_k$ overlaps those already present in the model under the measurements at index set $X_i$. The quality factor $q_{ik}$ is a measure of the alignment with the error of the model at $X_i$ with this basis function excluded. Because we are representing the mean functions using only the active basis functions, i.e., $\bar{Z}_k = 0$ when $\beta_k = \infty$, Equation (A110) uses only the $K$ active basis functions. Similarly, Equation (A111) only uses the $K$ active basis functions.

For computational efficiency, we can compute $s_{ik}, q_{ik}$ using $S_{ik} = \Phi_{ik}\mathcal{C}_i^{-1}\Phi_{ik}^T$, $Q_{ik} = \Phi_{ik}\mathcal{C}_i^{-1}(Y - \bar{Z}\Phi_i)^T$ in a similar way to [39] as follows:

$$s_{ik} = \Phi_{ik}\mathcal{C}_{i/k}^{-1}\Phi_{ik}^T = S_{ik} + \Phi_{ik}\mathcal{C}_{i/k}^{-1}\Phi_{ik}^T(\beta_k + \Phi_{ik}\mathcal{C}_{i/k}^{-1}\Phi_{ik}^T)^{-1}\Phi_{ik}\mathcal{C}_{i/k}^{-1}\Phi_{ik}^T$$

$$= S_{ik} + s_{ik}(\beta_k + s_{ik})^{-1}s_{ik} \rightleftharpoons s_{ik} = \frac{\beta_k + s_{ik}}{\beta_k}S_{ik} \tag{A124}$$

$$\Rightarrow s_{ik} \leftarrow (1 - \frac{1}{\beta_k}S_{ik})^{-1}S_{ik} = \frac{\beta_k S_{ik}}{\beta_k - S_{ik}} \tag{A125}$$

$$q_{ik} = \Phi_{ik}\mathcal{C}_{i/k}^{-1}(Y - \bar{Z}\Phi_i)^T$$

$$= Q_{ik} + \Phi_{ik}\mathcal{C}_{i/k}^{-1}\Phi_{ik}^T(\beta_k + \Phi_{ik}\mathcal{C}_{i/k}^{-1}\Phi_{ik}^T)^{-1}\Phi_{ik}\mathcal{C}_{i/k}^{-1}(Y - \bar{Z}\Phi_i)^T$$

$$= Q_{ik} + s_{ik}(\beta_k + s_{ik})^{-1}q_{ik} \tag{A126}$$

$$\Rightarrow q_{ik} \leftarrow \frac{\beta_k + s_{ik}}{\beta_k}Q_{ik} = \frac{\beta_k Q_{ik}}{\beta_k - S_{ik}} \tag{A127}$$

*Appendix E.2. Optimization of $\beta$, $\bar{Z}$*

**Derivation of Equation** (A108)

We differentiate $\mathcal{P}$ with respect to $\beta_k$

$$\frac{\partial \mathcal{P}}{\partial \beta_k} = \sum_{i=1}^{P} \frac{1}{2}\left(\beta_k^{-1} - |\beta_k + s_{ik}|^{-1} - q_{ik}^2(\beta_k + s_{ik})^2\right)$$

$$= \frac{1}{2}\beta_k^{-1}\sum_{i=1}^{P}\left((\beta_k + s_{ik})^{-2}(\beta_k(s_{ik} - q_{ik}^2) + s_{ik}^2)\right) \tag{A128}$$

We further adopt the approximation $s_{1k} \approx s_{2k} \approx \ldots \approx s_{Pk}$. Because $s_{ik}$ is a discrete measure of the overlapping between the basis functions, it should remain invariant with respect to different sampling grids $X_i$ given that the number of measurements is adequate and similar. Alternatively, the expectation maximization scheme can also be applied and is guaranteed to increase the likelihood $\mathcal{P}$ in each iteration until convergence. However, we opt for this gradient descent with approximations for its advantage in speed to obtain a reasonable initialization. This way, we set the approximated differentiation to zero:

$$\frac{\partial \mathcal{P}}{\partial \beta_k} \approx \frac{1}{2}\beta_k^{-1}(\beta_k + s_{1k})^{-2}\sum_{i=1}^{P}\left((\beta_k(s_{ik} - q_{ik}^2) + s_{ik}^2)\right) = 0 \tag{A129}$$

$$\Rightarrow \beta_k \leftarrow \theta_k = \left(\frac{\sum_{i=1}^{P} s_{ik}^2}{\sum_{i=1}^{P}(q_{ik}^2 - s_{ik})}\right) \tag{A130}$$

Because $\beta_k$ is a scale parameter, we need $\beta_k > 0$. Consequently, the optimal value for $\beta_k$ to maximize $\mathcal{P}$ depends on the sign of $\theta_k$. When $\theta_k > 0$, the maximum of $\mathcal{P}$ is achieved at $\beta_k = \theta_k$.

On the other hand, when $\theta_k \leq 0$, $\mathcal{P}$ is monotonically increasing with respect to $\beta_k$, we should have $\beta_k \leftarrow \infty$ in order to maximize $\mathcal{P}$.

More intuitively, Equation (A130) can be regarded as a weighted summation of the estimation of $\beta_k$ using each individual sample function, and it automatically assigns more weight to those with more measurements. Therefore, this optimization strategy is supposed to provide reasonable estimates even when the sampled functions have different numbers of measurements.

**Derivation of Equation** (A114)

We differentiate $\mathcal{P}$ with respect to $\bar{Z}$ and zero the derivative to obtain

$$\frac{\partial \mathcal{P}}{\bar{Z}} = -\frac{1}{2} \sum_{i=1}^{P} \left( -2Y_i \mathcal{C}_i \Phi_i^T + 2\bar{Z}\Phi_i \mathcal{C}_i^{-1}\Phi_T \right) = 0 \tag{A131}$$

$$\Rightarrow \bar{Z} \leftarrow \sum_{i=1}^{P} \left( Y_i \mathcal{C}_i^{-1}\Phi_i^T \right) \left( \sum_{i=1}^{P} (\Phi_i \mathcal{C}_i^{-1}\Phi_i^T) \right)^{-1} \tag{A132}$$

*Appendix E.3. Optimization of $\sigma$*

**Derivation of Equations** (A115) **and** (A116)

We use the expectation maximization strategy with latent variables $\tilde{Z}_i$. It is similar to that used in [30]. It introduces a surrogate function, the log likelihood for the complete data $\mathbb{E}_{\mathcal{Q}_{\tilde{Z}}}[\mathcal{P}_{\tilde{Z}}]$, which is easier to optimize; moreover, in theory, the process ultimately maximizes $\mathcal{P}$.

For the E-step, we calculate the posterior of $\tilde{Z}_i$.

$$\ln \Pr[\tilde{Z}_i|Y_i, \bar{Z}, \sigma, \beta] = \ln \frac{\Pr[Y_i|\tilde{Z}_i, \bar{Z}, \sigma, \beta] \Pr[\tilde{Z}_i|\beta]}{\Pr[Y_i|\bar{Z}, \sigma, \beta]} \tag{A133}$$

$$\propto -\frac{1}{2} \left( \tilde{Z}_i(\Psi_i \sigma^{-2} + \text{diag}(\beta))\tilde{Z}_i^T - 2\sigma^{-2}(Y_I - \bar{Z}\Phi_i)\Phi_i^T\tilde{Z}_i^T \right) \tag{A134}$$

Therefore,

$$\mathbb{E}_{\mathcal{Q}_{\tilde{Z}}}[\tilde{Z}_i] \leftarrow \sigma^{-2}(Y_i - \bar{Z}\Phi_i)\Phi_i^T \mathcal{S}_i \tag{A135}$$

$$\mathbb{E}_{\mathcal{Q}_{\tilde{Z}}}[\tilde{Z}_i^T\tilde{Z}_i] \leftarrow \mathcal{S}_i + \mathbb{E}_{\mathcal{Q}_{\tilde{Z}}}[\tilde{Z}_i]^T \mathbb{E}_{\mathcal{Q}_{\tilde{Z}}}[\tilde{Z}_i] \tag{A136}$$

where

$$\mathcal{S}_i = (\Psi_i \sigma^{-2} + \text{diag}(\beta))^{-1} \tag{A137}$$

**Derivation of Equation** (A117)

In the M-step, we need to maximum $\mathbb{E}_{\mathcal{Q}_{\tilde{Z}}}[\mathcal{P}_{\tilde{Z}}]$ conditioned on $\mathcal{Q}_{\tilde{Z}}$ with respect to $\sigma^{-2}$,

$$\mathcal{P}_{\tilde{Z}} = \sum_{i=1}^{P} \ln \Pr[Y_i, \tilde{Z}_i|\bar{Z}, \sigma, \beta] = \sum_{i=1}^{P} \ln(\Pr[Y_i|\tilde{Z}_i, \bar{Z}, \sigma] \Pr[\tilde{Z}_i|\beta])$$

$$= -\frac{1}{2} \sum_{i=1}^{P} \left( N_i \ln(2\pi\sigma^{-2}) + \sigma^{-2}||Y_i - (\tilde{Z}_i + \bar{Z})\Phi_i||_2^2 + \right.$$

$$\left. \sum_{k=1}^{K} \ln(2\pi\beta_k^{-1}) + \text{Tr}(\tilde{Z}_i \text{diag}(\beta)\tilde{Z}_i^T) \right) \tag{A138}$$

We differentiate $\mathbb{E}_{\mathcal{Q}_{\tilde{Z}}}[\mathcal{P}_{\tilde{Z}}]$ with respect to $\sigma^{-2}$ and set to 0

$$\frac{\partial \mathbb{E}_{\mathcal{Q}_{\tilde{Z}}}[\mathcal{P}_{\tilde{Z}}]}{\partial \sigma^{-2}} = \mathbb{E}_{\mathcal{Q}_{\tilde{Z}}} \left[ -\frac{1}{2} \sum_{i=1}^{P} \left( N_i \sigma^{-2} - \sigma^{-4}||Y_i - (\tilde{Z}_i + \bar{Z})\Phi_i||_2^2 \right) \right] = 0 \tag{A139}$$

$$\Rightarrow \sigma^{-2} \leftarrow \frac{\sum_{i=1}^{P} \mathbb{E}_{\mathcal{Q}_{\tilde{Z}}} \left[ ||Y_i - (\tilde{Z}_i + \bar{Z})\Phi_i||_2^2 \right]}{\sum_{i=1}^{P} N_i}$$

$$= \frac{\sum_{i=1}^{P}(Y_i - \bar{Z}\Phi_i)(Y_i - \bar{Z}\Phi_i - 2\Phi_i^T \mathbb{E}_{\mathcal{Q}_{\tilde{Z}}}[\tilde{Z}_i]^T)^T + \text{Tr}(\mathbb{E}_{\mathcal{Q}_{\tilde{Z}}}[\tilde{Z}_i^T\tilde{Z}_i]\Psi_i)}{\sum_{i=1}^{P} N_i} \tag{A140}$$

# Appendix F. Experiments

*Appendix F.1. Benchmark Simulation*

Figure A2 presents the application of the proposed BSFDA to the simulation benchmark (Scenario 1) outlined in [21]. Even though prior analyses have utilized this benchmark, the current experimental configuration is specifically adapted to highlight the method's capacity for uncertainty quantification. The experimental design consists of 20 functional observations, each sampled at either three points (with a 20% probability) or 10 points (with an 80% probability), determined via random assignment. The number of sampled functions is decreased from 200 to 20 to underscore the effect and estimation of uncertainties. The actual white noise standard deviation is 0.4472, whereas the estimated standard deviation is 0.4839. The component number is also correctly estimated as 3. The figure depicts the true underlying function, the discrete observational data, and the corresponding functional estimates, accompanied by their respective 95% truncated uncertainty intervals.

Notably, the uncertainty associated with sparsely sampled functions exhibits substantial inflation in regions devoid of observations. In contrast, in sampled regions, the uncertainty aligns closely with that of densely sampled functions, approximating twice the standard deviation of the white noise. Additionally, the uncertainty bounds for the estimated mean function are presented, demonstrating reduced variability relative to individual function estimates.

**Table A3.** Distributions of the estimated component number $\hat{r}$ for Scenario 1 (r = 3).

| $N_i$ | $\hat{r}$ | $\text{AIC}_{\text{PACE}}$ | AIC | BIC | $\text{PC}_{p1}$ | $\text{IC}_{p1}$ | $\text{AIC}_{\text{PACE}}^{2022}$ | $\text{BIC}_{\text{PACE}}^{2022}$ | fpca | BSFDA | $\text{BSFDA}^{\text{Fast}}$ |
|---|---|---|---|---|---|---|---|---|---|---|---|
| 5 | ≤1 | 0.000 | 0.000 | 0.155 | 0.005 | 0.000 | 0.000 | 0.000 | 0.000 | 0.000 | 0.000 |
| | =2 | 0.008 | 0.405 | 0.335 | 0.565 | 0.215 | 0.000 | 0.000 | 0.000 | 0.000 | 0.985 |
| | =3 | 0.000 | 0.580 | 0.380 | 0.410 | 0.735 | 0.650 | 0.880 | 0.645 | **0.995** | 0.015 |
| | =4 | 0.121 | 0.010 | 0.115 | 0.010 | 0.045 | 0.335 | 0.120 | 0.235 | 0.005 | 0.000 |
| | ≥5 | 0.870 | 0.005 | 0.015 | 0.010 | 0.005 | 0.015 | 0.000 | 0.120 | 0.000 | 0.000 |
| 10 | ≤1 | 0.000 | 0.000 | 0.000 | 0.000 | 0.000 | 0.000 | 0.000 | 0.000 | 0.000 | 0.000 |
| | =2 | 0.000 | 0.005 | 0.040 | 0.040 | 0.005 | 0.000 | 0.000 | 0.000 | 0.000 | 0.075 |
| | =3 | 0.000 | 0.980 | 0.670 | 0.955 | 0.985 | 0.880 | 0.920 | 0.645 | **1.000** | 0.910 |
| | =4 | 0.000 | 0.015 | 0.255 | 0.000 | 0.010 | 0.120 | 0.080 | 0.235 | 0.000 | 0.015 |
| | ≥5 | 1.000 | 0.000 | 0.035 | 0.005 | 0.000 | 0.000 | 0.000 | 0.120 | 0.000 | 0.000 |
| 50 | ≤1 | 0.000 | 0.000 | 0.000 | 0.000 | 0.000 | 0.000 | 0.000 | 0.000 | 0.000 | 0.000 |
| | =2 | 0.000 | 0.000 | 0.000 | 0.000 | 0.000 | 0.000 | 0.000 | 0.000 | 0.000 | 0.000 |
| | =3 | 0.000 | **1.000** | 0.830 | **1.000** | **1.000** | **1.000** | **1.000** | 0.890 | 0.980 | 0.945 |
| | =4 | 0.000 | 0.000 | 0.150 | 0.000 | 0.000 | 0.000 | 0.000 | 0.060 | 0.020 | 0.050 |
| | ≥5 | 1.000 | 0.000 | 0.020 | 0.000 | 0.000 | 0.000 | 0.000 | 0.050 | 0.000 | 0.005 |

**Table A4.** Distributions of the estimated component number $\hat{r}$ for Scenario 2 (r = 3).

| $N_i$ | $\hat{r}$ | $\text{AIC}_{\text{PACE}}$ | AIC | BIC | $\text{PC}_{p1}$ | $\text{IC}_{p1}$ | $\text{AIC}_{\text{PACE}}^{2022}$ | $\text{BIC}_{\text{PACE}}^{2022}$ | fpca | BSFDA | $\text{BSFDA}^{\text{Fast}}$ |
|---|---|---|---|---|---|---|---|---|---|---|---|
| 5 | ≤1 | 0.000 | 0.000 | 0.230 | 0.000 | 0.000 | 0.000 | 0.000 | 0.000 | 0.000 | 0.000 |
| | =2 | 0.000 | 0.205 | 0.395 | 0.000 | 0.140 | 0.050 | 0.075 | 0.000 | 0.000 | 0.960 |
| | =3 | 0.005 | 0.630 | 0.245 | 0.375 | 0.605 | 0.570 | 0.620 | 0.475 | **1.000** | 0.040 |
| | =4 | 0.125 | 0.155 | 0.110 | 0.440 | 0.210 | 0.345 | 0.275 | 0.350 | 0.000 | 0.000 |
| | ≥5 | 0.870 | 0.010 | 0.020 | 0.185 | 0.045 | 0.035 | 0.030 | 0.175 | 0.000 | 0.000 |
| 10 | ≤1 | 0.000 | 0.000 | 0.000 | 0.000 | 0.000 | 0.000 | 0.000 | 0.000 | 0.000 | 0.000 |
| | =2 | 0.000 | 0.000 | 0.170 | 0.000 | 0.000 | 0.000 | 0.000 | 0.000 | 0.000 | 0.000 |
| | =3 | 0.000 | 0.710 | 0.665 | 0.570 | 0.805 | 0.825 | 0.850 | 0.640 | **1.000** | 0.995 |
| | =4 | 0.005 | 0.260 | 0.135 | 0.355 | 0.185 | 0.175 | 0.150 | 0.235 | 0.000 | 0.005 |
| | ≥5 | 0.995 | 0.030 | 0.030 | 0.075 | 0.010 | 0.000 | 0.000 | 0.125 | 0.000 | 0.000 |
| 50 | ≤1 | 0.000 | 0.000 | 0.000 | 0.000 | 0.000 | 0.000 | 0.000 | 0.000 | 0.000 | 0.000 |
| | =2 | 0.000 | 0.000 | 0.000 | 0.000 | 0.000 | 0.000 | 0.000 | 0.000 | 0.000 | 0.000 |
| | =3 | 0.000 | 0.630 | 0.795 | 0.955 | 0.945 | **1.000** | **1.000** | 0.950 | **1.000** | 0.950 |
| | =4 | 0.000 | 0.320 | 0.185 | 0.045 | 0.055 | 0.000 | 0.000 | 0.020 | 0.000 | 0.050 |
| | ≥5 | 1.000 | 0.050 | 0.020 | 0.000 | 0.000 | 0.000 | 0.000 | 0.030 | 0.000 | 0.000 |

**Table A5.** Distributions of the estimated component number $\hat{r}$ for Scenario 3 (r = 3).

| $N_i$ | $\hat{r}$ | AIC$_{PACE}$ | AIC | BIC | PC$_{p1}$ | IC$_{p1}$ | AIC$_{PACE}^{2022}$ | BIC$_{PACE}^{2022}$ | fpca | BSFDA | BSFDA$^{Fast}$ |
|---|---|---|---|---|---|---|---|---|---|---|---|
| 5 | $\leq 1$ | 0.000 | 0.000 | 0.335 | 0.000 | 0.000 | 0.000 | 0.000 | 0.000 | 0.000 | 0.000 |
|  | =2 | 0.025 | 0.035 | 0.260 | 0.220 | 0.005 | 0.000 | 0.005 | 0.000 | 0.000 | 0.025 |
|  | =3 | 0.005 | 0.720 | 0.325 | 0.640 | 0.590 | 0.320 | 0.400 | 0.450 | **0.995** | 0.945 |
|  | =4 | 0.130 | 0.170 | 0.080 | 0.075 | 0.280 | 0.640 | 0.565 | 0.360 | 0.005 | 0.030 |
|  | $\geq 5$ | 0.840 | 0.075 | 0.000 | 0.065 | 0.125 | 0.030 | 0.030 | 0.190 | 0.000 | 0.000 |
| 10 | $\leq 1$ | 0.000 | 0.000 | 0.005 | 0.000 | 0.000 | 0.000 | 0.000 | 0.000 | 0.000 | 0.000 |
|  | =2 | 0.015 | 0.000 | 0.035 | 0.000 | 0.000 | 0.000 | 0.000 | 0.000 | 0.000 | 0.000 |
|  | =3 | 0.000 | 0.580 | 0.770 | 0.965 | 0.665 | 0.740 | 0.755 | 0.440 | **0.995** | 1.000 |
|  | =4 | 0.000 | 0.400 | 0.145 | 0.030 | 0.320 | 0.260 | 0.245 | 0.380 | 0.005 | 0.000 |
|  | $\geq 5$ | 0.985 | 0.020 | 0.045 | 0.005 | 0.015 | 0.000 | 0.000 | 0.180 | 0.000 | 0.000 |
| 50 | $\leq 1$ | 0.000 | 0.000 | 0.000 | 0.000 | 0.000 | 0.000 | 0.000 | 0.000 | 0.000 | 0.000 |
|  | =2 | 0.000 | 0.000 | 0.000 | 0.000 | 0.000 | 0.000 | 0.000 | 0.000 | 0.015 | 0.000 |
|  | =3 | 0.000 | 1.000 | 0.775 | 1.000 | 1.000 | **1.000** | **1.000** | 0.765 | 0.980 | 0.920 |
|  | =4 | 0.000 | 0.000 | 0.200 | 0.000 | 0.000 | 0.000 | 0.000 | 0.110 | 0.005 | 0.050 |
|  | $\geq 5$ | 1.000 | 0.000 | 0.025 | 0.000 | 0.000 | 0.000 | 0.000 | 0.125 | 0.000 | 0.030 |

**Table A6.** Distributions of the estimated component number $\hat{r}$ for Scenario 4 (r = 3).

| $N_i$ | $\hat{r}$ | AIC$_{PACE}$ | AIC | BIC | PC$_{p1}$ | IC$_{p1}$ | AIC$_{PACE}^{2022}$ | BIC$_{PACE}^{2022}$ | fpca | BSFDA | BSFDA$^{Fast}$ |
|---|---|---|---|---|---|---|---|---|---|---|---|
| 5 | $\leq 1$ | 0.000 | 0.000 | 0.315 | 0.000 | 0.000 | 0.000 | 0.000 | 0.000 | 0.000 | 0.000 |
|  | =2 | 0.015 | 0.020 | 0.180 | 0.160 | 0.015 | 0.000 | 0.000 | 0.000 | 0.000 | 0.000 |
|  | =3 | 0.015 | 0.710 | 0.410 | 0.640 | 0.560 | 0.515 | 0.575 | 0.370 | **1.000** | 0.975 |
|  | =4 | 0.145 | 0.185 | 0.070 | 0.095 | 0.260 | 0.450 | 0.390 | 0.515 | 0.000 | 0.025 |
|  | $\geq 5$ | 0.825 | 0.085 | 0.025 | 0.105 | 0.165 | 0.035 | 0.035 | 0.115 | 0.000 | 0.000 |
| 10 | $\leq 1$ | 0.000 | 0.000 | 0.010 | 0.000 | 0.000 | 0.000 | 0.000 | 0.000 | 0.000 | 0.000 |
|  | =2 | 0.000 | 0.000 | 0.005 | 0.000 | 0.000 | 0.000 | 0.000 | 0.000 | 0.000 | 0.000 |
|  | =3 | 0.000 | 0.830 | 0.775 | 0.920 | 0.900 | 0.750 | 0.760 | 0.350 | **0.995** | 0.990 |
|  | =4 | 0.000 | 0.150 | 0.190 | 0.045 | 0.085 | 0.250 | 0.240 | 0.380 | 0.005 | 0.010 |
|  | $\geq 5$ | 1.000 | 0.020 | 0.020 | 0.035 | 0.015 | 0.000 | 0.000 | 0.270 | 0.000 | 0.000 |
| 50 | $\leq 1$ | 0.000 | 0.000 | 0.000 | 0.000 | 0.000 | 0.000 | 0.000 | 0.000 | 0.000 | 0.000 |
|  | =2 | 0.000 | 0.000 | 0.000 | 0.000 | 0.000 | 0.000 | 0.000 | 0.000 | 0.010 | 0.000 |
|  | =3 | 0.000 | 0.945 | 0.835 | **1.000** | **1.000** | **1.000** | **1.000** | 0.730 | 0.950 | 0.935 |
|  | =4 | 0.000 | 0.055 | 0.140 | 0.000 | 0.000 | 0.000 | 0.000 | 0.160 | 0.040 | 0.055 |
|  | $\geq 5$ | 1.000 | 0.000 | 0.025 | 0.000 | 0.000 | 0.000 | 0.000 | 0.110 | 0.000 | 0.010 |

**Table A7.** Distributions of the estimated component number $\hat{r}$ for Scenario 5 (r = 6).

| $N_i$ | $\hat{r}$ | AIC$_{PACE}$ | AIC | BIC | PC$_{p1}$ | IC$_{p1}$ | AIC$_{PACE}^{2022}$ | BIC$_{PACE}^{2022}$ | fpca | BSFDA | BSFDA$^{Fast}$ |
|---|---|---|---|---|---|---|---|---|---|---|---|
| 5 | $\leq 4$ | 0.005 | 0.165 | 0.835 | 0.580 | 0.060 | 0.000 | 0.000 | 0.010 | 0.000 | 0.060 |
|  | =5 | 0.005 | 0.330 | 0.020 | 0.345 | 0.335 | 0.575 | 0.590 | 0.010 | 0.075 | 0.515 |
|  | =6 | 0.705 | 0.470 | 0.090 | 0.070 | 0.545 | 0.425 | 0.410 | 0.855 | **0.925** | 0.160 |
|  | =7 | 0.245 | 0.035 | 0.050 | 0.005 | 0.060 | 0.000 | 0.000 | 0.115 | 0.000 | 0.160 |
|  | $\geq 8$ | 0.040 | 0.000 | 0.005 | 0.000 | 0.000 | 0.000 | 0.000 | 0.010 | 0.000 | 0.105 |
| 10 | $\leq 4$ | 0.005 | 0.000 | 0.000 | 0.000 | 0.000 | 0.000 | 0.000 | 0.000 | 0.000 | 0.000 |
|  | =5 | 0.000 | 0.000 | 0.030 | 0.145 | 0.000 | 0.425 | 0.425 | 0.000 | 0.000 | 0.000 |
|  | =6 | 0.065 | 0.570 | 0.525 | 0.775 | 0.705 | 0.575 | 0.575 | 0.500 | **1.000** | 0.930 |
|  | =7 | 0.475 | 0.280 | 0.165 | 0.020 | 0.185 | 0.000 | 0.000 | 0.405 | 0.000 | 0.035 |
|  | $\geq 8$ | 0.455 | 0.150 | 0.030 | 0.060 | 0.110 | 0.000 | 0.000 | 0.095 | 0.000 | 0.035 |
| 50 | $\leq 4$ | 0.000 | 0.000 | 0.005 | 0.000 | 0.000 | 0.000 | 0.000 | 0.000 | 0.000 | 0.000 |
|  | =5 | 0.065 | 0.000 | 0.000 | 0.000 | 0.000 | 0.130 | 0.130 | 0.005 | 0.000 | 0.000 |
|  | =6 | 0.000 | 0.260 | 0.590 | 0.980 | 0.965 | 0.870 | 0.770 | 0.695 | **0.995** | 0.925 |
|  | =7 | 0.000 | 0.405 | 0.325 | 0.010 | 0.035 | 0.000 | 0.000 | 0.250 | 0.005 | 0.045 |
|  | $\geq 8$ | 0.935 | 0.335 | 0.080 | 0.010 | 0.000 | 0.000 | 0.000 | 0.050 | 0.000 | 0.030 |

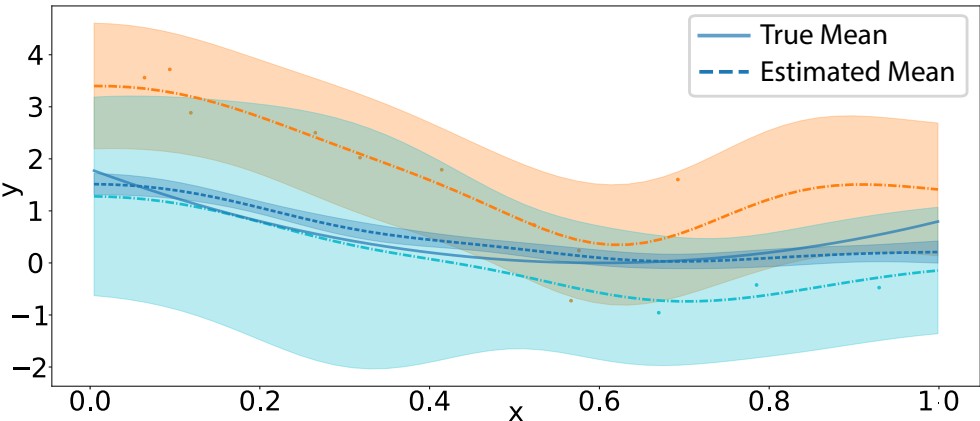

**Figure A2.** Application of the proposed BSFDA to the simulation benchmark from [21], illustrating the true mean function (blue), the observed measurements from two functions sampled at different densities (light blue for sparse, orange for dense), and the corresponding functional estimates with 95% truncated uncertainty intervals.

Appendix F.1.1. Performance of LFRM

To compare the latent factor regression model (LFRM) [18] as a dimension reduction model to ours, i.e., Bayesian scalable functional data analysis (BSFDA), we set the covariates in LFRM to zero, thus assigning standard Gaussian priors to the latent variables, analogously to our approach. We followed the simulation benchmark in [21] to select the number of components, focusing on Scenario 1 with 50 measurements per function (the densest data). Because LFRM does not estimate a mean function, we omitted the mean from the simulation run here.

**The following hyperparameters of LFRM need to be determined:**

- Gamma prior for white noise and correlated noise;
- Length scale;
- Number of basis functions;
- Number of iterations.

LFRM, with its default white noise prior, correctly identified the white noise variance (true value 0.2) in all tests. We thus retained this default. We tested different Gamma priors for correlated noise: the default prior, a noninformative-like (vagor) prior (same mean but 100 times the variance) and a low-noise prior (same variance but 100 times the mean). We maintained the number of locations for basis functions at 10, which is the default setting. For the length scale in LFRM, we first used the best estimate from our cross-validation (CV). We then tried all 10 CV-selected length scales, producing 100 basis functions in total. However, this required substantial time, so we performed only two repeated runs for this setting. We kept LFRM's default of 5000 burn-in iterations (25,000 total) with thinning at intervals of 5, verifying convergence through trace plots in line with [18]. Meanwhile, BSFDA was run 200 times as in Section 5.1, LFRM (10 length scales) two times, and all other settings 10 times.

Across repeated trials, LFRM consistently overestimated the true number of components (which was 3). Specifically,

- Standard LFRM estimated 10–14 components;
- LFRM with 10 length scales estimated 6–8 components;
- LFRM with a low-correlated-noise prior estimated 8–15 components;
- LFRM with a noninformative-like correlated-noise prior estimated 10–14 components.

In contrast, our method BSFDA produced a clear gap in the distribution of the precision parameters, effectively separating effective dimensions from redundant ones.

Several factors may explain LFRM's performance.

- **Correlated noise interference:** The correlated noise can obscure the true signal.
- **Prior specification:** LFRM's precision parameter priors are potentially less noninformative and not as sparse as those sparse Bayesian learning priors [30] in BSFDA.
- **Element-wise vs. column-wise precision:** The element-wise precision parameters in LFRM might compensate in a way that reduces the overall sparsity.

*Appendix F.2. Variational Inference vs. MCMC*

We conducted experiments using both Gibbs sampling (MCMC) and mean field approximation (variational inference) for the Bayesian PCA simulation [28] under varying noise levels, assuming that the true noise was known. In our experiments, "satisfactory estimation" was defined as the point when the fourth smallest precision (i.e., the inverse of variance) was at least 100 times smaller than the fifth—indicating that the four true signal dimensions (with variances [5, 4, 3, 2]) had been correctly identified. For computational tractability, we capped VI at 200,000 iterations (approximately 200 s) and MCMC sampling at 20,000 iterations (about 20 min), with a burn-in period of 200 iterations and thinning set to 10.

Figure A3 illustrates the runtime for VI and MCMC to identify the correct components. Our key findings are as follows:

1. When the noise level was close to the signal, neither MCMC or VI found the true dimension in the limited iterations (and probably never would have), because the data were heavily polluted.

2. As the noise level decreased toward zero, the number of iterations (and runtime) required for satisfactory estimation increased dramatically; VI began to fail around a noise level of $1 \times 10^{-4}$ and MCMC sampling around $1 \times 10^{-3}$, within the set time constraints.

3. Across the 10 noise levels (about $3 \times 10^{-3}$ to $2 \times 10^{-1}$) where both successfully identified the correct dimensionality, VI was consistently completed much faster than MCMC sampling. VI was $85.57 \pm 50.24$ times faster on average, in the range of 32.46 to 189.12.

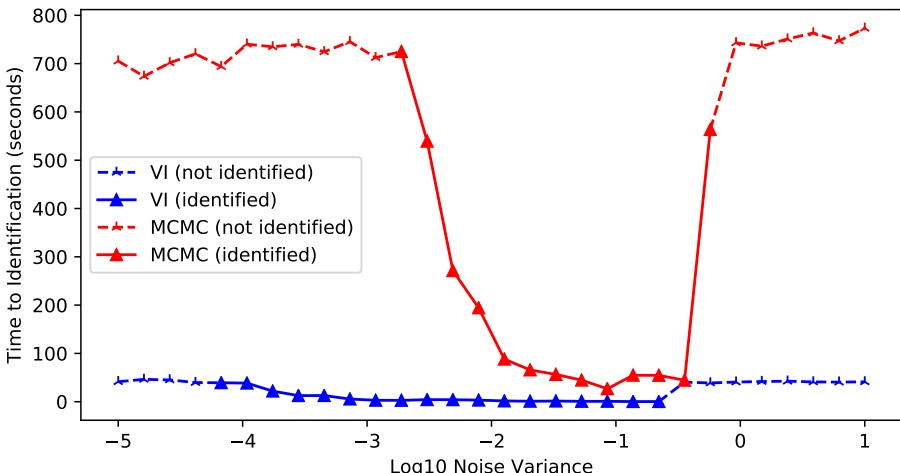

**Figure A3.** Time for variational inference and MCMC to identify the correct components in Bayesian PCA.

These results indicate that both MCMC sampling and VI become slower as the noise decreases due to strong dependencies in the posterior. We hypothesize this is because

both MCMC and VI suffer from the dependency introduced by low noise, which is a known long-standing issue with ongoing research methods, e.g., structured VI [38] or blocked/collapsed Gibbs samplers [40]. However, both MCMC and VI work well provided that there are sufficient iterations. This behavior suggests that the dependency induced by very low noise levels creates an optimization challenge rather than a fundamental modeling issue.

In summary, (1) VI is significantly faster than MCMC, (2) both methods slow down as the noise level decreases, and (3) both fail to recover the correct components when the noise is excessively high.

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
