# Peer review of "Integrated Model Selection and Scalability in Functional Data Analysis Through Bayesian Learning"

_algorithms, doi:10.3390/a18050254_

Round 1

Reviewer 1 Report

Comments and Suggestions for Authors

This paper tackles the weight determination challenge within functional principal component analysis (fPCA) using a Bayesian framework. However, several issues merit further clarification:

  • The paper would benefit from a more detailed explanation of the model presented in Equation (2), explicitly outlining how it connects with the principles of fPCA. In particular, why w_{ik} is in this particular form ? Is there any physical meaning ? Without a clear link to fPCA fundamentals, readers unfamiliar with the topic may struggle to follow the derivation and its implications.
  • Since the performance of fPCA is highly dependent on the choice of basis functions, the authors should include a discussion or practical guidelines for selecting an appropriate basis. Suggestions on criteria or methodologies for choosing the basis functions would greatly enhance the paper’s utility.
  • The manuscript does not clearly state how the weights are initialized within the proposed framework. Providing details on the initialization procedure and the rationale behind the chosen method would improve the reproducibility and understanding of the model.
  • In Section 3.2, the authors describe a scalable update strategy for active basis functions but do not justify why inactive components are not simply removed, as is common practice in sparse Bayesian learning. A discussion on the trade-offs between retaining versus removing these components would help clarify this design choice.
  • [R1] is a related paper published at ICML24. It investigates similar ideas through Bayesian nonparametric factorization for forecasting high-dimensional functional time series. Please clarify the difference of this paper to [R1].
  • For 2D data, what is the fundamental difference between the idea of this paper and treating the data as discrete but introduce a smooth (e.g., graph) prior, like that in [R2] ?

[R1] Deep Functional Factor Models: Forecasting High-Dimensional Functional Time Series via Bayesian Nonparametric Factorization

[R2] Bayesian Low-rank Matrix Completion with Dual-graph Embedding: Prior Analysis and Tuning-free Inference

Author Response

Thank you for your constructive reviews. We have prepared the point-by-point response for you in the attachment.

Reviewer 2 Report

Comments and Suggestions for Authors

"Integrated Model Selection and Scalability in Functional Data Analysis through Bayesian Learning" is an interesting text in which the authors, in their own words "propose a novel Bayesian framework with a nonparametric kernel expansion and a sparse prior, enabling direct modeling of measured data and avoiding the artificial biases from regridding".

The objective is reasonably described and the motivation for its study is well presented and substantiated in detail, although not with sufficient clarity. Fundamentally this has to do with the following aspects of the Introduction:

-The literature review

 It should be noted that in an article of this type, the literature review is not an objective, but rather a support for the presentation of the problem to be solved and the motivation to study it. Therefore, it does not have to be exhaustive, but rather necessary and sufficient. So, here it is very oversized, and the references are often "cited in a package" (see examples in lines 75 and 76) which is not at all enlightening for readers. Most of the time, one reference is enough, the one that is most enlightening. Therefore, the Bibliography is also unnecessarily large, which discourages readers from consulting it. Like this. I suggest that authors scrutinize the literature review, to avoid redundancy of references (and "packaged citations"). Then carry out another review with the aim of eliminating unnecessary self-citations. And, in this way, shorten and improve the quality of the Bibliography.

-The text that goes from lines 157 to 165

This text needs to be remodeled. Describing the structure of the rest of the text, the authors must indicate the sections and what is done in each one. Only in this way can it be a guide for readers.

In sections 2. Formulation, 3. Methods, and 4. Faster Variant the work is then developed. It is executed in an orderly and rigorous way, with a lot of mathematical foundations. The proofs are relegated to the appendix, which is a very good idea, to allow a more accurate reading of the important part of an extremely long text. As for the proofs, the Halmos end of proof symbol should be used: â–¡, instead of QED. It is more indicative and does not get lost in the middle of the text. In these three sections, I would highlight 2. Formulation for its clarity and constructive character.

In 5. Results, experiments are presented with a view to illustrating the performance of the tool created in this paper. It is worth highlighting the criteria for choosing the experiments and the good presentation and discussion of the results.

In 6. Discussion, which in my opinion should be called 6. Discussion and Conclusions, the current text is fine with one caveat: The authors do not present a critical look at their work here; In addition to the strengths, which are generally described, it is necessary to identify possible weaknesses that can provide clues for future research. Furthermore, with the number of tests mentioned in this work, weaknesses will certainly be found...

Author Response

(The authors gave the same response as above.)

Reviewer 3 Report

Comments and Suggestions for Authors

Report on "Integrated Model Selection and Scalability in Functional Data
Analysis through Bayesian Learning"

summary:
Functional data, including one-dimensional curves and higher-dimensional 1
surfaces, have become increasingly prominent across scientific disciplines. They offer a 2
continuous perspective that captures subtle dynamics and richer structures compared to 3
discrete representations, thereby preserving essential information and facilitating more 4
natural modeling of real-world phenomena, especially in sparse or irregularly sampled 5
settings. A key challenge lies in identifying low-dimensional representations and estimat- 6
ing covariance structures that capture population statistics effectively. We propose a novel 7
Bayesian framework with a nonparametric kernel expansion and a sparse prior, enabling 8
direct modeling of measured data and avoiding the artificial biases from regridding. Our 9
method, Bayesian scalable functional data analysis (BSFDA), automatically selects both sub- 10
space dimensionalities and basis functions, reducing computational overhead through an 11
efficient variational optimization strategy. We further propose a faster approximate variant 12
that maintains comparable accuracy but accelerates computations significantly on large- 13
scale datasets. Extensive simulation studies demonstrate that our framework outperforms 14
conventional techniques in covariance estimation and dimensionality selection, showing 15
resilience to high dimensionality and irregular sampling. The proposed methodology 16
proves effective for multidimensional functional data and showcases practical applicability 17
in biomedical and meteorological datasets. Overall, BSFDA offers an adaptive, continuous, 18
and scalable solution for modern functional data analysis across diverse scientific domains.

The proposed model in this manuscript is interesting. The manuscript is well-written and well-organized; however, it requires major revisions as follows:

Choice of Distributions Before Equation (2)

The manuscript considers Gamma and Normal distributions before equation (2). Could the authors clarify the rationale behind selecting these specific distributions?
Is it possible to consider alternative distributions? If so, what would be the impact on the model’s performance and generalizability?
Limiting Distribution of Estimators

The manuscript does not discuss the limiting distribution of the estimators. Could the authors provide theoretical insights or empirical justifications regarding their asymptotic behavior?
Reproducibility and Code Availability

To ensure the validity and reproducibility of the results, please provide all computational codes, including those used for plots, tables, simulations, and applications.
Addressing these points will strengthen the manuscript and enhance its theoretical and practical contributions.

Author Response

(The authors gave the same response as above.)

Reviewer 4 Report

Comments and Suggestions for Authors

The manuscript proposes a method called Bayesian scalable functional data analysis (BSFDA), an approach to large scale functional data analysis which automatically selects both subspace dimensionalities and basis functions, while implementing Bayesian variational inference to reducing computational cost. This manuscript makes significant contributions, is technically correct, and well written to warrant its eventual journal publication. Some revisions of the manuscript can be made to clarify its overall contributions.

Main Comments

  1. Section 3.1. Variational Bayesian Inference.
    Does this inference procedure underestimate parameter estimation uncertainty?
    I recall that the related literature recognizes that this is a limitation of the Variational Inference approach.

  2. While Section 6 Discusses the possibility of using robust principal components analysis (PCA) in future research, it would still help if Section 5 resents at least one result based on robust PCA to give some indication about how useful this robust method would be for the BSFDA framework. Besides, these days, easy to use software for robust PCA methods are commonplace. Though I would not characterized this as a required revision of the manuscript.
  3. Please provide a link to the relevant software code, so that readers can reproduce the results, equations, and algorithms presented in the manuscript.

Details

Line 127:  1.1. Contriubutions --> 1.1. Contributions

Line 144: 'while still being accurate'.  Please elaborate what you mean by 'accurate' Accuracy in what sense? How is accuracy defined. Please briefly mention.

I did not find any other typos in the manuscript.

Author Response

(The authors gave the same response as above.)

Round 2

Reviewer 1 Report

Comments and Suggestions for Authors

While the answers in the reply document are acceptable, the comparison with [R1] and [R2] has not been added in the revised paper. This does not help the readers to clearly see the contribution of this paper with respect to state-of-the-arts. Please add the comparison in the next version. 

Author Response

(The authors gave the same response as above.)

Reviewer 2 Report

Comments and Suggestions for Authors

The authors responded proficiently and very competently to the recommendations I made, therefore, in my opinion, in its present form, the text is ready for publication in Algorithms.

Author Response

Thank you for your constructive reviews and approval for publication.

Reviewer 3 Report

Comments and Suggestions for Authors

The revised manuscript is satisfactory.

Author Response

Thank you for your support. Because this comment didn't raise additional issues, we are not including new responses in this reply.

Round 3

Reviewer 1 Report

Comments and Suggestions for Authors

This version is good enough for acceptance.

Author Response

(The authors gave the same response as above.)
